# A Hierarchical Attractor Network Model of perceptual versus intentional decision updates

Anne Löffler [1,2✉], Anastasia Sylaidi[3], Zafeirios Fountas[4,5] & Patrick Haggard [2]

Changes of Mind are a striking example of our ability to flexibly reverse decisions and change our own actions. Previous studies largely focused on Changes of Mind in decisions about perceptual information. Here we report reversals of decisions that require integrating multiple classes of information: 1) Perceptual evidence, 2) higher-order, voluntary intentions, and 3) motor costs. In an adapted version of the random-dot motion task, participants moved to a target that matched both the external (exogenous) evidence about dot-motion direction and a preceding internally-generated (endogenous) intention about which colour to paint the dots. Movement trajectories revealed whether and when participants changed their mind about the dot-motion direction, or additionally changed their mind about which colour to choose. Our results show that decision reversals about colour intentions are less frequent in participants with stronger intentions (Exp. 1) and when motor costs of intention pursuit are lower (Exp. 2). We further show that these findings can be explained by a hierarchical, multimodal Attractor Network Model that continuously integrates higher-order voluntary intentions with perceptual evidence and motor costs. Our model thus provides a unifying framework in which voluntary actions emerge from a dynamic combination of internal action tendencies and external environmental factors, each of which can be subject to Change of Mind.

[1] Zuckerman Mind Brain Behaviour Institute, Columbia University, New York, NY, USA. [2] Institute of Cognitive Neuroscience, University College London, London, UK. [3] Spike AI Research Labs, London, UK. [4] Emotech Labs, London, UK. [5] Wellcome Centre for Human Neuroimaging, University College London, London, UK. ✉email: al3928@columbia.edu

People frequently change their minds about what to do. We may plan to go to the gym but end up watching TV, or we may cancel dinner plans after a tiring day at work. Initial action decisions are followed by continuous evaluation processes during which additional information is acquired and integrated with the initial intention. This updating can result in Changes of Mind (CoM), i.e., reversals of an initial decision. Most studies investigating CoM focussed on action choices driven by external, perceptual information[1–10]. For example, in the random-dot motion task, participants judge the direction of moving dots by reaching for a target that corresponds to the observed dot motion. When the perceptual evidence is noisy, movement trajectories occasionally indicate a CoM, e.g., the response is initiated towards the left, but is then redirected and ends in the right target (e.g., ref.[1]). This suggests that decision making continues after action initiation, allowing for updates of action decisions during ongoing movement execution.

However, few studies have considered CoM in the context of voluntary actions. Voluntary actions may be defined as actions in which people endogenously (internally) generate an intention regarding which of several actions to make[11]. Yet, in many cases, voluntary intentions need to be combined with external sensory inputs providing contextual information about how to act[12,13]. For example, if you want to go to a Japanese restaurant for dinner (endogenous intention), you will need to find the restaurant on a map (perceptual information) in order to know which way to go (motor action). We propose that Changes of Mind can occur with regard to both the perceptual and intentional components of voluntary action. This suggests two dissociable types of CoM. In the first type of CoM, decisions about external evidence may change, as in the random-dot motion task. For example, if you get lost on your way to the restaurant, you may ask someone for directions, thus obtaining new external information on how to act. In the second type of CoM, the action intention, or goal, may itself change—a process sometimes called goal-shifting[14]. For example, if you realise the Japanese restaurant is far away, you may decide to go to a nearby Italian restaurant instead.

Hence, voluntary actions rely on dynamic integration of multiple (internal and external) decision components, each of which can be subject to Change of Mind. In addition, voluntary actions are characterised by a hierarchal structure in which voluntary intentions are represented on a hierarchically higher level than sensorimotor information. Pacherie's hierarchical theory of intention[15] differentiates between higher-order distal intentions and lower-level motor intentions. Distal intentions specify the overarching goal of an action, i.e., what to do (e.g., going to a Japanese vs. Italian restaurant), whereas motor intentions specify how to implement the abstract intention into a specific motor action (e.g., turning left vs. right to get to the chosen restaurant). Flexible updates of lower-level motor intentions have been studied extensively. In double-step paradigms, for example, aimed movements must be rapidly updated based on changes in target locations[16–18]. By contrast, little is known about how and when people decide to pursue or abandon distal action goals that are generated internally instead of being instructed by external stimuli (but see ref.[13] for a recent study on goal reversals). This is surprising given that the decision to pursue vs. abandon one's own goals (e.g., to quit smoking) can have wide-ranging personal and social consequences[19].

In the current study, we tested the hypotheses that CoM about voluntary intentions depend on the strength of the initial intention and the cost associated with intention pursuit. For instance, a person should be more likely to pursue the intention of having sushi, the stronger that intention is[17,20,21]. Intentional strength in turn might depend on confidence regarding an internal decision[22] or choice values[23], e.g., a strong preference for sushi over pizza.

Yet, few goals are worth pursuing at *any* cost. Therefore, time- and effort-related movement costs might induce Changes of Intention, or at least make them more likely to occur[24,25]. In addition, we hypothesised that CoM may not only be accompanied by changes in objective action characteristics (e.g., changing movement trajectories), but may also shape our Sense of Agency (SoA)—the subjective experience of exerting control over actions and their outcomes[15,26]. Specifically, changing an ongoing movement could reduce SoA by making actions feel dysfluent[27,28]. Whether changing an endogenous intention would affect SoA is less clear. Previous research suggests that strong distal action goals boost SoA[29,30]. Consequently, deviations from initial intentions might decrease SoA[31]. In contrast, reconstructive theories view conscious intentions as retrospective confabulations[32]. People appear to experience actions as intentional, even when they were not part of an initial plan, or are not even their own[33,34]. On this view, reversals of endogenous intentions should not affect SoA.

Finally, we propose a new computational framework for CoM in voluntary action that successfully captured our experimental findings. Previous computational accounts of CoM have either used bounded accumulator models[1,7,8] or Attractor Network Models[2,3,5,6,35]. However, the existing models have been limited to decisions that are purely perception, and also unidimensional, in that they only rely on a single source of (sensory) evidence. By contrast, here we extend previous work by introducing a multimodal, hierarchically organised Attractor Network Model that continuously integrates multiple factors relevant to decision-making, namely: (1) higher-order, internal intentions, (2) external perceptual evidence, and (3) motor costs. Although some previous studies have considered how costs or rewards affect perceptual decisions, these studies treated reward/cost information as a static decision variable that simply biases choices or shifts decision thresholds across trials[8,36,37]. By contrast, in our model, each source of information provides independent information and needs to be integrated continuously and dynamically during the evolving decision process. Finally, our model accounts for the hierarchical organisation of voluntary actions[15,38,39] by combining an Attractor Network approach with noise reduction mechanisms derived from Hierarchical Gaussian Filters[40,41]. Specifically, we propose that higher-order intentions exert top-down noise control over lower-level sensorimotor processes. In line with this, previous studies have proposed that neural noise reduction may be a neurobiological marker of endogenous action control[42]. More broadly, others have proposed that higher-order brain areas representing abstract information (e.g., goals) may exert top-down control over lower-level sensorimotor areas by gating noisy inputs[43,44]. Thus, our model provides a neurobiologically-plausible framework through which higher-order endogenous intentions and lower-level sensorimotor information are flexibly and dynamically integrated over time, which can lead to different types of (intentional vs. perceptual) CoM.

## Results

In the current study, participants performed an adapted version of the random-dot motion task in which they had to integrate the perceptual decision about dot-motion direction (left/right) with an endogenous choice about which colour to paint the dots. Based on previous studies[1–8], we expected to observe perceptual CoM regarding the dot-motion direction. Importantly, the current paradigm allowed us to differentiate between trials with mere "perceptual Changes of Mind" ($CoM_P$) from trials with both a "perceptual + intentional Change of Mind" ($CoM_{P+I}$). In two experiments, we tested the hypotheses that CoM regarding the

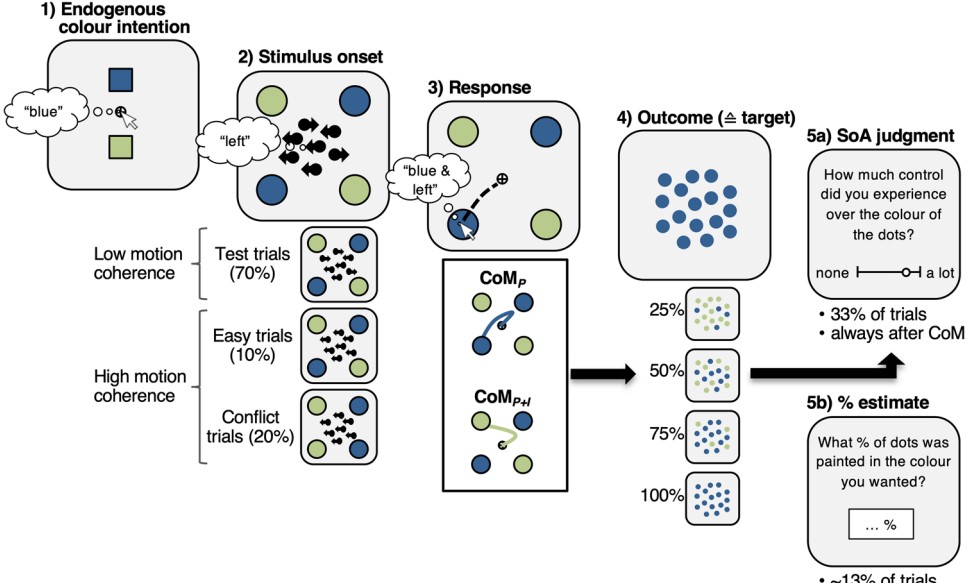

**Fig. 1 Behavioural task.** Participants generated an endogenous colour intention (**1**) that had to be integrated with the sensory input of the dot-motion stimulus (**2**). Responses were indicated by moving the cursor to the target that matched both the colour intention and dot-motion direction (**3**). Continuous movement trajectories were measured during response execution allowing for online classification of perceptual Changes of Mind (CoM$_P$) and perceptual + intentional Changes of Mind (CoM$_{P+I}$). Once participants reached the target, 25/50/75/100% of the dots were painted in the colour of the hit target (**4**). On some trials, participants were asked to provide Sense of Agency (SoA) judgements (**5a**) or to estimate the percentage of dots that matched their initial colour intention (**5b**).

initial intention is more frequent when intentions are weak (Exp. 1) and when the motor cost of intention pursuit is high (Exp. 2). In addition, in both experiments, subjective reports of Sense of Agency were obtained to investigate the effect of CoM on the phenomenology of action. Finally, a hierarchical Attractor Network Model is introduced that captures the dynamic decision processes underlying CoM in voluntary action.

**Experiment 1: Behavioural task.** Participants performed an adapted version of the random-dot motion task (Fig. 1). At the beginning of each trial, they freely chose between two colours (e.g., blue vs. green). Participants were instructed to say the chosen colour in their head, and on 10% of trials, were prompted to say their choice out loud. A random-dot motion stimulus and 4 targets, 2 of each colour, then appeared in pseudo-random locations. Targets appeared either 700–1000 ms before the dot-motion onset (early targets; 50% of trials), or at the same time as the dot-motion stimulus (late targets; 50% of trials). Using a touch pad, participants had to move the cursor to the target that matched both the perceived dot-motion direction and their endogenous colour choice (e.g., left-blue target). They were instructed to respond as fast and accurately as possible. The dot-motion stimulus disappeared as soon as participants initiated a response. However, due to sensorimotor delays, we expected to observe Changes of Mind about the dot-motion direction on some trials[1–4,7,8]. Once participants reached the target, 25/50/75/100% of dots were presented in the colour of the chosen target and participants were then asked how much control they experienced over the colour of the dots (SoA judgement) or what percentage of dots was painted in the colour they chose (% outcome estimate).

In the main condition of interest—test trials (70%)—targets of the same colour were presented in diagonally opposite locations (e.g., blue in top-right and bottom-left corner). Furthermore, motion coherence was low in test trials, with the precise value being determined individually prior to the experiment to ensure ~60% perceptual choice accuracy (see "Methods"). By contrast,

easy trials (10%) served as a baseline condition with high motion coherence (80% coherence). As expected, perceptual choice accuracy was significantly worse in test trials (M = 56.6%, SD = 9.1%) than in easy trials (M = 93.4%, SD = 7.0%, t(16) = 20.13, p < 0.001, d = 4.88) and RTs were significantly slower in test (M = 570.5 ms, SD = 58.3 ms) than in easy trials (M = 534.2 ms, SD = 41.5 ms, t(16) = 3.99, p = 0.001, d = 0.97).

**Changes of Mind.** Next, we checked whether difficulty of perceptual decisions in test trials resulted in perceptual CoM. In analogy to the original random-dot motion task, CoM was defined as a decision reversal regarding the dot-motion direction (e.g., initial response towards a target on the right followed by a switch to a left target). As expected, such perceptual changes were significantly more frequent in test (M = 7.64%, SD = 6.74%) compared to easy trials (M = 1.28%, SD = 2.33%; b = 1.84, 95% CI [1.19–2.63], OR = 6.27, $\chi^2$(1) = 45.69, p < 0.001). Furthermore, in line with previous findings (e.g., ref. [1]), the majority of CoM in test trials (M = 60.9%, SD = 16.5%, t(16) = 2.72, p = 0.015, d = 0.66) corrected an initial perceptual error (i.e., the response would have been an error had no CoM occurred). This suggests that perceptual CoM in our task was driven by continuous integration of sensory evidence after an initial response had already been initiated.

More importantly, the current paradigm allowed us to differentiate between trials in which difficult perceptual decisions only resulted in (1) a "perceptual CoM" (CoM$_P$) while the initial colour intention was pursued (e.g., switch from right-blue to left-blue target), or (2) a "perceptual + intentional CoM" (CoM$_{P+I}$) that additionally involved a change with respect to the initial intention (e.g., switch from right-blue to left-green target). Given the diagonal target arrangement, intention pursuit (CoM$_P$ without change of intention) was associated with longer movement paths than switching to the neighbouring target of different colour (CoM$_{P+I}$). Hence, when participants changed their mind about the dot-motion direction, they could save motor costs by switching to the target that did not match their initial colour

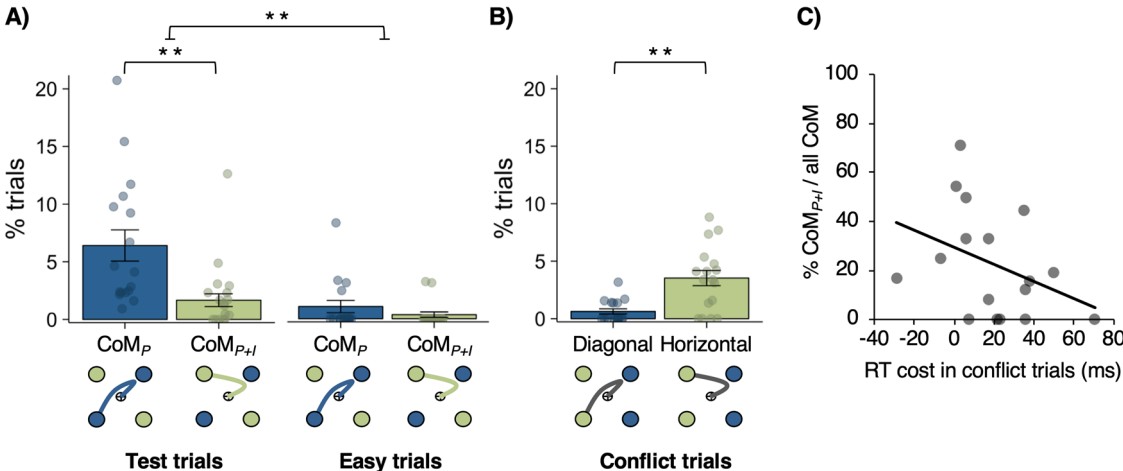

**Fig. 2 Changes of Mind (CoM) in Exp. 1 ($n = 17$ participants). A** Percentage of trials classified as perceptual CoM (CoM$_P$; blue) and perceptual + intentional CoM (CoM$_{P+I}$; green) in test and easy trials. **B** Percentage of conflict trials with diagonal (blue) and horizontal (green) movement corrections of partial errors that were induced by mismatches between colour intentions and dot-motion direction. In both **A** and **B**, data are presented as mean values ± 1 SEM (**$p < 0.001$). Dots represent data points from individual participants. Statistical significance was obtained using likelihood ratio tests to compare logistic mixed-effects regression models with vs. without the fixed effects of interest (see "Methods"). No correction for multiple comparisons was performed since all comparisons are orthogonal and were planned prior to data collection. **C** Correlation across participants between RT costs in conflict trials and frequency of CoM$_{P+I}$ (relative to overall percentage of all CoM), $\rho(15) = -0.50$, $p = 0.043$, two-tailed. Source data are provided as a Source Data file.

choice. Supplementary Figure 1 shows single-trial movement trajectories for CoM$_P$ and CoM$_{P+I}$ of an individual participant. The average frequency of each type of CoM in test vs. easy trials is shown in Fig. 2A.

In order to investigate the relative frequency of CoM$_P$ vs. CoM$_{P+I}$, only trials with CoM were included in a mixed-effects logistic regression analysis with CoM$_P$ (0) vs. CoM$_{P+I}$ (1) as outcome variable and trial condition (easy/test) as a fixed effect. The effect of trial condition was not significant ($b = -0.13$, 95% CI $[-1.93$ to $1.67]$, OR $= 0.88$, $\chi^2(1) = 0.02$, $p = .890$), suggesting that perceptual uncertainty only affected whether or not a perceptual CoM occurred, but did not affect whether participants changed their mind about their colour intention. Interestingly, in test trials, CoM$_P$ was more frequent (M $= 5.9\%$, SD $= 5.5\%$) than CoM$_{P+I}$ (M $= 1.7\%$, SD $= 2.2\%$), as indicated by an intercept that was significantly lower than 0 ($b_0 = -1.56$, 95% CI $[-2.47$ to $-0.89]$, OR $= 0.2$, $z = -4.42$, $p < 0.001$). Hence, when changing a perceptual decision based on new sensory evidence, participants pursued their colour intention more often than switching to the target of different colour, despite the extra motor costs of diagonal compared to horizontal movement adjustments. A similar trend was observed in easy trials, although overall CoM frequency was low in this condition and the intercept was not significantly different from 0 ($b_0 = -1.43$, 95% CI $[-3.30$ to $0.43]$, OR $= 0.24$, $z = -1.50$, $p = 0.132$).

**Did participants generate initial colour intentions?** As mentioned above, in 90% of trials, participants were not asked to verbalise their colour choice at trial start, and instead, colour choices were inferred from movement trajectories (e.g., movements initiated toward green target reflect colour choice of green). This minimised demand characteristics that might discourage participants from changing their initial colour choice when having to say it out loud. Yet, it raises the question whether participants indeed chose a colour at trial start (frame 1, Fig. 1), or instead, delayed their decision to stimulus onset (frame 2, Fig. 1). The fact that, overall, participants were reluctant to giving up their colour intentions suggests that they assigned a relatively

high importance to colour choices in the task. In addition, we included conflict trials (20%) to further investigate whether participants indeed generated colour intentions at trial start, even on trials where they did not have to verbalise their choice. In conflict trials, motion coherence was as high as in easy trials (80% coherence), but both targets of the same colour were on the same side of the screen. Consequently, in roughly 50% of conflict trials, there was a mismatch between intentional colour choice and dot-motion direction (e.g., a participant had chosen blue, both blue targets appeared on the right side, but the dots moved to the left). In this case, participants were instructed to respond according to the dot motion, and hence, move to a target that did not match their own colour choice. If participants did indeed generate initial colour intentions, colour-motion mismatches would induce response conflict. Consequently, reaction times (RTs) and error rates would, on average, be higher in conflict than easy trials even though the perceptual decision was equally easy in both conditions. These performance costs would be driven by trials in which conflict occurred. However, the inference is based on mean performance, and does not require explicitly identifying which specific trials involved conflict and which did not. Note that no response costs would be observed if participants did not make colour choices at trial start since, in that case, participants would simply respond based on dot motion direction without any conflict induced by colour choices.

We found that in conflict trials, RTs were indeed significantly slower (M $= 549.7$ ms, SD $= 45.8$ ms) and perceptual choice accuracy was descriptively lower (M $= 90.5\%$, SD $= 7.2\%$) than in easy trials (RTs: M $= 534.2$ ms, SD $= 41.5$ ms, $t(16) = 2.51$, $p = 0.023$, $d = 0.61$; accuracy: M $= 94.1\%$, SD $= 6.8\%$, $t(16) = 2.11$, $p = 0.051$, $d = 0.51$). These response costs were present even in early-target trials (RT cost: M$_\Delta = 29.7$ ms, SD$_\Delta = 32.7$ ms, $t(16) = 3.74$, $p = 0.002$, $d = 0.91$; accuracy cost: M$_\Delta = 3.63\%$, SD$_\Delta = 7.27$, $t(16) = 2.06$, $p = 0.056$, $d = 0.50$). This suggests that response costs in conflict trials were not simply driven by participants being surprised about the uncommon target configuration. Instead, participants seemed to generate initial colour intentions, which on some conflict trials did not match the dot-motion direction, hence inducing response costs.

In addition to errors and slowing of RTs, movement trajectories in conflict trials indicated that participants occasionally initiated a response towards one target, but then adjusted the movement to end in another target, similar to CoM in test/easy trials (Fig. 2B). However, note that CoM in conflict trials were not interpreted in the same way as CoM in test/easy trials, given the differences in target configuration and instructions. Instead, in conflict trials, movement adjustments presumably reflect partial errors in colour-motion mismatch trials. That is, participants initiated responses toward their chosen colour, but then corrected themselves to respond according to the dot motion as instructed. In line with this, we found that corrective movements in conflict trials occurred significantly more often than perceptual CoM in easy trials, despite dot-motion coherence being matched in both conditions ($b = 1.09$, 95% CI [0.39–1.92], OR = 2.97, $\chi^2(1) = 9.93$, $p = 0.002$). This confirms that corrections in conflict trials were not induced by perceptual noise, but instead, can be attributed to conflict induced by mismatches between colour intention and perceptual input.

Finally, in conflict trials, participants could correct ongoing movements in two ways (Fig. 2B) by either switching to the diagonally opposite target, or the horizontally neighbouring target. A mixed-effects logistic regression showed that participants overall preferred horizontal over diagonal movement corrections in conflict trials ($b_0 = 1.74$, 95% CI [0.97–3.24], OR = 5.71, $z = 3.46$, $p < 0.001$). This suggests that participants were sensitive to the higher motor costs of diagonal movement corrections and preferred less costly horizontal corrections in conflict trials. The fact that, in test trials, participants preferred diagonal ($CoM_P$) over horizontal ($CoM_{P+I}$) movements showed that most participants were willing to overcome these motor costs to pursue their colour intentions when possible. However, the frequency of $CoM_P$ relative to all CoM in test trials varied across participants (M = 77.4%, SD = 22.1%). Thus, participants may have differed in how much weight they assigned to the colour choice relative to the perceptual task, and hence, how strong their colour intentions were.

**Effect of intentional strength on changes of intention**. While participants were instructed to generate colour intentions at trial start, they were not explicitly told that they had to maintain their initial colour choice throughout the trial. In particular, participants did not receive any instructions as to whether they should stick with their initial colour intention when they changed their mind about the dot-motion direction. Instead, in trials with perceptual CoM, the decision between (a) switching to the other target of the same colour or (b) switching to the nearby target of different colour was endogenous. This enabled us to capture spontaneous changes with regard to the initial intention. Furthermore, the importance of pursuing colour choices was ambiguous on purpose to allow us to capture inter-individual differences in intentional strength—that is, the importance, or weight, a given participant assigned to the colour choice relative to the perceptual choice. For example, a participant who considered colour choices to have little task relevance, would generate weaker intentions, and should be less likely to stick with an initial colour choice when facing the higher cost of colour pursuit in CoM trials.

We tested whether participants with stronger colour intentions showed fewer changes of intention ($CoM_{P+I}$) relative to purely perceptual CoM ($CoM_P$). Individuals' average response costs in conflict compared to easy trials served as an indicator of the strength of colour intention, with higher response costs indicating stronger intentions. Since only 9/17 participants made errors in conflict trials, we focused on RT costs as an indicator of the strength of colour intention. As predicted, we found that, across participants, higher RT costs in conflict trials were indeed associated with a lower frequency of $CoM_{P+I}$ out of all $CoM\left(\frac{\%CoM_{P+I}}{\%CoM_P + \%CoM_{P+I}}\right)$ in test trials (Spearman's $\rho(15) = -0.50$, $p = 0.043$, 95% CI: [−0.07 to −0.76]; Fig. 2C). This suggests that participants with stronger colour intentions (and thus higher conflict costs) were less likely to change their intentions in test trials.

**Potential effect of target confusion**. One potential alternative interpretation of trials classified as $CoM_{P+I}$ needs to be addressed. It is possible that participants switched to a target of different colour because their initial movement was erroneously directed towards a target that did not match their colour choice due to difficulties in target detection. In that case, curved trajectories would not represent a genuine change of the initial intention, but rather a correction of an initial response selection error. However, a significant number of $CoM_{P+I}$ was observed even in test trials with early target onset in which participants had 700–1000 ms to identify target-colour locations (M = 1.37%, SD = 1.85%, $t(16) = 3.06$, $p = 0.008$, $d = 0.74$), thus rendering it unlikely that these $CoM_{P+I}$ were caused by target confusion. Moreover, participants were rewarded based on perceptual choice only, and hence, switching between horizontal targets merely based on colour would result in a potential monetary loss. Instead, if target confusion occurred, participants should switch to the target of different colour on the *same* side of the screen (rather than to the horizontally neighbouring target). Importantly, these vertical movement corrections were not classified as $CoM_{P+I}$ (see Section "Vertical movement corrections" below). However, vertical movement corrections were indeed observed on 3.24% of test trials (SD = 2.56%) and occurred significantly more often in late- than early-target onset test trials ($b = 1.18$, 95% CI [0.79–1.59], OR = 3.25, $\chi^2(1) = 38.76$, $p < 0.001$). This suggests that, when participants confused target colours due to difficulties in target detection (e.g., due to late target onset), they switched to the target of different colour that was on the same side of the screen. By contrast, switches to the horizontally neighbouring target ($CoM_{P+I}$) presumably represent genuine decision reversals with respect to the initial intention that were caused by a perceptual CoM about the dot-motion direction, rather than target confusion.

**Vertical movement corrections**. We had not initially predicted any vertical movement corrections since our hypotheses focused on CoM that were induced by noise in the random-dot motion stimulus, which in our task involved only the left-right dimension. The fact that vertical movement corrections occurred more often in late than early targets suggests that they may reflect a type of lower-level motor CoM where movements were updated after an initial response selection error caused by difficulty in target selection. Alternatively, vertical movement corrections may reflect true changes in the initial colour intention in the absence of a perceptual CoM. While this is possible, it seems unlikely that this is true for the majority of vertical movement corrections given that 1) initial colour choices were not made under time pressure, 2) participants' colour intentions appeared to be strong on average, and 3) vertical movement corrections incurred a cost without increasing rewards (as opposed to $CoM_{P+I}$, which increases rewards by increasing perceptual accuracy). Moreover, if vertical changes reflected real changes of colour intentions, one would expect the frequency of these changes to be negatively correlated with RT costs in conflict trials, similarly to the negative correlation of $CoM_{P+I}$ with RT costs. That is, the stronger participants' colour intentions are, the higher their performance

costs in conflict trials would be, and the less likely they should be to change their mind about the initial colour intention. However, as opposed to CoM$_{P+I}$, we did not find any negative correlations between individuals' RT costs in conflict trials and the frequency of vertical changes in test trials (Exp. 1: $r_S = 0.199$, $p = 0.445$; Exp. 2: $r_S = 0.238$, $p = 0.374$) nor in easy trials (Exp. 1: $r_S = -0.257$, $p = 0.319$; Exp. 2: $r_S = 0.359$, $p = 0.172$). Thus, while it is possible that some vertical changes reflected true changes of intention, we do not have any evidence to suggest that this is true for the majority of vertical movement corrections. Instead, we propose that they largely reflect a lower-level motor CoM in which the initial response was erroneously directed to a target that did not correspond to participants' colour intention.

**Summary and discussion Exp. 1**. In our paradigm, two types of Changes of Mind in voluntary action were dissociated based on movement trajectories: (1) "perceptual CoM" in which participants changed decisions about exogenous stimuli, requiring sensorimotor updates while the initial endogenous intention was maintained and (2) "perceptual + intentional CoM" where movement updates did not only reflect decision reversals about exogenous stimuli, but additionally, a change of the initial endogenous intention. Although the overall frequency of CoM was relatively low, the observed 7.6% CoM in test trials is clearly comparable with previous studies reporting 2–15% CoM in trials with similar motion coherences[1,4,7]. Further, several areas of cognitive theory, e.g., memory research, rely strongly on data from infrequent errors—no doubt because errors are highly informative about the processes generating performance[45]. Finally, the frequency of CoM varied systematically across trial conditions. Specifically, in line with previous studies on perceptual decision reversals (e.g., ref. [1]), we found that CoM was more frequent when sensory noise was high and when the initial perceptual decision was erroneous. Crucially, we found that the need to update an ongoing movement based on new sensory information occasionally induced a change in the higher-order goal intention regarding colour choice, suggesting that sensorimotor reprogramming triggered a re-evaluation of the initial goal itself.

Overall, the frequency of intention reversals was lower than that of perceptual decision reversals, suggesting that, within the context of the current task, the endogenous action goal occupied a primary place in the informational hierarchy, relative to the secondary place occupied by the sensory dot-motion stimulus. However, we further showed that the degree of such prioritisation of endogenous goals over sensory evidence varied across participants. Specifically, the frequency of changes of intention was inversely related to the strength of participants' initial intentions. That is, some participants generated stronger colour intentions as indicated by a high performance cost under endogenous-exogenous conflict. These participants were more likely to pursue their initial intention when adjusting an ongoing movement due to perceptual CoM. Inter-individual differences in intentional strength reflected the importance, or weight, participants assigned to colour choices in the task, relative to the dot-motion judgement. These differences in turn were presumably caused by differences in demand characteristics based on individuals' interpretation of the instructions[46], or the subjective value participants assigned to the colours[23], e.g., based on preferences for certain colours. Note that our design did not allow us to capture variability in intentional strength on a trial-by-trial basis, but rather, the strength of the colour choices throughout the task. However, intentions can vary in strength within people, and it is likely that this would affect the likelihood of a person changing an intention in a given situation[17,47].

In a second experiment, we manipulated the trade-off between intentions and their associated motor costs on a trial-by-trial basis by varying target distances within participants. We hypothesised that the frequency of intention reversals increases when the cost of pursuing the initial intention is high. This would provide more direct evidence that CoM regarding higher-order intentions can be caused by motor costs associated with such intentions over time. Furthermore, it would establish a means to experimentally induce a higher frequency of CoM about voluntary intentions.

**Experiment 2: effect of motor costs on intention reversals**. The task was identical to Exp. 1 with the following exceptions (Fig. 3): Target distance varied on a trial-by-trial basis within participants in order to manipulate the relative motor cost of intention pursuit after a perceptual CoM (Fig. 3A). Specifically, longer travel distances incur higher motor costs due to higher effort and/or longer movement duration. In 50% of trials of each condition, the targets of different colour were far (18°; i.e., far horizontal distance), whereas in the other 50% of trials, the targets of different colour were close (6°; i.e., close horizontal distance). To eliminate visual differences in target detection, the distance of targets from the centre was constant across conditions, i.e., for close horizontal targets, vertical distance was far and vice versa. Importantly, in the far-target condition, path lengths for CoM$_P$ and CoM$_{P+I}$ were roughly equal. Conversely, in the close-target condition, path length was substantially shorter for CoM$_{P+I}$ (Fig. 3B). Hence, in the close-target condition, switching to the neighbouring target of different colour allowed participants to save motor costs, rendering intention pursuit relatively more costly than in the far-target condition. This should increase the frequency of changes of intention in the close- compared to the far-target condition. In order to enhance the differences in motor costs between target-distance conditions, the cursor speed was 1.8 times slower than in Exp. 1, increasing overall travel distance of movements. In addition, target onset was early (700–1000 ms before dot motion onset) in 80% of trials in Exp. 2 in order to reduce the likelihood of target selection errors.

Overall, behavioural performance in Exp. 2 was comparable to Exp. 1 (see Supplementary Note 1). In order to investigate the effect of target distance on the frequency of CoM$_P$ vs. CoM$_{P+I}$, a mixed-effects logistic regression with target distance as a fixed effect (far/close, dummy-coded with far distance as reference level) was conducted for test trials. It revealed a significant effect of target distance ($\chi^2(1) = 15.47$, $p < 0.001$), with CoM$_{P+I}$ occurring more often in the close- (M = 2.59%, SD = 0.44%) than far-target condition (M = 1.48%, SD = 0.68%; $b = 0.76$, 95% CI [0.38–1.16], OR = 2.15, Fig. 3C). Interestingly, target distance did not have a significant effect in a model with no-CoM vs. CoM as outcome variable ($b = 0.06$, 95% CI [−0.08 to 0.20], OR = 1.06, $\chi^2(1) = 0.70$, $p = 0.404$). Hence, target distance did not affect whether or not participants changed their mind about the dot-motion direction, but affected whether or not participants pursued their initial colour choice when a perceptual CoM occurred. That is, cost of goal pursuit was relevant to decisions about goals, but was not relevant to decisions driven by current perceptual input.

Finally, we checked whether the effect of target distance on CoM$_{P+I}$ depended on target onset time, and hence, the degree to which action cost associated with a potential future change could be anticipated prior to action onset. Including target onset (early/late) in the model revealed no significant main effect of target onset on CoM$_P$ vs. CoM$_{P+I}$ ($b = -0.05$, 95% CI [−0.82 to 0.67], OR = 0.95, $\chi^2(1) = 0.02$, $p = 0.902$). There was a trend for an interaction between target distance and target-onset time, due to

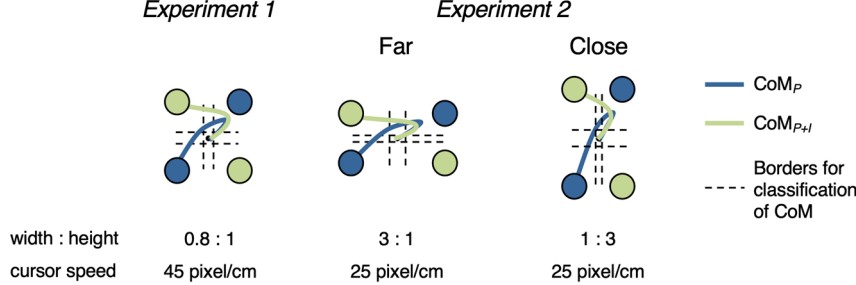

**A) Target distance**

width : height — Experiment 1: 0.8 : 1 — Far: 3 : 1 — Close: 1 : 3

cursor speed — 45 pixel/cm — 25 pixel/cm — 25 pixel/cm

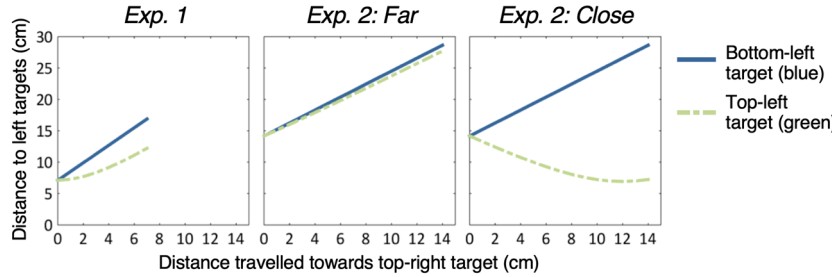

**B) Distance from other targets as a function of travelled distance**

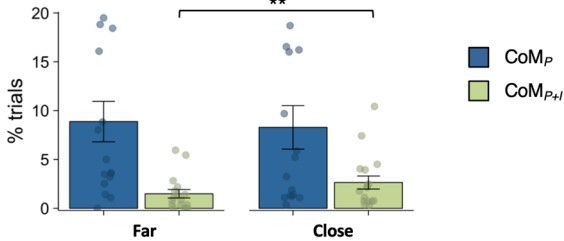

**C) Effect of target distance on % CoM$_P$ vs. % CoM$_{P+I}$ (Exp. 2)**

**Fig. 3 Experiment 2: manipulation of horizontal target distance. A** Target locations in Exp. 1 and 2. **B** Motor costs for each type of Change of Mind (CoM) as measured by the distance from the diagonal vs. horizontal target as a function of travelled distance (assuming straight movement trajectories towards targets). In the far-target condition, costs associated with each target were roughly equal, whereas in the close-target condition, the target of different colour was closer, hence rendering intention pursuit relatively more costly. **C** Effect of target distance on frequency of CoM$_P$ (blue) and CoM$_{P+I}$ (green) in Exp. 2 ($n = 16$ participants). Data are presented as mean values ± 1 SEM (**$p < 0.001$). Dots represent data points from individual participants. (likelihood ratio test for logistic mixed-effects models: **$p < 0.001$; no correction for multiple comparisons).

the effect of target distance being somewhat larger when target onset was late than when it was early. However, this effect was not significant ($b = 0.78$, 95% CI [−0.14 to 1.74], OR = 2.19, $\chi^2(1) = 2.75$, $p = 0.097$).

**Effect of changes of mind on sense of agency (Exp. 1 and 2).** In both experiments, participants were occasionally asked to judge how much control they experienced over the colour of the dots presented at the end of the trial. Participants provided Sense of Agency (SoA) judgements on a visual analogue scale ranging from 0 (no control) to 100 (a lot of control) after every trial with CoM and in 33% of no-CoM trials. We manipulated the percentage of dots that was coloured in the chosen target in order to increase variance in SoA judgements. Specifically, we predicted that SoA would be higher the more dots matched participants' initial colour intention[48–50]. More importantly, we assumed that in addition to the match between intended and obtained action outcome, SoA ratings would be modulated by whether or not a CoM occurred. In order to test this, the percentage of dots painted in the chosen colour was always 50% in trials with CoM, allowing us to investigate whether and how different types of

CoM affected SoA while keeping the action outcome (colour percentage) constant. For trials without CoM, outcome percentages were assigned randomly for each trial.

For analyses of SoA judgements, the data were collapsed across both experiments to increase power ($N = 33$). First, we checked whether SoA was sensitive to action outcomes. In test trials without CoM, SoA ratings increased linearly with the percentage of dots matching the colour of the hit target (linear contrast: $F(1, 32) = 164.91$, $p < 0.001$, $\eta_p^2 = 0.837$). When including experiment as a factor, no significant main effect of experiment, nor any interaction were observed (both $F < 1$). Hence, in both experiments, SoA ratings were sensitive to action outcomes showing that participants made appropriate use of the rating scale.

Next, we analysed the effect of CoM on SoA. Variability in trial numbers with CoM was high across participants [$n$CoM$_P$: M = 29.8, SD = 37.5, range: 1–159; $n$CoM$_{P+I}$: M = 7.3, SD = 9.0, range: 0–43] and 5/33 participants did not show any CoM$_{P+I}$. Therefore, linear mixed-effect models were used since they are recommended for analysing unbalanced and missing data[51]. Furthermore, they allowed us to include continuous predictors that varied on a trial-by-trial level, e.g., movement times. Participants were modelled as

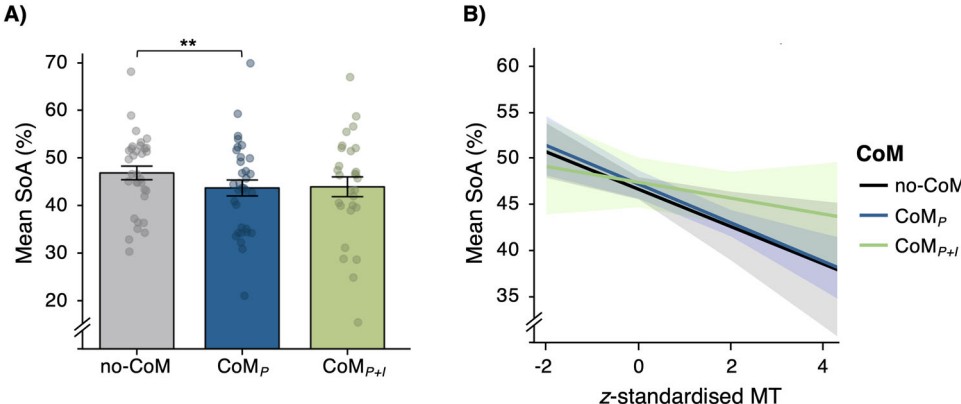

**Fig. 4 Effect of Changes of Mind (CoM) on Sense of Agency (SoA) ratings in Exp. 1 and 2 ($n = 33$ participants). A** Mean SoA ratings for each type of CoM (grey = no CoM; blue = perceptual CoM; green = perceptual + intentional CoM) in test trials. In both (**A**) and (**B**), data are presented as mean values ± 1 SEM. Dots represent data points from individual participants. Post hoc tests in a linear mixed-effects regression model revealed that SoA ratings were lower in $CoM_P$ than no-CoM trials (\*\*$p < 0.001$; Bonferroni-corrected $\alpha$-level of $0.05/3 = 0.017$). B) Predicted SoA ratings (marginal effects) for a mixed-effects model with CoM (no-CoM/$CoM_P$/$CoM_{P+I}$) and movement times (MT) as predictors, based on collapsed data from Exp. 1 and 2 (M ± 95% CI).

random intercepts. For no-CoM, only trials with 50% outcome were included. A model was specified that included CoM as a fixed effect (no-CoM/$CoM_P$/$CoM_{P+I}$; dummy coded with no-CoM trials serving as baseline) and SoA ratings as a continuous outcome variable. This model performed significantly better than a model without CoM as a predictor ($\chi(2) = 13.75$, $p = 0.001$). Post hoc pairwise comparisons with a Bonferroni-corrected $\alpha$-level of $0.05/3 = 0.017$ revealed that the effect of CoM on SoA ratings was driven by a significant decrease of SoA in $CoM_P$ (Fig. 4A; M = 43.8%, SD = 9.6%) compared to no-CoM (M = 47.1%, SD = 8.1%; $b = -3.02$, 95% CI [$-4.62$ to $-1.42$], $t(2169.9) = 3.70$, $p < 0.001$), whereas $CoM_{P+I}$ (M = 44.1%, SD = 11.0%) did not differ significantly from no-CoM trials ($b = -1.10$, 95% CI [$-3.49$ to 1.29], $t(2161.2) = 0.91$, $p = 0.366$). The difference between $CoM_P$ and $CoM_{P+I}$ was not significant ($b = 1.92$, 95% CI [$-0.48$ to 1.42], $t(2162.2) = 1.57$, $p = 0.118$). When adding experiment as a predictor, no main effect of experiment ($\chi(1) < 0.01$, $p = 0.924$), nor an interaction with CoM ($\chi(2) = 0.33$, $p = 0.847$) was found, suggesting that the effect of Changes of Mind on Sense of Agency was comparable across both experiments.

As Changes of Mind were classified based on movement trajectories, trials differed in terms of pure motor aspects. More specifically, movement times (MTs; i.e., time between response initiation and target hit) were shorter in no-CoM trials (M = 480.3 ms, SD = 246.8 ms) than in trials with $CoM_{P+I}$ (M = 975.8 ms, SD = 365.6 ms, $t(27) = 10.19$, $p < 0.001$, $d = 1.93$) and $CoM_P$ (M = 1089.5 ms, SD = 354.9 ms, $t(32) = 18.92$, $p < 0.001$, $d = 3.29$). To investigate if differences in MTs accounted for differences in Sense of Agency ratings, individuals' $z$-standardised MTs were included as a covariate in the model (Fig. 4B). This revealed a significant main effect of MTs ($\chi(1) = 24.32$, $p < 0.001$) driven by lower SoA ratings for longer MTs ($b = -1.81$, 95% CI [$-2.58$ to $-1.14$]). Furthermore, the effect of CoM on SoA disappeared ($\chi(2) = 1.51$, $p = 0.470$), and the decrease of SoA ratings in $CoM_P$ compared to no-CoM trials was not significant in the model including MTs ($t(2160.0) = 0.04$, $p = 0.970$). This suggests that the effect of $CoM_P$ on SoA was accounted for by differences in movement duration. Finally, there was no significant interaction between CoM and MTs ($\chi(2) = 2.22$, $p = 0.330$). In fact, longer MTs significantly reduced SoA judgements even when only no-CoM trials were considered ($b = -2.15$, 95% CI [$-3.58$ to $-0.72$], $\chi(1) = 8.65$, $p = 0.003$), suggesting that MTs

affected Sense of Agency judgements regardless of whether or not a CoM occurred.

**Summary and discussion Exp. 2**. In Exp. 2, the relative motor cost associated with intention pursuit was manipulated by varying target distances. When the distance to the alternative target colour was short compared to the initially-chosen colour, movement costs for perceptual + intentional CoM were low relative to a mere perceptual CoM where the initial intention was pursued. This caused an increased frequency of intention reversals compared to a condition where targets of both colours were roughly equally distant. Hence, motor costs influenced whether perceptual CoM caused a change in the movement required to realise an intention, or additionally, a change in the intention itself. This effect was present even when targets were presented late, suggesting that integration of motor costs occurred rapidly and dynamically as actions evolved. That is, even when participants could *not* anticipate action costs before dot-motion onset (late targets), motor costs affected decision making. Thus, action selection does not rely on full anticipation of motor costs[4], but instead, costs may be updated continuously as actions evolve[52–54]. Interestingly, in contrast to previous studies[4,8,37], we did not observe an overall increase in perceptual CoM in close compared to far targets. Hence, in our study, motor costs did not affect whether or not participants changed an ongoing action. However, motor costs did influence which aspects of action selection were changed (higher-order goals vs. lower-level sensorimotor decisions). It is possible that in the current study, participants were willing to correct their perceptual choices regardless of motor costs, given that they obtained additional monetary rewards for correct perceptual choices. By contrast, voluntary decisions were not associated with monetary incentives, and hence, differences in motor costs may have had a stronger impact on intention reversals than perceptual CoM per se.

Finally, across both experiments, reduced SoA was observed after perceptual CoM. However, this effect was statistically accounted for by differences in MTs between trials with and without movement updates. Participants may have used MTs as a proxy of (in)efficient motor performance or difficulty of action selection, which reduces SoA[27,28]. SoA was not modulated when participants changed their initial action goal. These findings are broadly in line with reconstructive accounts of conscious

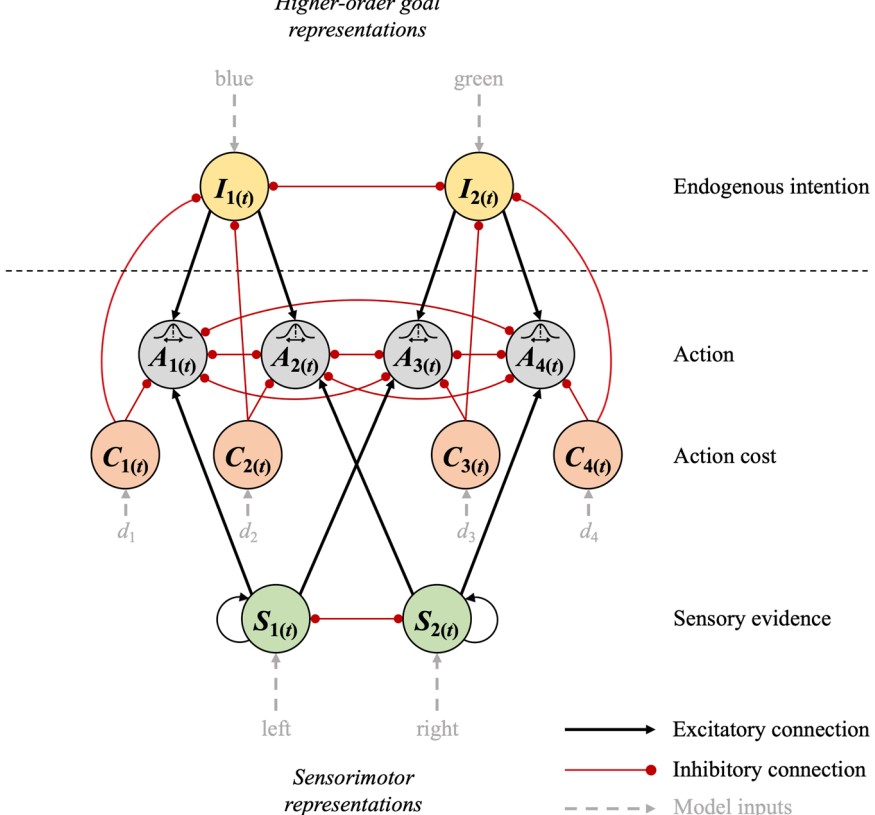

**Fig. 5 Hierarchical Attractor Network Model of Changes of Mind in voluntary action.** The network consists of 12 neural nodes that encode different pieces of information. Nodes are connected through excitatory (black) or inhibitory (red) connections. The action nodes $A_1$–$A_4$ compete against each other to determine which one of the four choice targets is selected. This competition takes into account information about (**1**) endogenous intentions (blue/green represented by nodes $I_1$ and $I_2$), (**2**) sensory information (left/right encoded by sensory nodes $S_1$ and $S_2$) and action costs ($C_1$–$C_4$) that depend on the distance $d$ to each target location. Intention nodes are represented on a hierarchically higher level than sensorimotor nodes, allowing for top-down regulation of the degree of variability in firing rates of the action nodes. All firing rates are updated continuously and can change dynamically. Hence, CoM can occur when one action node crosses the threshold for movement execution first, but later on, another action wins the competition. Different types of CoM can be dissociated based on which action the network switches to when a decision reversal occurs (e.g., perceptual CoM: switch from $A_1$ to $A_2$; perceptual + intentional CoM: switch from $A_1$ to $A_4$).

intention, which state that SoA is independent of the actual initial intention and instead relies on retrospective inference[32–34]. Indeed, as action goals are updated, predictions about action outcomes may be rapidly adjusted during action[55] without any consequences for subsequent inferences informing SoA. However, the absence of any significant effect of intention reversals on SoA in our study should be interpreted with caution since it is a null result based on low trial numbers. In particular, we cannot rule out that strong and sustained intentions may contribute to SoA.

**Hierarchical Attractor Network Model of CoM in voluntary action.** To model the detailed neurocognitive mechanisms underlying CoM in voluntary action, we explored a computational approach that could account for the dynamic integration of endogenous intentions with sensory evidence and motor costs. We propose an Attractor Network Model that consists of several nodes. Each node represents a population of neurons encoding different modalities of decision-relevant information (Fig. 5): (1) neural populations encoding the endogenous colour intention (e.g., blue vs. green; $I_1$ and $I_2$), (2) neurons that selectively respond to sensory information about left/right dot-motion ($S_1$ and $S_2$), and (3) neurons that calculate the movement cost according to the distance to each of the four target locations ($C_1$–$C_4$). Information from these neural populations is combined

by action nodes ($A_1$–$A_4$) that integrate all sources of information and specify the motor output, i.e., initiation of a movement towards the chosen target location. For example, action $A_1$ is selected for execution if the intention is blue ($I_1$ fires at a high rate), if the dots move left ($S_1$ fires at a high rate), and if the cost of moving to the left-blue target is relatively low ($C_1$ fires at a low rate). In other words, the firing rate of each action node reflects the strength of evidence in favour of a given action based on the combined information encoded in a distributed network.

The network is further characterised by a hierarchical structure that is based on hierarchical theories of intentional action[15,38,39] and decision-making[56]. Specifically, colour intentions are represented on a hierarchically higher level than sensorimotor information. That is, colour intentions reflect abstract, distal action goals with respect to the action outcome (coloured dots). By contrast, sensorimotor information about perceptual inputs (dot-motion stimulus) and movement-related costs inform *how* that goal can be achieved. Hence, hierarchy in the current model corresponds to the distinction between *what* goal to pursue vs. *how* to pursue it, and hence, the distinction between distal vs. motor intentions[15]. Hierarchy is implemented as top-down noise regulation in action selection through the intention nodes $I_1$ and $I_2$. Specifically, stronger intentions cause a decrease in noise, and thus, decreased variability in firing rates of the action nodes $A_1$–$A_4$. This is in line with previous studies showing that voluntary

intentions are associated with noise reduction in motor-related neural activity[42]. In addition, the implementation of hierarchy through noise regulation was inspired by Hierarchical Gaussian Filters, where the degree of noise (or volatility) of a hierarchically-lower variable can change over time, depending on the current state of a hierarchically higher variable[40,41]. Although Hierarchical Gaussian Filters were initially developed within the context of more abstract Bayesian inference models, a recent study showed that volatility estimates in these models were directly linked to neural activity in prefrontal cortex, which in turn predicted choice volatility (i.e., switching rate)[57]. Thus, we propose that neural noise regulation may be a valid candidate mechanism of top-down control that can be readily applied to biologically plausible Attractor Network Models. On a neuronal level, such changes in neural/behavioural volatility may be mediated by the dopaminergic system. Specifically, the dual-state theory proposes that the balance between $D1$ and $D2$ receptor activation affects signal-to-noise ratio of neural activity, and hence, may be crucial for the balance between stability and flexibility of actions[58].

Finally, at each level of the hierarchy, competition between neighbouring network nodes is implemented through lateral inhibition (e.g., $S_1$ vs. $S_2$, $I_1$ vs. $I_2$, etc.), resulting in a winner-take-all mechanism that determines the final behavioural outcome. Connections across the two hierarchical levels allow for integration of information. Specifically, action representations receive input from higher-order intentions and lower-level sensory evidence as well as information about motor costs. Hence, in the current model, decisions are made through a distributed consensus across different hierarchically organised neural populations[56].

Once one of the action nodes reaches a fixed firing rate threshold of $\theta = 40$ Hz, a movement towards the corresponding target location is initiated with a motor delay of 180 ms. Crucially, firing rates continue to be updated for 380 ms after initial threshold crossing due to a non-decision time consisting of sensory delays of 200 ms and motor delays of 180 ms (see ref. [2]). This allows for CoM after action initiation. In the model, a trial is considered to be a CoM if one action node crosses the firing rate threshold first, but later on a different action node crosses the threshold (and also surpassed all other nodes by 10 Hz to ensure a clear winning action). For example, the model may switch from action $A_1$ to $A_2$, reflecting perceptual CoM ($CoM_P$), i.e., a switch between actions that correspond to different sensory states ($S_1 \rightarrow S_2$) but the same colour intention ($I_1 \rightarrow I_1$). Alternatively, the network might switch from $A_1$ to $A_4$, reflecting perceptual + intentional CoM ($CoM_{P+I}$), and hence, a change in both the sensory state ($S_1 \rightarrow S_2$) as well as the colour intention ($I_1 \rightarrow I_2$). Finally, the model may switch between actions associated with different intentions but the same sensory state (e.g., $A_1 \rightarrow A_3$). Note that these (vertical) movement switches were considered to be a type of lower-level motor CoM in the behavioural task, which may reflect target selection errors where participants erroneously initiated a movement towards a target that did not correspond to their actual initial colour intention (e.g., due to difficulty in target detection). This assumption can be tested in the current model. That is, by defining the true colour intention on a given trial, we can analyse to what extent initial colour errors account for $CoM_{P+I}$ and vertical movement corrections, respectively. Finally, note that SoA results were not included in our computational model, given that our model focuses on the cognitive processes driving CoM about endogenous/exogenous decisions. Yet, our experimental results indicated that SoA was not directly related to these decision-making processes, but instead, largely depended on % action outcomes and movement times regardless of CoM.

**Model implementation and fitting**. Details of model implementation and fitting are provided in the "Methods" section. Briefly, firing rates of each neural population were modelled over time using a simplified version of a mean-field approach[59,60]. Updates in firing rates depended on (1) how strongly a given node was stimulated (based on external inputs and excitatory/inhibitory inputs from other nodes), (2) the node's firing rate on the previous time step, and (3) neural noise, which was initially set to 2 Hz in all nodes, but in action nodes, changed over time according to hierarchical noise control through voluntary intentions. The model was optimised to behavioural results from test trials of Exp. 1. The resulting model was then tested on far/close targets to check whether it could correctly reproduce the effect of target distance on intention reversals that we observed in test trials of Exp. 2. Model fitting was performed using a Covariance Matrix Adaptation Evolution Strategy (CMA-ES) algorithm.

**CoM in Attractor Network Model**. Simulations confirmed that the average outcomes produced by the fitted model closely matched participants' overall performance in Exp. 1 (see Supplementary Table 2). In addition, we showed that the full model proposed above performed better than alternative models with fewer parameters—e.g., models without cost nodes or without hierarchical noise control (see "Methods"). Most importantly, the model produced CoM on some trials. Figure 6A shows trial-averaged neural firing rates of action nodes $A_1$–$A_4$ for different trial types, time-locked to the onset of the first choice (i.e., first time the action threshold was crossed). Note that since CoM can occur at different points in time with respect to the first choice, averaging cancels out some of the fine-grained details of the second threshold crossing associated with CoM. For example, for $CoM_{P+I}$, action $A_3$ seems to stay below the action threshold on average. However, crucially, on a single trial classified as $CoM_{P+I}$, this action node—by definition—always crosses the threshold. This is further illustrated in Fig. 6B, which shows single-trial examples of action node firing rates and the resulting movement trajectories (see Supplementary Figs. 3–6 for additional examples). Both the single-trial and averaged firing rates show evidence of competition between action nodes when a CoM occurs. This is particularly pronounced between $A_1$ and $A_3$, thus illustrating the competition between whether to pursue the original colour intention, or instead, switch to the alternative colour.

Average frequencies of $CoM_P$ (M = 6.33%) and $CoM_{P+I}$ (M = 1.41%) in the model were highly comparable to CoM observed in Exp. 1 ($CoM_P$: M = 5.93%, $CoM_{P+I}$: M = 1.71%). The model also produced vertical movement corrections on 3.6% of trials (3.24% in Exp. 1). Simulations confirmed that 77.6% of these vertical switches corrected initial colour errors. This is in line with our interpretation of these vertical switches as target selection errors (motor CoM) in which the initial response was erroneously directed towards the wrong colour target, but was later corrected according to the actual underlying colour intention. By contrast, only 28.4% of $CoM_{P+I}$ were associated with an initial colour error. Instead, most $CoM_{P+I}$ produced by the model (56.6%) were associated with a correct initial colour choice, followed by a correction of a *perceptual* error that additionally involved a switch to the alternative colour. This suggests that, as opposed to vertical CoM, $CoM_{P+I}$ reflected a true change with respect to the initial colour intention, which was presumably driven by motor costs associated with intention pursuit.

Next, we tested whether the model could reproduce our behavioural findings showing that the frequency of changes in colour intentions depend on intentional strength and motor costs associated with intention pursuit. Indeed, the model was able to capture the effect of intentional strength on intention pursuit

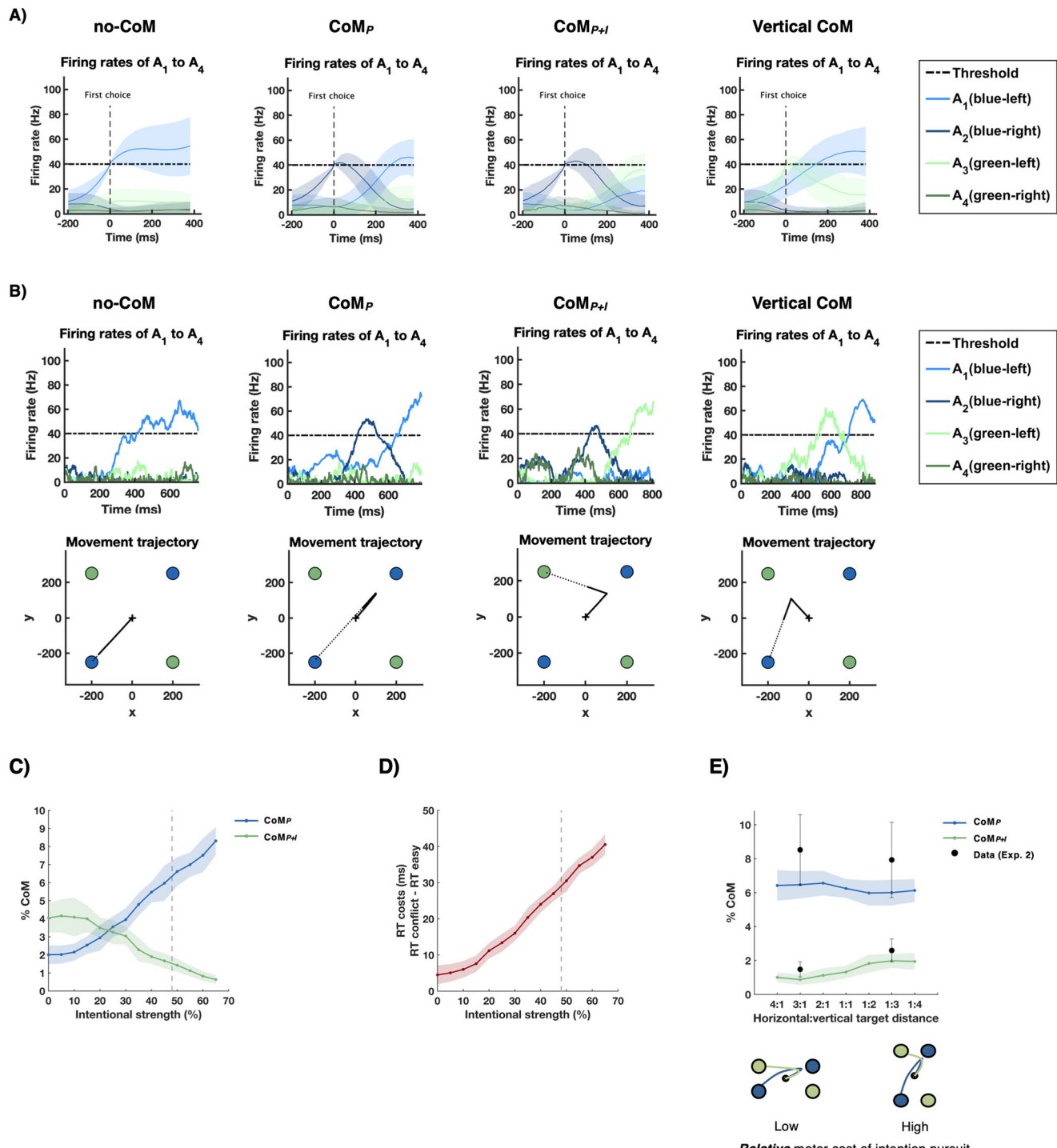

**Fig. 6 Changes of Mind (CoM) in attractor network model. A** Average firing rates of action nodes $A_1$ to $A_4$ (M ± 1 SD; $n = 30{,}000$ simulated trials) in trials without CoM (no-CoM), perceptual CoM (CoM$_P$), perceptual + intentional CoM (CoM$_{P+I}$), and vertical CoM. Firing rates were locked to time of first threshold crossing (first choice). For illustration purposes, only trials where the final choice was left-blue were included, except for CoM$_{P+I}$ where a change with respect to the colour intention resulted in a final left-green choice. **B** Single-trial simulations showing firing rates of action nodes (upper row) and the resulting movement trajectories (bottom row) for no-CoM, CoM$_P$, CoM$_{P+I}$ and vertical CoM. Dotted trajectories indicate completion of movements after the non-decision time of 380 ms (i.e., after the time period during which CoM can occur). **C** Effect of intentional strength on CoM$_P$ (blue) and CoM$_{P+I}$ (green). Stronger intentions result in lower frequency of intention reversals. **D** Stronger intentions increase RT costs in conflict trials (relative to RTs in easy trials). **E** Effect of target distance on CoM$_P$ (blue) and CoM$_{P+I}$ (green). Higher motor cost of intention pursuit increases frequency of intention reversals. Grey dashed lines in **C** and **D** indicate strength of intention in the model that was optimised to the results obtained in Exp. 1. In **C**–**E**, data are presented as the mean ± 1 SD from $n = 30$ model simulations with 1000 trials each. Black data points in **E** represent behavioural results from Exp. 2 ($n = 16$; M ± 1 SEM).

observed in Exp. 1. That is, when increasing the strength of colour intentions while keeping all other model parameters fixed, a decrease in the frequency changes of intention was observed, while the frequency of perceptual CoM without intention reversal increased (Fig. 6C). At the same time, stronger colour intentions also caused the model to produce larger RT costs in conflict trials relative to easy trials (Fig. 6D; see Supplementary Note 2 for more details). Thus, in line with our interpretation of the results from Exp. 1, inter-individual differences in the strength of colour intentions may have been the causal factor driving the observed correlation between the frequency of intention reversals in test trials and RT costs in conflict trials. Finally, the model fitted to data from Exp. 1 was also able to replicate the effect of target distance observed in Exp. 2. Specifically, when changing horizontal target distance (and hence, the relative cost associated with each action), the model correctly predicted an increase of changes of intention ($CoM_{P+I}$) in close compared to far horizontal targets. Although this increase was numerically small, it was robust and highly similar to the increase observed in the actual experiment (Fig. 6E). The model further predicted a slight decrease in $CoM_P$ for far compared to close targets, which was descriptively present in Exp. 2. Importantly, we further showed that the effects of strength of intention and motor costs on $CoM_{P+I}$ in the model were not mediated by potential changes in the rate of initial colour errors. That is, model predictions remained the same when we excluded trials where the model initially chose a target of the wrong colour (see Supplementary Figs. 7 and 8).

In summary, the current Attractor Network Model provides a biologically plausible neuro-computational mechanism through which dynamically changing information from different endogenous and exogenous sources is integrated by a network of neural populations that guide actions in a continuous manner, allowing for rapid CoM with respect to perceptual and/or intentional aspects of voluntary actions as they unfold. Our modelling results confirmed that the majority of CoM about colour intentions did not reflect corrections of initial colour errors, but instead, reflected a change in the colour intention that occurred after a correct colour target was initially selected. The frequency of such changes of intention in the model in turn depended on the strength of intentions and their associated motor costs, thus replicating the pattern of results observed in the behavioural experiments.

## Discussion

Previous studies of CoM have largely focused on updating of stimulus-driven actions based on new external evidence[1–10] (but see ref. [13] for a recent study on endogenous CoM). By contrast, the current study investigated changes of voluntary action decisions. Based on previous research on perceptual CoM, we developed a task where perceptual updates occasionally resulted in an additional change in a higher-order goal intention. In Exp. 1, we showed that the frequency of changes of intention was inversely related to the strength of participants' initial intentions. In Exp. 2, we found that higher motor costs induced more switches to a target that did not match the initial endogenous intention. Note that our study focused on intention reversals that were triggered by a CoM about external, perceptual information, which subsequently caused a re-evaluation of the initial intention based on its trade-off with motor costs. Even though we observed some changes between colour targets in the absence of perceptual CoM (vertical movement corrections), these changes appeared to largely reflect lower-level motor CoM that were caused by initial colour errors due to target uncertainty, instead of reflecting genuine changes in higher-order colour intentions. An intriguing

possibility for future studies may be to capture intentional CoM that are completely independent of any external changes, in order to further elucidate the processes through which endogenous decisions are updated continuously.

Finally, we propose that the cognitive mechanisms underlying the flexible nature of voluntary actions can be captured through dynamics in a Hierarchical Attractor Network Model that continuously integrates multiple sources of endogenous and exogenous information. Past computational accounts of CoM largely neglected such integrative processes, and instead, focused on decisions that are purely driven by a single source of (perceptual) evidence (e.g., refs. [1,2]). The current model provides an extension of this work by introducing a unified framework for different types of CoM in voluntary actions, which are guided by several pieces of not just weighted decision variables, but of hierarchically organised endogenous and exogenous information. In addition, in contrast to previous models of CoM, the current model explicitly allowed for an active role of action representations during the evolving decision-making process. That is, action nodes in the current model were not simply a mere output system of higher-order decision-making areas, but instead, played an essential role in determining the final action outcome by integrating various sources of decision-relevant information to guide action selection in a gradual and continuous manner. Thus, instead of a serial, feedforward hierarchy from decision to action, our model proposes an interactive hierarchy, in which information related to action can itself feed back to modify the decision. This is in line with recent theories that view decisions and movements as highly integrated processes, which evolve continuously and gradually over time, instead of representing strictly serial and segregated processes[56,61–63] (for a recent review, see ref. [64]). Moreover, in line with previous studies, our model assumes that multiple action representations evolve in parallel and that action selection is determined through a winner-take-all competition between these multiple co-existing affordances[61,65–67]. Finally, motor outcomes of the model (i.e., simulated movement trajectories) affected subsequent decision updates by causing dynamic changes in motor costs (i.e., changes in distance to each target). Thus, our model provides a common framework for decision making and action selection, and accounts for their reciprocal relation, instead of assuming strictly separate and serial processing of decisions and actions. In this context, the model makes further predictions that can be directly tested in future studies. For example, it is plausible that firing rates of action nodes (i.e., the strength of decision evidence in favour of an action) are directly linked to more fine-grained, gradually varying details of motor policies, such as movement speed or vigour, rather than mere categorical choices between action alternatives.

Finally, the current model implemented a hierarchical organisation. While previous work has largely focused on higher-order mechanisms related to meta-cognitive processes, e.g., uncertainty about sensory information[5,6], our model introduces top-down control through endogenous intentions that are independent from the sensory information itself. That is, we propose that abstract action goals that are generated internally can affect processing of external information on a lower, sensorimotor level. In line with this, previous studies have shown that within the frontal cortex, more anterior regions representing abstract information (e.g., goals) exert top-down control over more posterior regions involved in lower-level sensorimotor control[43,44]. Moreover, our current model proposes that noise control plays a crucial role in such top-down control. It has previously been suggested that higher-order areas exert top-down control by gating inputs/outputs of lower-level areas[44]. The noise reduction mechanisms implemented in the current model may be

fundamental to such gating of information: That is, noise reduction can enhance action-relevant information, thus allowing voluntary actions to be shielded from noisy sensory distractions[68].

One limitation of the current model is that CoM is largely driven by neural noise within the network. Noise in the input sources (e.g., sensory evidence) is disregarded. Yet, previous studies have shown that, in addition to the (constant) coherence level of a trial, moment-by-moment fluctuations of sensory evidence affect decision-making and can drive CoM[1,2]. Our model does not account for such within-trial fluctuations in the sensory stimulus, given that it was optimised to fit participants' average behaviour across trials, and thus, momentary fluctuations in evidence will average out. Similarly, we assumed that inputs into intention nodes were constant. However, it is possible that intentional strength fluctuates both within and across trials. Finally, cost inputs may be noisy given that they rely on (noisy) estimates of one's current distance from each alternative target. Future studies might extend the current model by accounting for variability in each model input both within and across trials, and thus, consider alternative sources of noise in addition to neural noise.

We propose that by studying when, why and how voluntary intentions are maintained vs. changed can provide important insights into the functional role and nature of intentions. Voluntary intentions have previously been conceptualised either as strong determining tendencies[69], or instead, as weak and labile plans[17,47,70]. Rigorous experimental methods to quantify the strength of any given intention have been lacking. Our results suggest that intentions vary gradually in strength, are evaluated continuously, and can be reversed even when an action has already been initiated. Our methods further show that these various features of intention rigidity/flexibility can be quantified and compared within and between individuals. Reversibility of intentions can be highly advantageous in that it allows people to flexibly adjust their behaviour to the current context. On the other hand, an important concept of the voluntary control of behaviour is the need for intention pursuit over long periods of time—e.g., when intending to quit smoking or lose weight. People may give up on these intentions because of new stimuli that can trigger decision reversals. For example, addiction relapse is often caused by exposure to drug-related external stimuli, in particular in individuals with high sensitivity to incentive cues[71]. More generally, disturbances in the balance between goal-shielding vs. goal-switching may be linked to a large range of psychiatric and neurological conditions, and hence, understanding the processes underlying this balance is crucial to well-being and mental health[19].

In conclusion, voluntary actions are shaped by continuous decision-making processes that integrate external information with endogenous intentions. The flexible nature of action selection allows agents to dynamically decide which intention to pursue and how to pursue it. Our study introduces a quantitative laboratory approach and computational model that can capture the neurocognitive processes underlying flexible goal-directed actions. This provides important insights into the nature of voluntary intentions, and the mechanisms underlying goal pursuit and its disturbances, with important social and personal implications.

## Methods

**Participants**. The study was approved by the UCL Research Ethics Committee. Participants provided written informed consent prior to the study. For Exp. 1, 21 right-handed participants were recruited through the ICN subject database. One participant did not reach the performance criterion in the training session (see below) and another participant withdrew after training. Two further participants were excluded, one due to technical issues during data collection and one due to strategic decision delay in the task (see below). The final sample consisted of 17 participants (13 female, age: M = 22.6 yr, SD = 3.1). For Exp. 2, 21 right-handed participants were initially invited for the experiment. Three participants did not reach the performance criterion during training (see below) and two further participants were excluded due to poor performance in the test session (>15% errors or misses in easy trials), resulting in a final sample of 16 participants (11 female, age: M = 23.2 yr, SD = 2.9). Participants received £7.50/h and a performance-based reward.

**Apparatus and stimuli**. The experiment was programmed in Matlab R2014a and the Psychophysics Toolbox[72]. Motion stimuli were generated using the Variable Coherence Random-Dot Motion code (https://shadlenlab.columbia.edu/resources/VCRDM.html). The stimuli were presented in a central aperture (4.5° diameter) with a stimulus density of 15.6 dots deg$^{-2}$ s$^{-1}$, at a screen refresh rate of 60 Hz. The percentage of dots that were displaced in the same direction determined the motion coherence and motion direction (left/right) was assigned randomly for each trial. In Exp. 1, target circles of 1.8° diameter were located at a distance of 9.6° from the centre of the screen (x = 6.0°, y = 7.5°). In Exp. 2, target distance varied on a trial-by-trial basis (far targets: x = 18°, y = 6°; close targets: x = 6°, y = 18°). Target colours were random pairs of blue, green, pink, and orange of comparable luminance. Participants were seated approximately 60 cm from a computer screen and moved a cursor to the targets using a Wacom Intuos Pro pen tablet. Movement trajectories were recorded at a sampling frequency of 125 Hz.

**Trial procedure**. Participants made endogenous choices between random pairs of 4 target colours (blue/green/pink/orange). Once they had chosen a colour, participants clicked on a central fixation cross and after a random delay of 700–1000 ms, the motion stimulus and 4 targets, 2 of each colour, appeared. In 50% of trials (80% in Exp. 2), targets were presented immediately after colour choice (early targets), whereas in the remaining trials, they appeared 700–1000 ms after colour choice, i.e., at the same time as the dot-motion stimulus (late targets). 500 ms after participants reached a target, 25/50/75/100% of the dots from the last 3 video frames were presented in the colour of the hit target (1 s). On 1/3 of trials, and after every CoM, participants were then asked to provide Sense of Agency (SoA) judgements on a visual analogue scale ranging from "none" to "a lot". On 1/5 of the remaining trials (~13% of trials overall), participants were asked to provide an estimate of the percentage of dots that matched their initial colour intention. Note that outcome judgements were included to motivate participants to pay attention to the action outcomes, and hence, render colour choices more meaningful within the context of the task. However, given that outcome judgements never appeared after CoM (which was always followed by SoA ratings), we did not further analyse them.

**Training session**. Participants had to pass a training session the day before the actual experiment. They were trained on the original two-choice motion discrimination task until they reached 70% accuracy in trials with 35% motion coherence. One participant failed to reach the criterion and was not invited for the experimental session. All other participants performed 160 additional trials with randomly varying motion strength (5–65% coherence) in order to obtain stable performance. Finally, an alternating staircase procedure was administered (see ref. 4 for details) to determine the motion coherence at which a participant's accuracy was ~60% (coherence: M = 11.8% SD = 4.1%). This level was chosen to maximise the frequency of perceptual CoM[1]. During training, trial-by-trial error feedback (red dots) was provided.

**Experimental session**. After a short practice block, participants were given 1 h to complete as many trials as possible (M = 358.2, SD = 37.5) in Exp. 1. In Exp. 2, participants completed two identical experimental sessions in which they were given 1.25 h each to complete as many trials as possible (M = 815.6, SD = 57.2). The duration of Exp. 2 was increased compared to Exp. 1 in order to obtain a sufficient number of CoM$_P$ and CoM$_{P+I}$ for each target distance condition. To motivate participants to be fast and accurate, they won 1 p for every correct perceptual choice. After each block of 30 trials, participants received feedback about their perceptual choice accuracy. There was no trial-by-trial error feedback, but a "too early!" message was shown when participants initiated a response before stimulus onset. Furthermore, a "too slow!" message was shown if response initiation exceeded a certain deadline or if the target was not reached within 3 s after response initiation. All trials with warning messages were repeated later on, at a randomly selected trial. In order to induce fast response initiation, the response deadline was initially 1000 ms, but decreased by 50 ms after every block if a participant had less than 10% trials with CoM and less than 15% misses. RTs were defined as the point in time at which the cursor left a central circle of 1.1° diameter, at which point the motion stimulus disappeared. Previous studies showed that, due to sensorimotor delays, CoM occurs even when the external stimulus is removed at action onset[1].

In both the training and test session, participants were instructed to fixate the central cross throughout each trial. Electrooculography was used to monitor eye movements and, whenever necessary, participants were reminded to keep fixation.

**Classification of CoM**. Trials with CoM were classified online based on movement trajectories: If the cursor position exceeded 10% of both the $x$- and $y$-distance towards a given target, but then ended in the diagonally opposite target (of the same colour), the trial was classified as $\text{CoM}_P$. If it ended in the horizontally neighbouring target (of different colour), it was classified as a $\text{CoM}_{P+I}$. In Exp. 2, the absolute coordinates that had to be exceeded differed between target distance conditions (Fig. 3A), due to the different target locations. Using relative rather than absolute coordinates in Exp. 2 ensured that CoM classification was not biased by differences in movement angles across target distance conditions. Importantly, CoM typically occurred much later than the applied classification criteria. On average, CoM occurred roughly halfway through the movement, i.e., when 44.99% (SD = 7.73%) of the total distance towards the initial target had been completed.

All main analyses of perceptual CoM ($\text{CoM}_P$ and $\text{CoM}_{P+I}$) focused on test and easy trials only. By contrast, movement corrections in conflict trials were interpreted in a slightly different manner due to differences in target arrangement and instructions (see "Results").

**Movement analysis**. Movement trajectories were analysed in Matlab R2014b. All trials that had been classified as CoM during the task were inspected individually. Trials with double CoM (i.e., trials in which participants changed their mind more than once in a single trial) were excluded from all analyses (Exp. 1: 0.93% of all trials; Exp. 2: 0.83%). In addition, trials in which initial movement trajectories were not clearly directed towards one of the targets (e.g., circular trajectories or vertical movement initiation; Exp. 1: 0.13%; Exp. 2: 0.47%) were excluded. Furthermore, velocity profiles of reaching movements were analysed. Note that participants might have initiated a response in any direction in order to comply with the short response deadline, subsequently choosing a target only after having left the home position. In that case, curvature away from the initial trajectory would not be a CoM, as the initial trajectory would not reflect commitment to a specific target. Completely excluding any element of strategic delay for individual trials is difficult. However, frequent stopping shortly after movement initiation (i.e., velocity = 0 at some point after movement initiation) even in trials with straight trajectories would clearly indicate strategic decision delay. In Exp. 1, one participant stopped in 28.6% of straight trajectories, with an average stopping duration of 351.2 ms and was therefore excluded from all analyses. Such stopping was rare in all other participants (stop frequency: M = 7.4%, SD = 5.2%; stop duration: M = 157.9 ms, SD = 40.8 ms). Note that this percentage of trials with stopping is highly comparable to the percentage of trials with CoM, and hence, can be attributed to decision uncertainty and vacillation, rather than strategic decision delay. In Exp. 2, movement velocities indicated that none of the participants showed strategic decision delay.

**Statistical analyses**. Given the small percentage of trials with CoM, mixed-effects logistic regression models were used for analyses of CoM frequency[51]. Model fitting was performed using Maximum-likelihood estimation with the lme4 package[73] in R[74]. Binomial models with a logit link were specified. To investigate CoM, two types of binary outcome variables were analysed: Either no-CoM (0) vs. CoM (1) for analyses of overall frequency of perceptual CoM (regardless of type of CoM), or $\text{CoM}_P$ (0) vs. $\text{CoM}_{P+I}$ (1) for analyses of different types of CoM within CoM trials. Participants were modelled as random intercepts. Including random slopes did not change any of the results and only one of the models performed significantly better when random slopes were added. Hence, all models reported contain random intercepts only. Parameter estimates $b$ and 95% profile confidence intervals are reported in log-odds space, and odds ratios are reported to facilitate interpretation. Statistical inference was performed by comparing models with vs. without a given fixed effect using likelihood-ratio tests. Satterthwaite approximation for degrees of freedom was used[75]. All other analyses (comparison of means with ANOVAs/$t$ tests; two-tailed) were performed in IBM SPSS Statistics for Windows, version 21[76]. For RT analyses, only correct trials within ±3 SD of the individual's average RT in each condition were included. No differences in RTs were found between trials with vs. without CoM (see Supplementary Fig. 2). Hence, RT analyses included all correct trials regardless of whether or not CoM occurred.

**Statistics and reproducibility**. Each experiment was only conducted once and was not replicated independently. However, the overall behavioural results in Exp. 2 were consistent with the main findings from Exp. 1 (e.g., overall rate of CoM; see Supplementary Note 1), despite a slightly different task design.

**Attractor Network Model**. The model was implemented and fitted in Python 3.7. All model code is available on GitHub, including a Matlab implementation of the model that can be used to run simulations and plot single-trial model outcomes (https://github.com/AnneLoffler/AttractorNetwork-CoM).

**Network architecture**. The Attractor Network Model consists of 12 neural nodes that are grouped into different modules according to the source of information they represent (Fig. 5):

1. Two intention nodes ($I_1$, $I_2$) that encode the voluntary intention (blue/green)
2. Two sensory nodes ($S_1$, $S_2$) that selectively respond to dot-motion direction (left/right)
3. Four cost nodes ($C_1$–$C_4$) that calculate the cost associated with each action based on distance to each target location
4. Four action nodes ($A_1$–$A_4$) that correspond to the 4 possible action alternatives, and hence, location of the choice targets (left/right top/bottom)

Each node represents a population of neurons whose firing rates change dynamically over time. The firing rates of intention nodes ($I_1$, $I_2$) and sensory nodes ($S_1$, $S_2$) depend on model inputs whose intensities corresponded to the strength of intention and strength of sensory evidence (i.e., motion coherence), respectively. Firing rates of cost nodes ($C_1$–$C_4$) depend on the distance $d$ to each target location. Hence, intention, sensory and cost nodes are input nodes that receive direct model inputs. Action nodes do not receive any direct external inputs, but instead, integrate information from all other network nodes to determine the behavioural outcome (i.e., movement trajectory towards one of the four targets). Integration of information is achieved through neural connectivity. Colour intentions and sensory inputs have excitatory effects on action nodes, whereas costs have inhibitory effects. Furthermore, neurons that encode the same modality of information, e.g., sensory nodes $S_1$ and $S_2$, but respond selectively to a specific input (e.g., left vs. right dot motion), compete against each other through lateral inhibition. This ensures that over time, a single choice option is selected through a winner-take-all mechanism that supresses competing choice alternatives.

Due to considerations of parsimony, network connections were assumed to be symmetric. For example, $I_1$ and $I_2$ had equally strong connections onto their corresponding action nodes (and similar for $S_1$ and $S_2$, etc.). Additionally, inhibitory competition between action nodes associated with the same colour intention (i.e., $A_1$ vs. $A_2$ and $A_3$ vs. $A_4$) was assumed to be stronger than competition between action nodes associated with different colour intentions (e.g., $A_1$ vs. $A_3$). This was because actions associated with the same colour intention corresponded to diagonally opposite targets, respectively, and hence, movements in either direction were mutually exclusive (i.e., competition is stronger). Moreover, the effects of costs on intention nodes were assumed to be weaker than the effects of costs on action nodes. Note that this assumption was necessary since action costs would otherwise completely suppress intentions at trial start. Due to considerations of parsimony, we chose to fix the weight from costs to intention nodes to be 0.5 of the weight from costs to action nodes, instead of fitting two separate weights for each cost parameter. Finally, sensory nodes had self-excitatory connections, representing temporal integration of sensory evidence from the dot-motion stimulus[2,59].

**Modelling firing rates**. In order to compute the firing rates of each neural node over time, a simplified version of the mean-field approach introduced by Wong and Wang[59] was used (see ref. [60]). That is, instead of modelling individual spiking neurons, the overall firing rate of a given neural population (node) was calculated for each point in time. Firing rates of each node were updated in time steps of 1 ms and depended on (1) how strongly a given node was stimulated (based on external model inputs and excitatory/inhibitory inputs from other nodes), (2) the node's firing rate on the previous time step, and (3) neural noise. Hence, the following equations were used to determine the firing rate $r$ of a given node $i$ at time point $t$.

First, the total stimulation that each node received at time $t$ was calculated according to Eq. 1. Stimulation depended on direct external inputs into that node (if any), plus the sum of neural inputs from all other nodes (and itself in case of auto-connections). The neural input that node $i$ received from node $j$ depended on the firing rate of $j$ at the previous time step weighted by its connectivity to $i$ as defined by the weight matrix $W$

$$\text{stim}_{i,t} = \text{in}_{i,t} + \sum_{j=1}^{12} r_{j,t-1} W_{i,j} \tag{1}$$

Recurrent updates in firing rates were then computed as a function of the node's previous firing rate and the current stimulation $\text{stim}_{i,t}$ it received. While our mean-field approach did not rely on explicit modelling of synaptic mechanisms, we introduced a base time constant $\tau$ of 100 ms for all neural nodes, imitating the effect of slow NMDA receptors, which have been shown to be the primary contributor to slow temporal integration of evidence[59]. Using the Euler–Maruyama approximation for differential equations[60,77], the firing rate of node $i$ at time $t$ was calculated then as follows

$$r_{i,t} = r_{i,t-1} + (\text{stim}_{i,t} - r_{i,t-1})\tau^{-1} \tag{2}$$

Finally, random Gaussian noise $s$ was added to the firing rate of each node

$$r_{i,t} = r_{i,t} + s_{i,t} \text{ with } s_{i,t} \sim N(0, \sigma_{i,t}^2) \text{ and } \sigma_{i,t}^2 \geq 0 \tag{3}$$

The degree of neural noise varied according to $\sigma^2$, which was initially set to 2 Hz for all nodes. However, according to the assumption of top-down noise regulation through higher-order intentions, $\sigma^2$ of each action node $A_1$ to $A_4$ varied as a function of the state of intention nodes $I_1$ and $I_2$. Specifically, similarly to the original formulation of Hierarchical Gaussian Filters[40,41], increased firing rates in higher-order intentions caused a reduction of noise in the corresponding, lower-level action nodes. That is, higher firing of $I_1$ caused a reduction of noise in its associated action nodes $A_1$ and $A_2$ (Eq. 4a) and $I_2$ caused noise reduction in action

nodes $A_3$ and $A_4$ (Eq. 4b). Noise reduction in action nodes was directly proportional to firing rates of intention nodes, where $h$ is a factor that indicates the degree of hierarchical noise reduction. For example, if $h = 1$ and $I_1$ fires at 50% of its maximum firing rate, noise in $A_1$ and $A_2$ is reduced by 50%

$$\sigma^2_{A1,t/A2,t} = \sigma^2_0 - h \frac{r_{I1,t-1}}{100} \sigma^2_0 \qquad (4a)$$

$$\sigma^2_{A3,t/A4,t} = \sigma^2_0 - h \frac{r_{I2,t-1}}{100} \sigma^2_0 \qquad (4b)$$

Firing rates were restricted to a range of 0–100 Hz. Firing rates of all neural nodes were set to an initial starting value of 10 Hz at the beginning of each trial. Once one of the action nodes reached a fixed firing rate threshold of $\theta = 40$ Hz (and surpassed all other action nodes by at least 10 Hz to ensure a single winning action), a movement was initiated with a motor delay of 180 ms. Movement direction corresponded to the chosen target location (according to the winning action node) and movement speed was constant at 0.7 pixels/ms, resulting in a movement duration of ~450 ms for no-CoM trials, in line with movement times measured in Exp. 1. For simplicity, movements were simulated with straight trajectories towards the chosen target, without additional motor noise. While this does not result in realistic trajectory shapes observed in reaching tasks, it provided us with sufficient detail to approximate motor costs by calculating the Euclidean distance between the current cursor location and each of the 4 targets. As cursor position changes, motor costs associated with each target change according to the relative distance to each target at a given point in time (see Fig. 3B).

Movement execution towards a chosen target continued even if the action node dropped below the threshold, unless another action node reached the firing rate threshold, in which case the movement was redirected towards the new target choice. Firing rates continued to be updated for 380 ms after initial threshold crossing due to a non-decision time consisting of sensory delays of 200 ms and motor delays of 180 ms (see ref. [2]). This allowed for decision updates even after initial action onset [1,2]. In line with previous models (e.g., ref. [1,2]), firing rate updates were stopped after the non-decision time (i.e., no more CoM could occur since no more new evidence was obtained) and the movement was completed according to the final target choice.

**Model inputs.** External model inputs were simulated at a rate of $f_{in} = 60$ Hz. Inputs into sensory nodes were presented after the sensory delay of 200 ms. The respective strength of inputs into $S_1$ and $S_2$ corresponded to the strength of sensory evidence, i.e., dot-motion coherence, and was calculated as

$$\text{in}_{S1,S2} = f_{in}\left(1 \pm \frac{\text{coh}}{100}\right), \qquad (5)$$

with coh corresponding to the % coherence and $+/-$ indicating whether or not motion direction corresponded to the neurons' preferred motion direction. By analogy, inputs into intention nodes depended on the strength of the endogenous colour intention col, i.e., the relative endogenous evidence in favour of a given colour

$$\text{in}_{I1,I2} = f_{in}\left(1 \pm \frac{\text{col}}{100}\right) \qquad (6)$$

Equations 5 and 6 ensured that model inputs were normalised. Hence, the total input into the network was constant across different levels of sensory/endogenous strength. Similarly, the input into cost nodes was set to an equal value of 60 Hz at trial start. Once a movement was initiated, costs were updated relative to changes in Euclidean distance between the current position and each target location. Consequently, the total external model inputs (sensory, endogenous, and costs) were balanced, and thus, only the relative strength of evidence from each source affected action selection.

**Model fitting.** Eight model parameters were fitted, relating to the connectivity weights between neural nodes, the strength of decision evidence, and the degree of top-down noise modulation through intentions:

*Network connectivity weights*

1. Intentional nodes to action nodes (e.g., $w_{I1 \to A1}$)
2. Sensory nodes to action nodes (e.g., $w_{s1 \to A1}$)
3. Cost nodes to action nodes (e.g., $w_{C1 \to A1}$) and intention nodes (e.g., $w_{C1 \to I1}$)
4. Sensory auto-connections (e.g., $w_{S1 \to S1}$)
5. Lateral inhibition (e.g., $w_{I1 \to I2}$)
   *Decision evidence*
6. Strength of sensory evidence (dot motion coherence)
7. Intentional strength (colour intention)
   *Hierarchical control*
8. Degree of hierarchical noise control

To reduce the number of free parameters, the strength of connectivity weights was assumed to be symmetrical (e.g., $w_{I1 \to A1} = w_{I1 \to A2}$). A covariance matrix adaptation evolution strategy (CMA-ES) algorithm with maximum likelihood updates was used to fit model parameters[78,79]. CMA-ES is a randomised search algorithm suited to explore our parameter space about which initial knowledge is constrained. We first informally explored different sets of parameters to

approximate values that would yield reasonable behavioural outcomes (i.e., RTs within 500–1000 ms, perceptual error rates of ~40%, CoM of ~10% with more $\text{CoM}_P$ than $\text{CoM}_{P+I}$, and low rates of missed/early responses). We then performed a global optimisation test with single- and multi-objective CMA-ES, which validated that one of the convergence regions was on the intuitively preferable parameter values. Finally, this convergence region was used to initialise a set of local optimisation runs to fine-tune parameters. A multi-objective CMA-ES was applied[80,81] to minimise the error of model predictions from actual behavioural outcomes in Exp. 1. Specifically, the model was optimised according to the following behavioural outcomes: RTs, % perceptual choice accuracy, % $\text{CoM}_P$, % $\text{CoM}_{P+I}$, and % colour changes that occurred without a perceptual change (vertical switches). In addition, the model was fitted to minimise early responses (i.e., responses before stimulus onset), misses (RTs > 1 s), and maximise colour choice accuracy (e.g., the true intention was blue and an action node associated with blue was selected). Note that for colour accuracy, only initial choices were considered since final colour errors after a CoM may reflect $\text{CoM}_{P+I}$, and hence, a change in the colour intention rather than an error with respect to the initially chosen colour.

Each optimisation run consisted of 10000 trials. Powell's method was used to obtain local minima[82]. Supplementary Table 1 summarises the initial parameter values and step sizes that were used for the optimisation as well as the final, fine-tuned parameters obtained with the CMA-ES approach. Using the fitted model parameters, 30 simulations of 1000 trials each were then conducted to derive model predictions for a given experimental condition (e.g., simulation of effect of target distance). Model predictions strongly overlapped with participants' actual behaviour in Exp. 1 (Supplementary Table 2) and Exp. 2 (see "Results" section and Supplementary Fig. 8).

Finally, we checked whether the network maintained a stable steady state in the absence of external inputs. We ran additional simulations where all external model inputs were set to 0. These simulations confirmed that the network maintained a stable steady state with below-threshold background activity (average firing rates of action nodes $A_1$–$A_4$: M = 7.73 Hz).

**Model comparisons.** The model obtained through the CMA-ES procedure yielded a good fit to the behaviour observed in test trials of Exp. 1. To check whether this model performed better than simpler versions of the network, we compared the full model to two alternative models with fewer parameters: (1) a model without hierarchical noise control (i.e., when $h$ was fixed to 0 and (2) a model without action cost parameter (i.e., when $w_{C \to A}$ was fixed to 0). The Akaike Information Criterion (AIC) was used to evaluate model fits, and the relative likelihood of the respective models was then obtained as follows

$$L = \exp\left(\frac{\text{AIC}_{\text{full model}} - \text{AIC}_{\text{alt model}}}{2}\right) \qquad (7)$$

Model comparisons revealed that the model without hierarchical noise control was only 0.139 times as probable (AIC = 35.58) as the full model with 8 parameters (AIC = 31.64). Furthermore, the model without an action costs was only 0.0005 times as likely (AIC = 47.05) as the full model. Thus, the attractor network model we proposed here (Fig. 5) was the model that best accounted for the behavioural results observed in Exp. 1.

**Reporting summary.** Further information on research design is available in the Nature Research Reporting Summary linked to this article.

## Data availability
The datasets generated during and/or analysed during the current study are publicly available on the Open Science Framework, https://osf.io/wbex8/ (https://doi.org/10.17605/OSF.IO/WBEX8). Source data are provided with this paper.

## Code availability
All model code is publicly available on GitHub, https://github.com/AnneLoffler/AttractorNetwork-CoM.

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

## Acknowledgements

This work was supported by the European Research Council (Grant no. 323943), and by a Fellowship of the Max Planck Society (Max Planck School of Cognition) awarded to PH. The authors thank Lucie Charles and Eoin Travers for helpful comments on a previous draft of this paper.

## Author contributions

A.L. and P.H. conceived and designed the experiments. A.L. performed the data collection and analysis. A.L., A.S., and Z.F. implemented the computational model and performed the model simulations and fitting. A.L. drafted the paper and all other authors provided crucial feedback and revisions. All authors approved the final version of the paper.

## Competing interests

The authors declare no competing interests.
