## [Peer Review File · Nature Communications]

Reviewers' comments:

Reviewer #1 (Remarks to the Author):

The manuscript titled "Two ways to change your mind: Effects of intentional strength and motor costs on changes of intention" investigates changes of mind (CoM) caused by various factors, particularly, external (random-dot motion) stimulus, endogenous voluntary intention (colour of the choice targets), and motor costs (movement trajectory towards choice targets). The study nicely combines both novel experimental task paradigm and computational (neural network) modelling. The manuscript is overall well written.

Unlike previous studies that investigate CoM based on perceptual choices, the experiment in this current work first involves voluntary (endogenous) intentions/decisions (choosing a choice target colour) followed by perceptual motion discrimination. This is a novel task paradigm, as it reveals how CoM can arise not only from the usual sensory evidence (of dot-motion) but also from change of intention (by subsequent selection of a different colour). The data shows that changes of intention become more distinct in conflicting trials in which the coherent dot-motion is opposite in direction to that of the initially selected colour. Further, motor cost seems to have an influence on change of intention - there is tendency to minimise motor cost. Hence, internal goal or intention can be readjusted due to external environment/stimulus. An extensive neural network model, that consists of several nodes/modules, is simulated to account for these behavioural observations. Also, in the experiment, lower sense of agency (SoA), a subjective experience of control on actions/outcomes, seems to be associated with longer movement times although the results are not too conclusive.

From a wider perspective, incorporating endogenous cognitive processing in perceptual choice task is not too new. This include previous studies on reward-based perceptual decision-making (e.g. by WT Newsome and colleagues) or more generally cognitive control (e.g. Cohen, Dunbar & McClelland, 1990). That said, from the perspective of CoM, I believe this work is an excellent contribution to the field. However, I have some queries and comments for the authors.

Major queries and comments:

1. There is limited description of the data. For example, movement/reaction times for correct (CoM and non-CoM) and error (CoM and non-CoM) trials are not described/presented. Also, are there any impulsive trials in which movement is made prior to stimulus offset? Showing a movement/reaction time distribution would be helpful. Further, are there overall less correct-to-error CoMs than error-to-correct CoMs? In the model, how is CoM defined? When trajectories change directions (with condition as in the experiment; e.g. lines 810-812)? Are there any late CoM trials? Some of these data could be mentioned in the Supplementary Material.

2. Differences in reaction times (RTs) seem to be very small, e.g. between test and easy trials the

difference is ~40 ms while standard deviation SDs are ~40-60 ms, and between easy and conflict trials the difference is ~15 ms (the p values in lines 247-248 are not small). Why are the differences so small? How can your claims be certain?

3. Regarding the experimental task, it is unclear what the % of coloured dots in Outcome (step 4 in Fig. 1) is for. Is it even necessary? The main text for this is also not described clearly. E.g. line 441- has some description, but it is unclearly described.

4. The description of the model is limited. There is no justification of why a nonlinear mean-field model is selected, and not simpler neural network model (e.g. Bogacz et al., 2006). How are the inputs to the intention I's selected in each trial? At each time point, what is the mapping from the firing rates of modules A's to spatial coordinates in motor trajectory space? How to convert distance to target to the cost for module C's (e.g. in Fig. S4.1, top right corner)? Why the model cannot reach the choice targets (e.g. Fig. 6A bottom)?

5. In the model, the decrease in noise due to stronger intention does not seem to be neurobiologically plausible. Why cannot an increase in signal from intention be sufficient? The motivation/evidence for this noise reduction is based on abstract models (e.g. leaky integrate-and-fire) and may perhaps not be directly applicable to more biological mean-field models(?).

6. The CoM in the model is based on pure noise fluctuations as in Albantakis & Deco and other modelling work. How should this be reconciled with more recent work by Atiya et al., PLoS Computational Biology? In fact, Atiya et al., Nature Communications 2019 showed that pure noise fluctuations in the same mean-field model cannot lead to CoM nor account for a variety of CoM patterns.

7. Modelling the movement execution (lines 927-930) is not quite realistic. Also the stopping of the firing rates after the non-decision time is unrealistic - the choice target has to be reached to register a choice as in the experiment. Compare this with Atiya et al., 2019, in which the motor output is a continuous process.

8. In equation 1 (line 905), should not the equation have the weight W multiply by the synaptic gating variables s , as in Wong and Wang (2006), instead of the firing rates r ?

9. In equation 3 (line 912), should not the noise be generated using Ornstein-Uhlenbeck process as in Wong and Wang (2006)?

10. Is equation 4 (line 921) implemented in an ad hoc manner? Or is there theoretical derivation or experimental evidence to support this particular form?

11. How do you know intention biases the model constantly over time as shown in Fig. S4.1, and not affect transiently, e.g. as in reward biased perceptual decisions (e.g. see Newsome's work)?

12. The modelling could be much clearer if there are trial-averaged firing rates plotted.
13. Having the background/initial firing rate of the network at 10 Hz (line 923) may not be stable. Please demonstrate evidence that this is a stable steady state of the network.
14. Why are the SoA results not modelled?
15. No testable model predictions has been provided. Please provide them.
16. The Abstract and Introduction need to be written more clearly and concisely. For example, in the Abstract the task is not described clearly - what makes it novel(?). I have to read the main text to understand it. Also, there is no mentioning of the SoA results in the Abstract. In the Introduction, previous works seem to be stated for sake of stating without saying anything about their issues or bottlenecks facing them. Particularly, the motivation of the work needs to be stated more clearly. The Introduction reads a bit too long and there are abrupt jumps from one paragraph to another.

Minor/specific comments and suggestions:

1. The notion of referring the term "Changes of Movement" to lower level sensorimotor action selection is not very clear, and may even be problematic, as internal intentions can also be reflected in the movement changes. There should be clearer definitions. Perhaps I am missing something here.
2. Line 65. "... circumstances." Please provide citation.
3. Line 146. "25/50/75/100% of dots ..." Explain what this is for.
4. Lines 178-179. Are the RTs for CoM or non-CoM trials?
5. Fig. 2C. Please state the correlation coefficient value. Also, why are there negative RT costs, i.e. easy trials take longer than conflict trials? The model does not seem to be able to exhibit this (Fig. 6C)?
6. Line 217. State the value of the extra motor costs.
7. Lines 385-386. Would this results be consistent with work such as Burk et al., PLoS One 2014, and more recently, Zgonnikov et al., Judgment & Decision Making, 2019?
8. Line 591. How is the Yan et al. 2016 work used in this current work?
9. Lines 705-710. There is description of higher level or top-down cognitive processing/control on lower sensorimotor information. Perhaps Atiya et al.'s works could also be discussed here?

10. Why the return trajectories in CoMov trials typically avoid the initial starting point? Wouldn't such deviations take longer and are not as optimal as moving straight back through the initial starting point on their way to the other choice target?

11. The (spatial) quantification/measure for motor costs should be made clearly right up front. Initially I thought it was based on movement/reaction times. But the motor costs based on time generally seem to be generally small (< 100 ms).

12. Table S3.2. Why is there still quite a big difference in reaction times between model and experiment? Can this be optimised further? What is meant by "double CoM"? Changing decisions twice?

13. Fig. S4.1, bottom. Please show samples of the various CoM simulated trials here, to be consistent with the above panels.

14. In conflict trials, given that the discrimination of motion coherence takes precedence over the initially selected colour (intention), any "change of intention" may not be a true voluntary change in intention. Should this still be considered a change of intention? Perhaps this should be discussed maybe in the Discussion section?

Reviewer #2 (Remarks to the Author):

The manuscript "Two ways to change your mind: Effects of intentional strength and motor costs on changes of intention" presents behavioural experiments and a model aimed at investigating changes of mind based on external and internal information. In our opinion, this manuscript would be a great fit for Nature Communications. We find the data convincing, the model compelling, and think the paper is a good example of high-quality science. The authors anticipated several concerns about their complex experiment, and address each of these concerns with appropriate analyses. We especially appreciate the authors making code publicly available (in both Matlab and Python too!). Overall, we are very happy to recommend it for publication.

Before publication, we think two major concerns listed below need to be addressed. Minor concerns listed below are merely suggestions, mainly about how to make the manuscript more accessible to a wider audience.

Major Concerns:

1) We think that the authors are overinterpreting CoMov+Int and under-interpreting vertical changes of mind. First, we think the authors did a good job of convincing us that participants in this experiment

actually cared about color, at least some of the time. They show that despite a higher motor cost, participants generally expend more effort to stick with their chosen color rather than change colors (even though participants are being paid for left/right choices and not color).

Second, participants sometimes changed color choices on the same side of space (i.e., vertical changes; ~3% of all trials in experiment 1). In our reading, the authors interpret these changes as errors - trials where participants were simply confused about where their preferred color was (footnote page 12). The authors argue (top of Page 13) that since participants made more vertical color changes on late-target-onset trials, then this suggests that vertical changes are due to difficulties in target detection. We disagree and think that most changes of mind (about dot-motion, color, etc.) would increase under less preparation time. Further in the footnote of Page 12, the authors argue that "there was no reason for participants to change their mind about the target colour unless they had changed their mind about the dot-motion direction". We disagree here as well. There was no reason for participants to care about color at all, and yet it seems in this experiment that they did. In fact, we find vertical color changes in these data very interesting and compelling. On these trials, there is absolutely no reason for participants to change their movement from one color to another on the same side - it doesn't impact their monetary outcomes, and it costs energy. Yet participants do this anyway. We think that these trials reflect some combination of color errors due to noise, but also genuine changes in color preference online.

Third, participants sometimes changed side of space (i.e., their choice about dot motion), and also changed color (i.e., CoMov+Int). In our reading, the authors interpret these trials as genuine changes of mind about color preference. We find this confusing. If dot-motion and color evidence are independent (as is discussed in the paper, implemented in the model, and as we assume as well), then how can genuine changes of color preference only happen when dot-motion changes happen? There are even descriptively more solely color changes (vertical; 3.2%) than combinations of dot-motion and color changes (CoMov+Int; 1.7%). We also interpreted the authors as arguing that in CoMov+Int trials, participants cared about colour and genuinely changed their color preference. We think that, in contrast to vertical trials, participants did have a reason to make a horizontal change - less energy cost. In this case, if a participant changed their mind about dot-motion, but wanted to conserve energy, they could make a horizontal change. In this case, participants did not need to think about color at all on these trials. In this way, we think the authors are not holding vertical changes and CoMov+Int to the same logical standard.

Overall, our interpretation is that there are actually three interesting kinds of changes of mind in these experiments - those solely about dot-motion, those solely about color, and those regarding both. We think that any of these changes are going to be made up of genuine changes based on evidence, system noise, and errors. We strongly recommend reformatting the arguments of this manuscript to equally consider all types of changes of mind. In our opinion, these data are very interesting and the manuscript in its current form is underselling these results.

2) As we read it, dot motion sensory evidence in the model is represented as fixed inputs over time on a single-trial basis (as in the top left panel of Figure S4.1). However, motion evidence on a single-trial level in the random dot motion task is quite variable from timepoint-to-timepoint and trial-to-trial, especially at low dot motion coherences like in the current manuscript. As-is the model in the manuscript implies that perceptual changes are driven solely by internal neural noise, rather than a combination of neural noise and external changes in the stimulus. However, we know that moment-to-moment fluctuations in dot motion energy influence choices and changes of mind. Resulaj et al., 2009 Figure 3a has a great depiction of average motion energy contributing to all trials vs. those when a Change of Mind occurred.

We highly recommend either addressing this simplification/limitation in the manuscript, or alternatively updating the model to account for noisy stimulus input (structured in a way consistent with noise in the dot motion stimulus). We know of two great resources for calculating motion energy given frames of dot motion stimuli: (1) https://github.com/KianiLab/Waskom_JVision_2018, and (2) <https://github.com/arielzylberberg/MotionEnergy>

We see in the Matlab and Python code provided, that the authors have normally-distributed noise for color and dot motion inputs commented out (e.g., Line 159 in ANM.py), and are interested as to why this is the case.

In contrast, we think that modelling color preference inputs as fixed over time is a reasonable assumption. Although related to point #1, this assumption would mean that any changes of color are errors, or correcting for initial errors. In our opinion, changes of dot-motion are especially interesting because, due to the noisy stimulus, changes can also go from initially-correct to also correct options (in the case of dot-motion in opposing directions at the beginning and end of a trial) - which speaks to our ability to adapt movements online to changes in the world.

Minor Concerns:

1) Figure 3C caption reads "Predicted effect [...]". Please clarify if this is real participant data from Experiment 2.

2) This manuscript had many abbreviations compared to other papers, which made it difficult to read. For example, the first paragraph of Page 19 referenced CoM, MT, SoA, CoMov, CoMov+Int, etc. We feel reasonably familiar with these topics and still found these passages hard to read. We are worried other readers outside of this specific area may have an even harder time deciphering these passages. Any effort to spell out some of these words, or substitute language, especially in these dense sections, will help improve the accessibility and impact of this work in our opinion.

3) On Page 9, Line 209, the authors mention MELR, which is not defined until the Methods section.

4) Early vs. late target onset is first referenced in the results in an analysis. This factor is discussed in the full methods appearing at the end of the manuscript, but it would be helpful to mention this factor in

the abbreviated methods within the main text.

5) Page 4, Lines 77 onward, defined Changes of Movement as involving changes of "motor intentions". Then the authors immediately go on to talk about the independent concept "Changes of Intention". We understand that these are technical terms, but the intermixing of words made this argument difficult to follow. Perhaps changing the names of these terms might help? For example, Changes of Movement is confusing because participants need to change their movement to reflect either a new goal about dot motion, about color choice, or both. Perhaps terms like a perceptually-based change of mind and an internally-based change of mind.

6) Labels for bar colors in figure 3C would be appreciated.

7) On the first paragraph of Page 17, We find the followup analysis of an interaction effect with a p-value of 0.097 unnecessary.

8) Selfishly, we recently published a very relevant review on this topic that might make a nice addition as a citation:

Wispiński, N. J., Gallivan, J. P., & Chapman, C. S. (2018). Models, movements, and minds: bridging the gap between decision making and action. *Annals of the New York Academy of Sciences*.

...since this experiment does a great job of exactly filling the void we propose exists, it would be great to link these two papers!

Craig S. Chapman & Nathan Wispiński

Reviewer #3 (Remarks to the Author):

The authors aimed to examine how internal intentional and external environmental factors influence the change of mind and to provide an attractor network model as a tool to explain observed outcomes.

I found that the attempt to combine internal and external factors are interesting. However, I am not convinced that the outcomes they acquired can be generalized beyond this specific experimental paradigm. First of all, entire their arguments are depended on a couple of % differences in CoM frequency. Furthermore, their classification of CoM is much generous than previous studies and yet the overall occurrence itself is pretty low. It is unsurprising that internal conflict can be reflected in curved trajectories and combined with external factors (e.g. Dotan et al., 2019 TICs). Also it should reflect the strength of subject intention and importance. I think the authors have ignored accumulated over a decade of research using continuous trajectories and heavily relying on a particular study -- which happened to use the term CoM first, making it something new -- and build upon a complex argument. Therefore, I am not fully convinced by the novelty of claims and the generalizability/reproducibility

beyond this specific task. It might have a better fit for much-specialized journal. I also found that the attractor network model is a reason to try. However, it is unclear why this model is much better than any other existing models.

Reviewer #1 (Remarks to the Author):

The manuscript titled "Two ways to change your mind: Effects of intentional strength and motor costs on changes of intention" investigates changes of mind (CoM) caused by various factors, particularly, external (random-dot motion) stimulus, endogenous voluntary intention (colour of the choice targets), and motor costs (movement trajectory towards choice targets). The study nicely combines both novel experimental task paradigm and computational (neural network) modelling. The manuscript is overall well written.

Unlike previous studies that investigate CoM based on perceptual choices, the experiment in this current work first involves voluntary (endogenous) intentions/decisions (choosing a choice target colour) followed by perceptual motion discrimination. This is a novel task paradigm, as it reveals how CoM can arise not only from the usual sensory evidence (of dot-motion) but also from change of intention (by subsequent selection of a different colour). The data shows that changes of intention become more distinct in conflicting trials in which the coherent dot-motion is opposite in direction to that of the initially selected colour. Further, motor cost seems to have an influence on change of intention - there is tendency to minimise motor cost. Hence, internal goal or intention can be readjusted due to external environment/stimulus. An extensive neural network model, that consists of several nodes/modules, is simulated to account for these behavioural observations. Also, in the experiment, lower sense of agency (SoA), a subjective experience of control on actions/outcomes, seems to be associated with longer movement times although the results are not too conclusive.

From a wider perspective, incorporating endogenous cognitive processing in perceptual choice task is not too new. This include previous studies on reward-based perceptual decision-making (e.g. by WT Newsome and colleagues) or more generally cognitive control (e.g. Cohen, Dunbar & McClelland, 1990). That said, from the perspective of CoM, I believe this work is an excellent contribution to the field. However, I have some queries and comments for the authors.

Authors' reply: We thank the reviewer for their overall positive evaluation of our manuscript and the detailed comments on how to further improve the manuscript. We further thank the reviewer for pointing out previous studies investigating endogenous components in perceptual decision making, in particular with regard to the integration of reward-based and sensory information. We have added the following reference to the work by Rorie et al. (2010) in the introduction (page 5):

Rorie, A. E., Gao, J., McClelland, J. L., & Newsome, W. T. (2010). Integration of sensory and reward information during perceptual decision-making in lateral intraparietal cortex (LIP) of the macaque monkey. *PLoS one*, 5(2), e9308. <https://doi.org/10.1371/journal.pone.0009308>

Please note that in response to comments raised by both reviewers 1 and 2, we have now changed the terms for different types of Changes of Mind (CoM) in our manuscript. We now refer to CoM that only involves a perceptual change (without change in colour intention) as 'perceptual CoM' (CoM_P) instead of 'Change of Movement' (CoMov) in our previous version. CoM that involve both a change with respect to the perceptual decision *and* colour intention are now referred to as 'perceptual + intentional CoM' (CoM_{P+I}) instead of 'Change of Movement + Intention' (CoMov+Int) in our previous version. We have replaced these terms throughout our manuscript and also use the new terms

in our replies to the reviewers' comments below. We have furthermore added a glossary of these terms and their abbreviations, together with brief definitions, at the beginning of the manuscript in order to facilitate understanding. Finally, we have changed the title of our manuscript to "Alternative Ways to Change Your Mind: A Hierarchical Attractor Network Model of perceptual vs. intentional decision updates."

Major queries and comments:

1. There is limited description of the data. For example, movement/reaction times for correct (CoM and non-CoM) and error (CoM and non-CoM) trials are not described/presented. Also, are there any impulsive trials in which movement is made prior to stimulus offset? Showing a movement/reaction time distribution would be helpful. Further, are there overall less correct-to-error CoMs than error-to-correct CoMs? In the model, how is CoM defined? When trajectories change directions (with condition as in the experiment; e.g. lines 810-812)? Are there any late CoM trials? Some of these data could be mentioned in the Supplementary Material.

Authors' reply: We thank the reviewer for pointing out the missing details in the description of our data. We have now added the following information in the main *Results* section for Exp. 1 (page 8): "[...], in line with previous findings (e.g., Resulaj et al., 2009), the majority of CoM in test trials ($M = 60.9\%$, $SD = 16.5\%$, $t(16) = 2.72$, $p = .015$, $d = 0.66$) corrected an initial perceptual error (i.e., the response would have been an error had no CoM occurred). This suggests that perceptual CoM in our task was driven by continuous integration of sensory evidence after an initial response had already been initiated."

We have further added the following information for Exp. 2 in Supplementary Material S2: "[...], similarly to Exp. 1, the majority of perceptual CoM in Exp. 2 corrected an initial error ($M = 61.5\%$, $SD = 10.8\%$, $t(15) = 4.26$, $p < .001$, $d = 1.07$). A mixed-effects logistic regression model with accuracy as outcome variable and CoM type (CoM_p/CoM_{p+1}) and Exp (1 vs. 2) did not reveal any significant main effects or interactions (all $p > .05$), indicating that across both experiments, accuracy was comparable for different types of CoM."

Furthermore, we analysed reaction and movement times in more detail. We have added a new section 'Reaction and movement times for CoM vs. no-CoM trials' to Supplementary Material S3, including the following information and figure:

"Figure S3 illustrates RT and MT distributions in test trials for error vs. correct responses, separately for each experiment and CoM vs. no-CoM trials. For illustration purposes, RTs and MTs were z-scored for each subject. A linear mixed-effects regression with (non-standardized) RTs as outcome variable and Experiment (1 vs. 2), Error (correct vs. error), and CoM (no-CoM vs. CoM) as fixed effects revealed that RTs were significantly slower in error trials compared to correct trials ($b = 7.76$, 95% CI [0.78, 14.74], $t(12344.8) = 2.18$, $p = .029$). This was particularly pronounced in Exp. 2 (interaction Experiment x Error: $b = -8.96$, 95% CI [-17.35, -0.57], $t(12342.6) = -2.09$, $p = .036$). None of the other main effects or interactions were significant (all $p > .05$). Note that only correct trials were included in all the RT analyses reported in the main text, thus avoiding that differences in RTs between correct vs. error trials confounded RT results.

For MTs, a significant effect of CoM was observed ($b = 484.4$, 95% CI [439.8, 528.9], $t(12401.1) = 21.31$, $p < .001$), reflecting the fact that CoM trials were associated with longer MTs than no-CoM trials. This effect was particularly pronounced in error trials (sign. interaction CoM x Error: $b = 85.4$,

95% CI [15.6, 155.2], $t(12395.7) = 2.40$, $p = .017$). The prolongation of MTs in error vs. correct CoM may have been caused by 1) erroneous CoMs occurring later during the movement than correct CoM, thus increasing overall path length, and/or 2) participants slowing down due to uncertainty in erroneous CoM compared to correct CoM. While we observed a modest trend for both potential mechanisms (results not shown), neither effect reached statistical significance ($p > .05$). Hence, the prolongation of MTs in error vs. correct CoMs may have resulted from a mixture of effects across different participants/trials. Finally, a main effect of Experiment on MTs was observed ($b = 227.9$, 95% CI [76.2, 379.5], $t(33.4) = 3.03$, $p = .005$), which was expected given that movement paths were longer in Exp. 2 than Exp. 1 by design (see *Methods* section). As expected, this difference was particularly pronounced in CoM trials (sign. interaction between Experiment x CoM: $b = 103.7$, 95% CI [51.7, 155.6], $t(12400.3) = 3.91$, $p < .001$).”

Figure S3. Histograms of reaction times (A) and movement times (B) in test trials, separately for Exp. 1 and 2. RTs and MTs were z-scored for each subject for illustration purposes. Trials were then split into error (red) vs. correct trials (blue), and no-CoM (top) vs. CoM trials (bottom).

Regarding the reviewer’s question about impulsive trials: Given that this task was a reaction time task, there were no trials in which movements were made prior to stimulus offset (since stimulus offset is locked to the subject’s response initiation; see pages 6 and 35). If subjects moved before stimulus *onset*, a “too early” warning message was shown and the trial was repeated later on during the experiment (page 35).

Concerning the classification of CoM in the model: CoM was defined based on neural firing rates of the action nodes, as described in the caption of Figure 5: “Hence, CoM can occur when one action node crosses the threshold for movement execution first, but later on, another action wins the competition.” We further added the following to the main text to emphasise this aspect (page 24–25): “In the model, a trial was considered to be a CoM if one action node crossed the firing rate threshold at first, but later on a different action node crossed the threshold (and also surpassed all other nodes by 10 Hz to ensure a clear winning action).”

Based on these criteria, we can be confident that trials classified as CoM in the model were not driven by small, noisy fluctuations in firing rates, but instead, incurred a reliable switch between different action nodes. This approach to classifying CoM in neural network models was based on previous

work reported by Albantakis and Deco (2011). Importantly, the way we classified CoM based on firing rates is identical to a classification based on movement trajectories in our model. This is because, for simplicity, we did not assume any noise in the models' output (e.g., movement noise). Therefore, there is a perfect correspondence between the winning action node and the movement trajectory produced by the model (see below for a more detailed discussion of motor outputs produced by our model). By contrast, in the actual data, movement trajectories are noisy, and hence, additional criteria were applied to ensure that changes in participants' movement directions reliably indicated a CoM. That is, the initial movement had to exceed 10% of both the x- and y-distance towards one target, and then end at a different target (see *Methods* section; page 35). Thus, in conclusion, in both the experimental data and our computational model, we ensured that CoM classification was reliable and was not driven by small, noisy fluctuations in neural firing (model) or participants' movements trajectories (experimental data).

Regarding the reviewer's question about "late CoM trials": Trials were classified as CoM based on the criteria mentioned above, regardless of the point in time at which they occurred (as long as they occurred after 10% of the x- and y-distance to a given target). On average, CoM occurred roughly halfway through the movement, at 44.99% ($SD = 7.73\%$) of the *distance* towards the initial target (we have now added this on page 36 of our manuscript). However, note that the latency of the CoM of course depends on movement speed. Crucially, since the dot-motion stimulus disappeared immediately at response onset, the sensory evidence driving CoM always occurred *before* the initial response – most likely within the last 300-400 ms before response onset/stimulus offset, according to the non-decision time. The precise point in time at which the CoM decision was made on a single trial is difficult to reverse engineer based on kinematics alone, and given that this was not the primary focus of our work, we did not attempt to further differentiate CoM trials according to early vs. late CoM onset.

2. Differences in reaction times (RTs) seem to be very small, e.g. between test and easy trials the difference is ~40 ms while standard deviation SDs are ~40-60 ms, and between easy and conflict trials the difference is ~15 ms (the p values in lines 247-248 are not small). Why are the differences so small? How can your claims be certain?

Authors' reply: While the absolute differences in average RTs may be small, our reported effect sizes of Cohen's d indicate medium to large effects (e.g., $d = 0.97$ for the RT difference between test and easy trials). This is because the within-subject difference in RTs between easy and test trials is highly consistent across subjects (standard error of mean difference between test vs. easy trials: SEM = 8.69 ms). Thus, even relatively small differences in RTs between conditions can yield large and reliable effects as long as the condition effect is consistent across subjects. Please note that the reported standard deviations of 40-60 ms do not capture these within-subject differences since they simply refer to SDs of average RTs *across* participants in a given condition. By contrast, the reported statistical test (paired t -test) compares changes in RTs between test and easy trials *within* participants. Our inferences are based on the within-participant contrasts. The history of mental chronometry shows several latency effects with means that are numerically small, but nevertheless highly reliable. We provided effect sizes and confidence intervals throughout the manuscript to facilitate interpretation of our results.

3. *Regarding the experimental task, it is unclear what the % of coloured dots in Outcome (step 4 in Fig. 1) is for. Is it even necessary? The main text for this is also not described clearly. E.g. line 441- has some description, but it is unclearly described.*

Authors' reply: We thank the reviewer for pointing out the lack of clarity regarding the % of coloured dots. Please note that the colour manipulation was largely implemented to encourage participants to use the SoA scale sensitively. It thus served as a sanity check for SoA modulations, instead of being a variable of interest to our main hypotheses. We have now added the following clarification on pages 18–19:

“We manipulated the percentage of dots that was coloured in the chosen target in order to increase variance in SoA judgments. Specifically, we predicted that SoA would be higher the more dots matched participants' initial colour intention (Sato & Yasuda, 2005; Kawabe, 2013; Chambon, Sidarus, & Haggard, 2014). More importantly, we assumed that in addition to the match between intended and obtained action outcome, SoA ratings would be modulated by whether or not a CoM occurred. In order to test this, the percentage of dots painted in the chosen colour was always 50% in trials with CoM, allowing us to investigate whether and how different types of CoM affected SoA while keeping the action outcome (colour percentage) constant.”

4. *The description of the model is limited. There is no justification of why a nonlinear mean-field model is selected, and not simpler neural network model (e.g. Bogacz et al., 2006). How are the inputs to the intention I's selected in each trial? At each time point, what is the mapping from the firing rates of modules A's to spatial coordinates in motor trajectory space? How to convert distance to target to the cost for module C's (e.g. in Fig. S4.1, top right corner)? Why the model cannot reach the choice targets (e.g. Fig. 6A bottom)?*

Authors' reply: Please note that our computational model is in fact a simplified neural network model. That is, instead of modelling activity of individual neurons, we approximated mean activity rates of neural populations ('nodes') based on a linear combination of inputs they receive. We further used a simple, linear approximation of the differential equations describing updates in neural firing rates over time (see Equation 2: Euler-Maruyama method as described in Miller, 2016; Hahne et al., 2017). In that sense, our model is very similar in complexity to the models described in Bogacz et al.'s work (Bogacz, Brown, Moehlis, Holmes, & Cohen, 2006). Bogacz et al. report that more complicated models can be reduced to simple drift diffusion models. The main reason we cannot further reduce our current model to a drift-diffusion model is that in our task, participants need to integrate multiple sources of evidence to make decisions (exogenous sensory evidence, endogenous voluntary intention, and motor costs). These decision variables are conceptually different and are presumably encoded in distinct neural structures in the brain (i.e., 'neural nodes' in our network). While drift-diffusion models are highly successful in modelling decisions involving a single decision variable, network models more readily allow for integration of a multitude of dynamically varying decision variables (Lo & Wang, 2006; Cisek, 2012; Christopoulos, Bonaiuto, & Andersen, 2015). Moreover, network models provide a neurobiologically-plausible framework for how these mechanisms may be implemented in the brain. That is, interconnectivity between brain areas that encode different sources of decision evidence allows for continuous integration of decision-relevant, multi-modal information. Thus, our model provides a mathematically parsimonious, yet conceptually

valid, account of the mechanisms through which multiple decision variables are dynamically integrated over time.

The input to the intention nodes I was obtained through model fitting. Note that the strength of these inputs corresponds to the strength of colour intentions. We did not fit this parameter (or any of the other model parameters) on a trial-by-trial basis, but instead, aimed to model participants' average performance in test trials. Hence, the fitted value of the intentional input can be interpreted as the overall strength of colour intentions, across the task. While it is possible that intentional strength varies on a trial-by-trial basis, we did not have a single-trial indicator of intentional strength, but instead, only measured intentional strength across trials, as indicated by the RT cost in conflict trials. Note that fitting average performance is standard practice in computational modelling: That is, a single parameter estimate is obtained across the whole task, instead of fitting parameter values to individual trials.

The simulation of movement trajectories produced by the model is described on page 41 of the manuscript. We acknowledge that some aspects could have been described in greater detail, and have thus edited the paragraph to provide further clarification:

“Movement direction corresponded to the chosen target location (according to the winning action node) and movement speed was constant at 0.7 pixels/ms, resulting in a movement duration of ~450 ms for no-CoM trials straight movement trajectories, in line with movement times measured in Exp. 1. For simplicity, movements were simulated with straight trajectories towards the chosen target, without additional motor noise. While this does not result in realistic trajectory shapes observed in reaching tasks, it provided us with sufficient detail to approximate motor costs by calculating the Euclidean distance between the current cursor location and each of the 4 targets. As cursor position changes, motor costs associated with each target change according to the relative distance to each target at a given point in time (see Fig. 3B). Movement execution towards a chosen target continued even if the action node dropped below the threshold, unless another action node reached the firing rate threshold, in which case the movement was redirected towards the new target choice. Firing rates continued to be updated for 380 ms after initial threshold crossing due to a non-decision time consisting of sensory delays of 200 ms and motor delays of 180 ms (Albantakis & Deco, 2011). This allowed for decision updates even after initial action onset. In line with previous models (e.g., Resulaj et al., 2009; Albantakis & Deco, 2011), firing rate updates were stopped after the non-decision time (i.e., no more CoM could occur since no more new evidence was obtained) and the movement was completed according to the final target choice.”

Please note that the latter aspect is the reason for why the model does not reach the target location: The updating of firing rates, and hence occurrence of CoM, is limited by the non-decision time because the stimulus disappears at action onset. Hence, after action onset, only stimulus information that has not been evaluated yet (due to sensorimotor delays) can realistically inform decision-making. We thus only continued model simulations for the duration of the non-decision time of 380 ms. After the non-decision time, a point of ‘no return’ is reached causing the currently chosen target to be the final choice. Note that this approach was necessary by design since the stimulus disappeared at action onset. Furthermore, this approach is in line with the vast majority of previous models in which decision updates were limited by the duration of the non-decision time (Resulaj et al., 2009; Albantakis & Deco, 2011; Albantakis, Branzi, Costa, & Deco, 2012; Burk, Ingram, Franklin, Shadlen, & Wolpert, 2014; van den Berg et al., 2016).

Finally, regarding the computation of motor costs: The transformation of distance from each target to inputs into the cost nodes is described on page 42 (below equation 6): “[...], the input into cost nodes was set to an equal value of 60 Hz at trial start. Once a movement was initiated, costs were updated relative to changes in Euclidean distance between the current position and each target location.” For example, if distance to one target doubles relative to the initial distance, inputs into the corresponding cost nodes double. Thus, for simplicity, motor costs only considered the Euclidean distance to each one of the 4 possible target locations. While other aspects (e.g., biomechanical costs; Cos, Bélanger, & Cisek, 2011) may further contribute to movement costs, the simplified approximation of costs based on distance seemed sufficient in capturing participants’ behaviour in our current task.

5. In the model, the decrease in noise due to stronger intention does not seem to be neurobiologically plausible. Why cannot an increase in signal from intention be sufficient? The motivation/evidence for this noise reduction is based on abstract models (e.g. leaky integrate-and-fire) and may perhaps not be directly applicable to more biological mean-field models(?).

Authors’ reply: We thank the reviewer for raising these concerns. Hierarchical noise control in our model was motivated by evidence from neuroimaging/EEG studies that have shown that such top-down modulation of neural noise is in fact neurobiologically plausible, and may be an important hallmark of voluntary action control. Most notably, a previous EEG study showed that *variability* (i.e., noise) in the ‘readiness potential’ reduces gradually prior to onset of voluntary actions (Khalighinejad, Schurger, Desantis, Zmigrod, & Haggard, 2018). This noise reduction was stronger when actions were initiated endogenously as opposed to exogenously, thus suggesting that neural noise reduction may be a neural marker of endogenous action control. More broadly, other studies have proposed that higher-order areas representing abstract information (e.g., goals) may exert top-down control over lower-level sensorimotor areas through noise control (O’Reilly, 2010; Badre & Nee, 2018). The general principle that neural networks achieve functional processing through variability quenching has been modelled in detail elsewhere (Churchland et al., 2010). Given the fact that colour intentions in our task were 1) endogenously generated and 2) represented a conceptually higher-order goal of the action with respect to its outcome, we believe that noise control through colour intentions is a valid, and indeed, crucial component of our model. In fact, a model without hierarchical noise control performed significantly worse than the model with hierarchical noise control (see *Results* section page 26 and *Methods* section page 44). Furthermore, removing the hierarchical structure from our model would mean that sensory evidence and endogenous intentions are equivalent conceptually and mathematically. However, based on our task design and prior research, our model necessitates distinct computational mechanisms underlying higher-order endogenous intentions vs. lower-level sensory processing.

Based on the literature reviewed above, top-down neural noise modulation provides a plausible candidate mechanism through which such a distinction may be achieved, and we show that it can be readily applied to Attractor Network Models. Even though the basic idea for hierarchical noise control was initially developed for more abstract Bayesian inference models (Mathys, Daunizeau, Friston, & Stephan, 2011; Mathys et al., 2014), a recent study showed that ‘volatility’ estimates in these types of models were directly linked to neural activity in prefrontal cortex, which in turn predicted choice volatility (i.e., switching rate; Deserno et al., 2020). On a neuronal level, such changes in neural/behavioural volatility may be mediated by the dopaminergic system. Specifically, the ‘dual-state’ theory proposes that the balance between *D1* and *D2* receptor activation affects signal-to-noise

ratio of neural activity, and hence, may be crucial for the balance between stability and flexibility of actions (Durstewitz & Seamans, 2008).

Hence, we believe that our modelling approach has strong empirical and theoretical foundations, and provides a neurobiologically plausible approach through which higher-order endogenous intentions and lower-level sensorimotor information are flexibly and dynamically integrated over time. We acknowledge that some of this background information was cut short in the original version of the manuscript. We therefore edited our introduction (page 5) and model sections (page 24) to include more details of the information provided above and motivate the need for a hierarchical structure in our model more clearly.

6. The CoM in the model is based on pure noise fluctuations as in Albantakis & Deco and other modelling work. How should this be reconciled with more recent work by Atiya et al., PLoS Computational Biology? In fact, Atiya et al., Nature Communications 2019 showed that pure noise fluctuations in the same mean-field model cannot lead to CoM nor account for a variety of CoM patterns.

Authors' reply: We thank the reviewer for pointing out the differences between our model and the model introduced by Atiya and colleagues (Atiya, Rañó, Prasad, & Wong-Lin, 2019; Atiya et al., 2020). The authors of that paper reported Changes of Mind in an attractor network model similar to ours. However, in contrast to our model, Atiya et al.'s model included an additional 'uncertainty-monitoring module' that established a direct link between decision uncertainty and subsequent decision reversals. The authors reported that they did not observe Changes of Mind without excitatory feedback from this uncertainty-monitoring module to lower-level nodes that integrate sensory evidence. As the reviewer points out, this suggested that noisy fluctuations alone could not account for behavioural patterns typically observed in CoM experiments. However, Atiya et al. also noted that while noisy fluctuations alone were not sufficient in their model, noise still played an essential role. In fact, the uncertainty-monitoring module received direct input from the sensory decision module. That is, high uncertainty in that model was, at least to some extent, still driven by noise in the sensory evidence itself. In that sense, Changes of Mind still largely depend on noise fluctuations in the Atiya model, as they do in our model and previous network models (Albantakis & Deco, 2011; Albantakis et al., 2012; Yan, Zhang, & Wang, 2016) as well as drift-diffusion models of CoM (Resulaj et al., 2009; Burk et al., 2014; van den Berg et al., 2016). We thus believe that our current model and the model proposed by Atiya et al. are more similar than they are different. While it would in theory be possible to introduce an additional uncertainty-monitoring module in our model, we do not believe that this would be within the scope of the current manuscript: 1) because we do observe significant numbers of Changes of Mind in our model even without an additional uncertainty-monitoring module, and 2) because we do not have experimental measures of participants' decision confidence that would justify adding additional model parameters to fit participants' behaviour.

7. *Modelling the movement execution (lines 927-930) is not quite realistic. Also the stopping of the firing rates after the non-decision time is unrealistic - the choice target has to be reached to register a choice as in the experiment. Compare this with Atiya et al., 2019, in which the motor output is a continuous process.*

Authors' reply: We acknowledge that the movement trajectories produced by our model are simplistic. However, there are 3 main reasons for why we did not model continuous movement trajectories as in the Atiya et al. (2019) paper.

1. The primary purpose of our model was not to replicate the precise, continuous trajectory shapes produced by participants, but instead, to capture in a categorical sense whether and when CoM decisions would be made. We only simulated movement trajectories in order to obtain estimates of motor costs, which were approximated based on target distance, and hence, did not require capturing the more fine-grained details of participants' movements (see discussion about motor costs in our reply to the reviewer's comment #4 above). Thus, the main testable predictions of our study can be captured by the current simplified version of the model, which treats trajectories as an approximation of average motor behaviour.
2. In this context, it is important to note that our model was optimized to fit participants' *average* behaviour. Fitting the precise, continuous movement trajectories on a trial-by-trial level would require a much higher level of complexity in our model in order to account for motor policies, learning, biomechanical constraints, movement noise, and other factors that contribute to motor behaviour on a single trial (Wolpert & Ghahramani, 2000; Todorov & Jordan, 2002; Wolpert & Landy, 2012). We do not believe that our current study has sufficient power to justify such an increase in model complexity, nor was it our goal to capture motor-related processes in such detail. Having said that, we believe that this would be a fascinating extension of our work, and we have in fact already addressed this as an important area for future research in our discussion (page 31): *"In this context, the model makes further predictions that can be directly tested in future studies. For example, it is plausible that firing rates of action nodes (i.e., the strength of decision evidence in favour of an action) are directly linked to more fine-grained, gradually-varying details of motor policies, such as movement speed or vigour, rather than mere categorical choices between action alternatives."*
3. As mentioned above (see our reply to comment #4 above), firing rate updates were stopped after the non-decision time because in our task, the stimulus was turned off as soon as an action had been initiated. Thus, after action onset, only stimulus information that had not been processed yet due to sensorimotor delays (i.e., the non-decision time) could drive decision updates. Hence, even though mere motor processing has to continue until the target is reached, the decision-making processes determining which target is selected are assumed to be terminated earlier on. This assumption is in line with the vast majority of previous models of CoM where decision-making processes were only modelled for the duration of time where new decision evidence was available (i.e., decision time + non-decision time; Resulaj et al., 2009; Albantakis & Deco, 2011; Albantakis et al., 2012; Burk et al., 2014; van den Berg et al., 2016).

On a more general note, we would like to point out that both our experimental design and modelling choices were informed by previous studies that used reaction time tasks. By contrast, the more recent study by Atiya et al. (2020) used a task design where stimulus duration was controlled experimentally, instead of being determined by participants' reaction time. In this study, the authors captured CoM that were driven by slower-acting mechanisms related to uncertainty monitoring, which can drive CoM later on during the trial. This work by Atiya et al. is an important contribution

and provides a cautionary tale indicating that there might be other potential mechanisms driving CoM, in addition to mere evidence-based updates. However, as discussed above (see our reply to comment #6), our model does not include such an uncertainty-monitoring module that could drive perceptual CoM independently of noise in the sensory nodes. We thus do not believe that modelling decision processes beyond evidence-driven updates would be informative in our current model/experimental design.

8. In equation 1 (line 905), should not the equation have the weight W multiply by the synaptic gating variables s , as in Wong and Wang (2006), instead of the firing rates r ?

Authors' reply: While our mean-field model was inspired by the general structure of Wong and Wang (2006), our model implementation is a further simplification, which is more strongly based on the approach introduced by Miller (2016). We have now stated this more clearly in our manuscript (pages 39–40). As mentioned above, in our model, the firing rate of a given node is simply a function of the weighted inputs it receives from other nodes (and external model inputs). Thus, as opposed to Wong and Wang, we did not model mechanisms on a synaptic/neurotransmitter level, and hence, did not introduce a synaptic gating variable s . Instead, in our model, we assumed a single base time constant τ of 100 ms corresponding to the effect of the slow NMDA receptors in Wong & Wang (2006). This is because Wong & Wang (2006) show that slow temporal integration is primarily mediated by NMDA receptors. However, note that synaptic gating variables were not modelled explicitly in our study, mainly because our model already has a large number of parameters due to its relatively complex structure (12 neural nodes and their reciprocal connectivity).

We have added the following sentence to our *Methods* section (page 40) to clarify this aspect: “*While our mean-field approach did not rely on explicit modelling of synaptic mechanisms, we introduced a base time constant τ of 100 ms for all neural nodes, imitating the effect of slow NMDA receptors, which have been shown to be the primary contributor to slow temporal integration of evidence (Wong & Wang, 2006).*”

9. In equation 3 (line 912), should not the noise be generated using Ornstein-Uhlenbeck process as in Wong and Wang (2006)?

Authors' reply: We thank the reviewer for pointing out the Ornstein-Uhlenbeck process as an alternative way to model neural noise. Note that Wong & Wang used an Ornstein-Uhlenbeck process to model the effect of AMPA. However, as mentioned above, our model focused on NMDA as a primary contributor to evidence integration, and hence, is a simplification of the Wong & Wang (2006) model.

Additionally, the hierarchical structure of our model was inspired by Hierarchical Gaussian Filters, where the variance of Gaussian noise is governed by activation in the hierarchical layer above (Mathys et al., 2011; Mathys et al., 2014). Thus, modelling neural noise as random Gaussian noise was a decision that followed naturally from implementing Hierarchical Gaussian Filters.

Implementing an Ornstein-Uhlenbeck process in this hierarchical structure is not trivial since the Ornstein-Uhlenbeck is a Gaussian random walk with a drift towards zero, and thus, the process needs

to maintain its own, independent state. Consequently, our initial implementation of hierarchy in the model cannot directly be translated to noise produced by an Ornstein-Uhlenbeck process.

However, to check the effect of alternatively using Ornstein-Uhlenbeck noise, we adapted the procedure in the following way: We replaced equation (3) with

$$r_{i,t} = r_{i,t} + x_{i,t} \quad \text{where} \quad dx_{i,t} = -\theta x_{i,t} dt + \sigma W_t \quad \text{with } \theta > 0$$

θ represents a free parameter that was manually tuned to ensure that the mean and variance of the model's noise was equal to the original model. Furthermore, in line with the original model, σ was regulated by higher ordered intentions, with stronger intentions causing a decrease in variability (see equation 4 in the manuscript).

Our analysis revealed that the model with Gaussian noise performed better (model cost: 2.58) than the model with Ornstein-Uhlenbeck noise (model cost: 15.49). The difference in model performance was largely due to the fact that the Ornstein-Uhlenbeck process caused the network to be more volatile. Specifically, the model produced a high percentage of early responses (response before stimulus onset: 46.6% of trials). In the remaining trials, the model overestimated the frequency of CoM (perceptual CoM: $M = 14.52\%$, $SD = 35.23\%$; perceptual + intentional CoM: $M = 9.01\%$, $SD = 28.63$) and produced double CoM in 35.5% of trials. While it may be possible to further adjust the Ornstein-Uhlenbeck process to produce more stable model behaviour, this would require additional adjustments that cannot be directly derived from our original implementation of Hierarchical Gaussian Filters in the Attractor Network Model. Given the importance of hierarchical noise control in our model, we are therefore convinced that random Gaussian noise is the most appropriate method for our purposes.

10. Is equation 4 (line 921) implemented in an ad hoc manner? Or is there theoretical derivation or experimental evidence to support this particular form?

Authors' reply: Equation 4 states that the decrease in noise in actions nodes is directly proportional to the current firing rates of intention nodes. This implementation was conceptually derived from Hierarchical Gaussian Filters where the strength of evidence on a hierarchically higher level is directly inversely related to variance on a hierarchically lower level (Mathys et al., 2011; Mathys et al., 2014). However, note that Hierarchical Gaussian Filters were originally developed within the framework of Bayesian inference. In our paper, we used an attractor network model as our main formalisation reference and introduced the conceptual detail of the hierarchical order based on Hierarchical Gaussian Filters. Although it would in theory be possible to implement more complex, non-linear noise modulations in our model, we aimed to use the most straightforward, parsimonious approach whereby hierarchical top-down control has a linear effect on noise and only depends on one free parameter (h) indicating the strength of top-down control. This parameter was then optimized to fit participants' performance. Importantly, a model without hierarchical top-down control ($h = 0$) performed significantly worse than our model with hierarchical control (see pages 26 and 44). Thus, our implementation of hierarchy captured participants' behaviour well, and in fact, was crucial in accounting for some of the variance observed in the data.

11. How do you know intention biases the model constantly over time as shown in Fig. S4.1, and not affect transiently, e.g. as in reward biased perceptual decisions (e.g. see Newsome's work)?

Authors' reply: We thank the reviewer for raising this important issue. Indeed, the question of how long-lasting vs. transient intentions are is very closely linked to the main question of our paper investigating the stability vs. flexibility of voluntary intentions. Crucially, our behavioural results showed that voluntary colour intentions were not only strong (thus causing performance cost when conflicting sensory evidence was presented), but were also highly persistent as indicated by a low frequency of Changes of Intention relative to purely sensorimotor Changes of Mind. This was true even when Changes of Intention incurred a high motor cost. This suggests that, even though colour choices were somewhat arbitrary, participants' actions were strongly informed by colour intentions throughout the duration of an ongoing trial. Consequently, the assumption in our model that endogenous intentions have a persistent effect throughout an ongoing action is directly supported by our behavioural data. In general, an intention can be considered a future-oriented mental state, that persists until its satisfaction conditions are met - unless some additional event warranting CoM occurs.

With regard to the reviewer's reference to the work by Newsome's group (Rorie et al., 2010): It is important to note that our task design is very different from previous studies showing that reward can transiently bias perceptual decisions. In these tasks, choosing the high reward option when sensory evidence is weak increases overall reward rate, and thus, is optimal. Thus, if a given choice option is consistently associated with high rewards, perceptual choices will be biased towards that choice option. By contrast, in our design, colour choice and sensory evidence were completely independent (randomization of 2 x 2 target locations) and monetary rewards were purely based on perceptual accuracy (with reward magnitude being equal for all target locations). Hence, colour intentions in our task did not serve to transiently 'bias' perceptual choices, but instead, represented an independent decision variable that participants were explicitly instructed to take into account when deciding between the 4 choice targets. We have now addressed this important difference between previous studies and our current study in the introduction (page 5). Finally, in our study, we presented colour outcomes at the end of each trial and occasionally asked participants to rate the % of coloured dots or their Sense of Agency over colour outcomes. Thus, by design, colour intentions were directly relevant for participants throughout the whole duration of a trial. This presumably increased the persistence of colour intentions in our task compared to previous work studying the more subtle, implicit effects through which rewards can transiently bias perceptual choices.

12. The modelling could be much clearer if there are trial-averaged firing rates plotted.

Authors' reply: We thank the reviewer for this suggestion. We have now added the trial-averaged firing rates (see figure below) to Figure 6 (panel A) of the manuscript. Furthermore, we have added the following paragraph on page 26 of the manuscript:

“**Fig. 6A** shows trial-averaged neural firing rates of action nodes A_1 to A_4 for different trial types, time-locked to the onset of the first choice (i.e., first time the action threshold was crossed). Note that since CoM can occur at different points in time with respect to the first choice, averaging cancels out some of the fine-grained details of the second threshold crossing associated with CoM. For example, for CoM_{p+l} , action A_3 seems to stay below the action threshold on average. However, crucially, on a single trial classified as CoM_{p+l} , this action node - by definition - always crosses the threshold. [...]

Both the single-trial and averaged firing rates show evidence of competition between action nodes when a CoM occurs. This is particularly pronounced between A_1 and A_3 , thus illustrating the competition between whether to pursue the original colour intention, or instead, switch to the alternative colour.”

Figure 6. Changes of Mind in Attractor Network Model. A) Average firing rates of action nodes A_1 to A_4 in trials with no-CoM, CoM_P , CoM_{P+I} , and vertical CoM. Firing rates were locked to time of first threshold crossing (first choice). Only trials where the final choice was left-blue were included, except for CoM_{P+I} where a change with respect to the colour intention resulted in a final left-green choice.

13. Having the background/initial firing rate of the network at 10 Hz (line 923) may not be stable. Please demonstrate evidence that this is a stable steady state of the network.

Authors' reply: We thank the reviewer for raising this important issue. In order to demonstrate that the network has a stable steady state, we ran simulations where all inputs into the sensory nodes, intention nodes and cost nodes were set to 0. We set initial firing rates of all nodes to 10 Hz as in the original model, and all connections between nodes were kept at the same weights. We ran 30 simulations with 1000 trials each, with each trial lasting 1 sec. As expected, removing model inputs caused neural firing rates to fluctuate randomly around a stable steady state (see left panel in figure below). However, the average firing rate was slightly lower than the initial value of 10 Hz due to a slight dominance of inhibitory over excitatory connections (average firing rate of action nodes A_1 to A_4 : $M = 7.73$ Hz; see right panel in figure below).

A) Single-trial simulation

B) Average firing rates

Figure illustrating steady state of network (for review purposes only). Firing rates of 12 neural nodes in a model without external inputs into sensory, intention, or cost nodes. A) Single-trial firing rates. B) Average firing rates from 30 simulations with 1000 trials each [M (SD)]. Top row = sensory and intention nodes; middle row = cost nodes; bottom row = action nodes.

We acknowledge that our initial formulation in the manuscript was misleading: The 10 Hz served as an initial value that was assigned to each neural node, rather than representing the true steady-state background firing rate. We have now changed the sentence “All neurons started with a background firing rate of 10 Hz” to the following sentence in order to avoid misleading language (page 41): “Firing rates of all neural nodes were set to an initial starting value of 10 Hz at the beginning of each trial”.

Furthermore, we added the following sentence to the *Methods* section (page 44):

“Finally, we checked whether the network maintained a stable steady state in the absence of external inputs. We ran additional simulations where all external model inputs were set to 0. These simulations confirmed that the network maintained a stable steady state with below-threshold background activity (average firing rates of action nodes A_1 to A_4 : $M = 7.73$ Hz).”

14. Why are the SoA results not modelled?

Authors' reply: In our study, we were primarily interested in the processes producing reversals in decisions about exogenous and/or endogenous sources of evidence. We then tested the hypothesis that SoA judgments are directly affected by the processes underlying such CoM (e.g., a reversal in the initial endogenous intention may reduce SoA). However, this did not seem to be the case. Instead, SoA seemed to largely depend on the % colour outcome and movement times, but was not modulated by exogenous/endogenous decision reversals per se. We thus do not believe that including SoA in our model would be relevant, in particular, given the additional complexity this would introduce. While it would in theory be possible to add an additional neural layer to the network that could predict SoA judgments based on movement times and action outcomes (% coloured dots), this would not be directly linked to the decision-making processes we aimed to model with the attractor network. In order to clarify this aspect for the reader, we have now added the following sentence to the manuscript on page 25:

“[...] SoA results were not included in our computational model, given that our model focuses on the cognitive processes driving CoM about endogenous/exogenous decisions. Yet, our experimental results indicated that SoA was not directly related to these decision-making processes, but instead, largely depended on % action outcomes and movement times regardless of CoM.”

15. No testable model predictions has been provided. Please provide them.

Authors' reply: Please note that we provided testable model predictions in the earlier version of our manuscript (page 31): “In this context, the model makes further predictions that can be directly tested in future studies. For example, it is plausible that firing rates of action nodes (i.e., the strength of decision evidence in favour of an action) are directly linked to more fine-grained, gradually-varying details of motor policies, such as movement speed or vigour, rather than mere categorical choices between action alternatives.”

16. The Abstract and Introduction need to be written more clearly and concisely. For example, in the Abstract the task is not described clearly - what makes it novel(?). I have to read the main text to understand it. Also, there is no mentioning of the SoA results in the Abstract. In the Introduction, previous works seem to be stated for sake of stating without saying anything about their issues or bottlenecks facing them. Particularly, the motivation of the work needs to be stated more clearly. The Introduction reads a bit too long and there are abrupt jumps from one paragraph to another.

Authors' reply: We thank the reviewer for pointing out the lack of clarity in the abstract and introduction. We have now made substantial changes to both sections. Briefly, we edited the text in order to 1) state more clearly what the gap in the existing literature is, 2) describe how we fill that gap, 3) explain the task in a clearer, more straightforward way, and 4) motivate our model in more detail by adding more background information on previous models and their limitations. We have also made small changes to the structure of the introduction in order to improve the flow. We have intentionally left out the Sense of Agency results in the abstract due to word limitations, and given that SoA results only make up a small part of our manuscript compared to the main CoM findings and our computational model. However, if the reviewer feels strongly that SoA results should be added to the abstract, we are happy to make further changes to the abstract.

Minor/specific comments and suggestions:

1. *The notion of referring the term "Changes of Movement" to lower level sensorimotor action selection is not very clear, and may even be problematic, as internal intentions can also be reflected in the movement changes. There should be clearer definitions. Perhaps I am missing something here.*

Authors' reply: We thank the reviewer for pointing out the lack of clarity. We have now changed the terms for the different types of CoM to 'perceptual CoM' (CoM_P) and 'perceptual + intentional CoM' (CoM_{P+I}). We have further added a glossary of these terms and their acronyms at the beginning of the manuscript to further facilitate understanding.

2. *Line 65. "... circumstances." Please provide citation.*

Authors' reply: Please note that we have edited the introduction and have removed this sentence since we agree that it was vague and lacking clarity. We have now rewritten the corresponding section in order to refer more specifically to previous studies and the questions that remain to be answered in the existing literature.

3. *Line 146. "25/50/75/100% of dots ..." Explain what this is for.*

Authors' reply: We have now added a clarification on pages 18–19 (see our previous reply to comment # above). The differences in % outcome were introduced to increase relevance of colour choices with respect to the action outcome, and to introduce variability in SoA judgments about these action outcomes (see our reply to comment #3 above). We have clarified this aspect in the revised version of this manuscript.

4. *Lines 178-179. Are the RTs for CoM or non-CoM trials?*

Authors' reply: We thank the reviewer for raising this question. RT analyses reported in the manuscript include all correct trials, regardless of whether CoM occurred or not. As reported above, in our study, we did not find any differences in RTs between CoM and no-CoM trials. Hence, our results are not affected by whether or not CoM trials are included in the RT analyses. We have now added the following clarification in the manuscript (page 37): "No differences in RTs were found between trials with vs. without CoM (see Supplementary Material S3). Hence, RT analyses included all correct trials regardless of whether or not CoM occurred."

5. *Fig. 2C. Please state the correlation coefficient value. Also, why are there negative RT costs, i.e. easy trials take longer than conflict trials? The model does not seem to be able to exhibit this (Fig. 6C)?*

Authors' reply: As reported on lines 297-298 of the original manuscript, the correlation coefficient for this analysis is "Spearman's $\rho(15) = -.50, p = .043, 95\% \text{ CI } [-.07, -.76]$ " (p. 12). Given the relatively small sample size for a correlation analysis and the presence of outliers, we reported the

Spearman's rho correlation coefficient for this analysis since it is more robust than Pearson's r (de Winter, Gosling, & Potter, 2016). We have now additionally included the correlation coefficient to the figure caption of Fig. 2C.

There are indeed two participants with negative RT costs indicating faster RTs in conflict than test trials. There are several potential reasons for this:

1. It may simply be due to noise. Note that 1 of the 2 participants has an RT cost only slightly below 0 (-6.4 ms), which could be caused by measurement error or noise.
2. It may be that those participants were 'lucky' - that is, by chance, their colour intentions may have matched the dot-motion direction in more than 50% of conflict trials. Note that motion coherence was higher in conflict than test trials. Thus, if colour intentions matched the dot motion direction on the majority of conflict trials, RTs would be faster given the absence of conflict and lower difficulty of the perceptual decision.
3. These participants may have been particularly good at 'disengaging' from, or inhibiting, their voluntary colour intention when being presented with conflicting sensory information (e.g., better cognitive control or task-switching ability; Miyake et al., 2000). Thus, RT costs would be reduced in these participants, and may in fact be negative given the higher motion coherence in conflict compared to test trials (see 2).

Most importantly, the majority of participants (15/17 participants) showed positive RT costs, and hence, on average, RTs were significantly slower in conflict than test trials. Given that our model was optimized to capture *average* performance, it thus did not produce negative RT costs - at least not on average across 30,000 simulated trials. In that context, it is important to note that simulations were conducted with large trial numbers specifically with the aim to average out any potential effects of noise or chance, and hence, reasons 1) and 2) mentioned above would not be of relevance within the context of our model. Reason 3) raises an interesting question about how interindividual differences in cognitive control may affect people's ability to suppress their own intentions when facing conflicting external evidence. However, this is beyond the scope of our current study.

6. *Line 217. State the value of the extra motor costs.*

Authors' reply: Please note that the precise value of the extra motor cost depends on *when* during the movement CoM occurs (i.e., the distance already travelled towards the initial choice target, and thus, the current distance from the alternative choice targets). This is shown in Fig. 3B, illustrating how motor costs for each alternative choice target depend on the initially travelled distance. We have now included an additional reference to Fig. 3B in the manuscript section of Exp. 1 (pages 16 and 41) in order to clarify this aspect.

7. *Lines 385-386. Would this results be consistent with work such as Burk et al., PLoS One 2014, and more recently, Zgonnikov et al., Judgment & Decision Making, 2019?*

Authors' reply: We thank the reviewer for pointing this out. Indeed, some previous studies have shown an overall decrease in perceptual CoM when motor costs associated with CoM were high compared to when they were low (Burk et al., 2014; Moher & Song, 2014; Zgonnikov, Atiya, O'Hora, Rañó, & Wong-Lin, 2019). By contrast, in our study, we did not find such an overall decrease in CoM

in far vs. close targets, but instead, found a shift in the *type* of CoM that occurred. We have now included the following in the discussion section to address this (page 21):
“Interestingly, in contrast to previous studies (Burk et al., 2014; Moher & Song, 2014; Zgonnikov et al., 2019), we did not observe an overall increase in perceptual CoM in close compared to far targets. Hence, in our study, motor costs did not affect whether or not participants changed an ongoing action. However, motor costs did influence which aspects of action selection were changed (higher-order goals vs. lower-level sensorimotor decisions). It is possible that in the current study, participants were willing to correct their perceptual choices regardless of motor costs, given that they obtained additional monetary rewards for correct perceptual choices. By contrast, voluntary decisions were not associated with monetary incentives, and hence, differences in motor costs may have had a stronger impact on intention reversals than perceptual CoM per se.”

8. *Line 591. How is the Yan et al. 2016 work used in this current work?*

Authors’ reply: In our introduction, the paper by Yan et al. (2016) is cited as an example of an attractor network model in the context of decision-making and Changes of Mind (among other examples, such as Albantakis & Deco, 2011; Albantakis et al., 2012; Atiya et al., 2019; Atiya et al., 2020). Our approach is similar to Yan et al. in that we used a mean-field approach to approximate the neural activity of populations of neurons (based on the original mean-field model proposed by Wong & Wang, 2006). However, in order to clarify that we have used a simplified version of this approach based on Miller (2016), we have now removed the reference to Yan et al. (2016) from the *Methods* section. Instead, we state the following (page 39): “In order to compute the firing rates of each neural node over time, a simplified version of the mean-field approach introduced by Wong and Wang (2006) was used (see Miller, 2016).”

9. *Lines 705-710. There is description of higher level or top-down cognitive processing/control on lower sensorimotor information. Perhaps Atiya et al.'s works could also be discussed here?*

Authors’ reply: We thank the reviewer for this comment. We have now added a reference to the work by Atiya et al. in that section (p. 31):
“Finally, the current model implemented a hierarchical organisation. While previous work has largely focused on higher-order mechanisms related to meta-cognitive processes, e.g., uncertainty about sensory information (Atiya et al., 2019; Atiya et al., 2020), our model introduces top-down control through endogenous intentions that are independent from the sensory information itself. That is, we propose that abstract action goals that are generated internally can affect processing of external information on a lower, sensorimotor level. In line with this, previous studies have shown that within the frontal cortex, more anterior regions representing abstract information (e.g., goals) exert top-down control over more posterior regions involved in lower-level sensorimotor control (O’Reilly, 2010; Badre & Nee, 2018). Moreover, our current model proposes that noise control plays a crucial role in such top-down control. It has previously been suggested that higher-order areas exert top-down control by ‘gating’ inputs/outputs of lower-level areas (Badre & Nee, 2018). The noise reduction mechanisms implemented in the current model may be fundamental to such gating of information: That is, noise reduction can enhance action-relevant information, thus allowing voluntary actions to be shielded from noisy sensory distractions (Kilintari et al., 2018).”

10. *Why the return trajectories in CoMov trials typically avoid the initial starting point? Wouldn't such deviations take longer and are not as optimal as moving straight back through the initial starting point on their way to the other choice target?*

Authors' reply: Yes, such corrections are more costly in terms of path length, however, they are less costly in terms of motor control/biomechanics. Note that moving straight back through the initial starting point would require fully stopping the initial movement first, after which a new movement in the exact opposite direction has to be initiated. Such deceleration-acceleration patterns are not typical of natural reaching movements. Instead, people tend to initiate fast and smooth movements towards a general target direction, decelerating only once they approach the final target location in order to increase precision (Phillips & Triggs, 2001; Todorov, 2004; Hsieh, Liu, & Newell, 2017). Thus, the curved trajectories in perceptual CoM (previously referred to as 'CoMov') are presumably caused by a tendency to maintain the initial movement speed while adjusting the general direction of the movement to the new target location. The movement will only be decelerated once it approaches the final target location. In this sense, CoM in our study resembles the rapid adjustments to trajectory direction that occur when the target of a reaching movement is unexpectedly shifted (Goodale, Pélisson, & Prablanc, 1986). In line with our observations, curved trajectories for diagonal Changes of Mind have previously been reported by van den Berg et al. using a similar 4-target arrangement (van den Berg et al., 2016).

Additionally, note that when CoM occurs in our task, there is a strong competition between action nodes, in particular between the nodes corresponding to the decisions about whether to pursue the initial colour intention vs. switching to the alternative colour (see firing rates of action nodes A_1 and A_3 in Fig. 6A-B). Thus, competition between different choice alternatives may further contribute to the curvature in movement corrections observed in our task.

11. *The (spatial) quantification/measure for motor costs should be made clearly right up front. Initially I thought it was based on movement/reaction times. But the motor costs based on time generally seem to be generally small (< 100 ms).*

Authors' reply: Yes indeed, motor costs are defined based on spatial distance from each target. Note that this induces costs both in terms of effort and in terms of movement times (movement times for perceptual CoM are ~114 ms longer than movement times for perceptual + intentional CoM; linear mixed-effects model: $\chi(1) = 21.1, p < .05$). In our study, we cannot dissociate whether temporal or effort costs contributed more strongly to cost-induced changes in CoM and we therefore intentionally avoided referring to motor costs as effort vs. temporal costs, and instead, used motor costs as a general term to reflect the current distance from each target. As mentioned in an earlier response above, we now refer to Fig. 3B earlier on during the manuscript in order to emphasize the fact that motor costs were quantified based on the spatial distance to each target. We have further added the following sentences:

“[...] to approximate motor costs by calculating the Euclidean distance between the current cursor location and each of the 4 targets. As cursor position changes, motor costs associated with each target change according to the relative distance to each target at a given point in time (see Fig. 3B).” (page 41).

“[...] longer travel distances incur higher motor costs due to higher effort and/or longer movement duration” (page 16).

12. Table S3.2. *Why is there still quite a big difference in reaction times between model and experiment? Can this be optimised further? What is meant by "double CoM"? Changing decisions twice?*

Authors' reply: RTs predicted by the model are indeed 75.2 ms slower than RTs observed in the data. With our current model, this was the best fit we obtained. Note that the model was optimized based on a cost function that included a combination of 9 different outcome variables (see table S3.2). The cost for each outcome variable was represented in terms of a *relative* difference between observed and predicted value. For example, the 75.2 ms difference in RTs represents a 13.2% deviation from the actual observed value. Defining costs based on relative error was done in order to ensure that all outcome variables had the same scale in the cost function. However, it means that the *absolute* deviations from observed values are larger for variables with larger means (such as RTs), whereas they appear more negligible for variables with smaller means (e.g., CoM). Finally, while it may be possible to obtain better fits with additional model parameters (e.g., changes in decision threshold), this would come at the cost of higher model complexity, and a potential risk of overfitting the data. Note that we were fitting data from Exp. 1 to predict the effects of target distance observed in Exp. 2. Hence, for our purposes, generalizability of the model across different experiments (and subjects) was more important than minimizing the error to each individual outcome variable. Regarding double-CoM: We have now added a brief clarification in the main manuscript, explaining that "...double CoM (i.e., trials in which participants changed their mind more than once in a single trial) were excluded from all analyses" (p. 36).

13. Fig. S4.1, bottom. *Please show samples of the various CoM simulated trials here, to be consistent with the above panels.*

Authors' reply: We thank the reviewer for this suggestion. We have now added additional figures to the Supplementary Material illustrating examples of single-trial simulations for each type of CoM (Fig. S5.2-S5.4).

14. *In conflict trials, given that the discrimination of motion coherence takes precedence over the initially selected colour (intention), any "change of intention" may not be a true voluntary change in intention. Should this still be considered a change of intention? Perhaps this should be discussed maybe in the Discussion section?*

Authors' reply: Please note that any trials interpreted as "Changes of Intention" were only measured in test trials. In conflict trials, we did not interpret movement corrections in the same way as in test trials since, as the reviewer correctly notes, these trial types were very different. Indeed, we refer to movement changes in conflict trials as "diagonal and horizontal movement corrections of partial errors induced by mismatches between colour intentions and dot-motion direction" (caption Fig. 2). We chose these terms in order to disambiguate these corrections in conflict trials from the Changes of Mind observed in test trials. Additionally, we explained on pages 10-11 how movement corrections in conflict trials were interpreted differently from CoM observed in test trials. We have now added some additional clarification stating explicitly that "CoM in conflict trials were not interpreted in the same way as CoM in test/easy trials, given the differences in target configuration and instructions" and that the "main analyses of CoM focused on test trials only."

Reviewer #2 (Remarks to the Author):

The manuscript “Two ways to change your mind: Effects of intentional strength and motor costs on changes of intention” presents behavioural experiments and a model aimed at investigating changes of mind based on external and internal information. In our opinion, this manuscript would be a great fit for Nature Communications. We find the data convincing, the model compelling, and think the paper is a good example of high-quality science. The authors anticipated several concerns about their complex experiment, and address each of these concerns with appropriate analyses. We especially appreciate the authors making code publicly available (in both Matlab and Python too!). Overall, we are very happy to recommend it for publication.

Before publication, we think two major concerns listed below need to be addressed. Minor concerns listed below are merely suggestions, mainly about how to make the manuscript more accessible to a wider audience.

Authors’ reply: We thank the reviewers for their positive evaluation of our manuscript and appreciate their constructive feedback on how to further improve the paper. We will address each concern raised by the reviewers in the comments below.

Please note that in response to comments raised by both reviewers 1 and 2, we have now changed the terms for different types of Changes of Mind (CoM) in our manuscript. We now refer to CoM that only involves a perceptual change (without change in colour intention) as ‘perceptual CoM’ (CoM_P) instead of ‘Change of Movement’ (CoMov) in our previous version. CoM that involve both a change with respect to the perceptual decision *and* colour intention are now referred to as ‘perceptual + intentional CoM’ (CoM_{P+I}) instead of ‘Change of Movement + Intention’ (CoMov+Int) in our previous version. We have replaced these terms throughout our manuscript and also use the new terms in our replies to the reviewers’ comments below. We have furthermore added a glossary of these terms and their abbreviations, together with brief definitions, at the beginning of the manuscript in order to facilitate understanding. Finally, we have changed the title of our manuscript to “Alternative Ways to Change Your Mind: A Hierarchical Attractor Network Model of perceptual vs. intentional decision updates.”

Major Concerns:

1) We think that the authors are overinterpreting CoMov+Int and under-interpreting vertical changes of mind. First, we think the authors did a good job of convincing us that participants in this experiment actually cared about color, at least some of the time. They show that despite a higher motor cost, participants generally expend more effort to stick with their chosen color rather than change colors (even though participants are being paid for left/right choices and not color).

Second, participants sometimes changed color choices on the same side of space (i.e., vertical changes; ~3% of all trials in experiment 1). In our reading, the authors interpret these changes as errors - trials where participants were simply confused about where their preferred color was (footnote page 12). The authors argue (top of Page 13) that since participants made more vertical color changes on late-target-onset trials, then this suggests that vertical changes are due to difficulties in target detection. We disagree and think that most changes of mind (about dot-motion,

color, etc.) would increase under less preparation time. Further in the footnote of Page 12, the authors argue that "there was no reason for participants to change their mind about the target colour unless they had changed their mind about the dot-motion direction". We disagree here as well. There was no reason for participants to care about color at all, and yet it seems in this experiment that they did. In

fact, we find vertical color changes in these data very interesting and compelling. On these trials, there is absolutely no reason for participants to change their movement from one color to another on the same side - it doesn't impact their monetary outcomes, and it costs energy. Yet participants do this anyway. We think that these trials reflect some combination of color errors due to noise, but also genuine changes in color preference online.

Third, participants sometimes changed side of space (i.e., their choice about dot motion), and also changed color (i.e., CoMov+Int). In our reading, the authors interpret these trials as genuine changes of mind about color preference. We find this confusing. If dot-motion and color evidence are independent (as is discussed in the paper, implemented in the model, and as we assume as well), then how can genuine changes of color preference only happen when dot-motion changes happen? There are even descriptively more solely color changes (vertical; 3.2%) than combinations of dot-motion and color changes (CoMov+Int; 1.7%). We also interpreted the authors as arguing that in CoMov+Int trials, participants cared about colour and genuinely changed their color preference. We think that, in contrast to vertical trials, participants did have a reason to make a horizontal change - less energy cost. In this case, if a participant changed their mind about dot-motion, but wanted to conserve energy, they could make a horizontal change. In this case, participants did not need to think about color at all on these trials. In this way, we think the authors are not holding vertical changes and CoMov+Int to the same logical standard.

Overall, our interpretation is that there are actually three interesting kinds of changes of mind in these experiments - those solely about dot-motion, those solely about color, and those regarding both. We think that any of these changes are going to be made up of genuine changes based on evidence, system noise, and errors. We strongly recommend reformatting the arguments of this manuscript to equally consider all types of changes of mind. In our opinion, these data are very interesting and the manuscript in its current form is underselling these results.

Authors' reply: We very much appreciate the reviewers' thoughtful comments on a potential additional type of Change of Mind: Vertical changes that occurred between targets of different colour on the same side of the screen. While we agree with the reviewers that these changes are interesting in principle, our task design was optimized to test our hypotheses regarding perceptual CoM that were induced by noise in the random-dot motion stimulus. These perceptual changes were accompanied by a switch to one of the 2 targets on the *other* side of the screen: Either the target with the same colour ('perceptual CoM' = CoM_p , previously referred to as CoMov) or the other colour ('perceptual + intentional CoM' = CoM_{p+I} , previously referred to as CoMov+Int). By contrast, vertical changes were not triggered by noise in the sensory stimulus, but instead, were largely associated with late target onset.

Given the strong link to target-onset time, we interpreted vertical changes as initial 'target selection errors' that were corrected later on during the movement. We used this wording to clearly differentiate these changes from CoM that were driven by noise in the dot-motion stimulus

(CoM_P/CoM_{P+I}). However, as the reviewers correctly point out, this interpretation does not preclude considering them ‘real’ Changes of Mind given that, in general, CoM is strongly linked to processes involved in performance monitoring and error correction (Yeung & Summerfield, 2012).

Furthermore, we agree with the reviewers that the distinction we made is somewhat artificial, and that decision-making and response selection/execution should be regarded as a continuum of processes that can all be subject to Changes of Mind. We thus changed our wording to refer to vertical changes as ‘motor CoM’ to reflect that these changes can indeed be considered a type of Change of Mind, albeit different from the other two types of CoM that were induced by noise in the random-dot motion stimulus.

We further changed the manuscript to acknowledge a second possible interpretation of these changes as ‘real’ Changes of Intention (in the absence of a perceptual CoM). As the reviewers rightly point out, we had not sufficiently addressed this interpretation in our initial version of the manuscript. However, while it is possible that some vertical changes reflect CoM about an initial colour choice, we remain sceptical that this is true for the majority of cases due to the following reasons:

1. As the reviewers point out, we showed that participants’ colour intentions were very strong overall. This caused changes of intention to be rare even after a perceptual CoM triggered a re-evaluation of the initial colour intention (i.e., deciding whether or not it was worth pursuing the intention after perceptual CoM). It is not clear to us why participants would re-evaluate their initial colour choices in the absence of such an externally-triggered action update. At least, purely endogenous colour changes should be *less* frequent than colour changes following a perceptual CoM. Thus, to us, the fact that vertical changes occurred *more* frequently than CoM_{P+I} indicates that the mechanisms underlying these changes are different and likely do not reflect genuine changes in participants’ colour intentions.
2. Purely endogenous colour changes should be particularly rare given that vertical changes incur a motor costs, but do not increase rewards. By contrast, CoM_{P+I} increases rewards by increasing perceptual accuracy and, at the same time, reduces motor costs relative to purely perceptual CoM (CoM_P). Thus, while no new information about the colour intention itself became available during CoM_{P+I}, its trade-off with motor costs changed when a perceptual CoM occurred (according to the current distance from each alternative target). As shown in Exp. 2 and our computational model, motor costs played an essential role in participants’ decision updates. Consequently, similarly to our reasoning in 1), we think it is unlikely that participants would change their colour intentions without a change in motor cost trade-offs, particularly given that initial colour intentions were strong and that changing the intention would incur an unnecessary motor cost.
3. Initial colour choices were not made under time pressure. That is, participants could take as much time as they wanted when choosing between the two colours at trial start, further reducing the likelihood of them changing their colour choices later on without an externally-induced need to re-evaluate the colour choice (see 1 and 2 above). By contrast, the time to prepare a movement associated with each colour choice was reduced when target onset was late compared to when it was early, thus increasing the likelihood of initial response errors where movements were initiated towards the wrong target colour.
4. If vertical changes reflected ‘real’ changes of intention, one would expect the frequency of these changes to be negatively correlated with RT costs in conflict trials, similarly to the negative correlation of CoM_{P+I} with RT costs in Exp. 1. That is, the stronger participants’ colour intentions are, the higher their performance costs in conflict trials would be, and the less likely they should be to change their mind about the initial colour intention. However, as opposed to CoM_{P+I}, we did not find any negative correlations between individuals’ RT costs in conflict trials and the frequency of vertical changes in test trials (Exp. 1: $r_s = .199$, $p = .445$; Exp. 2: $r_s = .238$, $p = .374$)

nor in easy trials (Exp. 1: $r_s = -.257$, $p = .319$; Exp. 2: $r_s = .359$, $p = .172$). Thus, while it is in theory possible that some vertical changes reflected true changes of intention, we do not have any evidence to suggest that this is true for the majority of vertical changes.

5. Instead, our model provided evidence in support of our interpretation of vertical changes as a type of lower-level ‘motor CoM’, which corrected initial target colour selection errors. First, our model confirmed that colour intentions were very strong overall, thus rendering CoM about colour intentions in the absence of perceptual CoM unlikely. Furthermore, we showed that 77.6% of vertical changes in our model were associated with an initial response that did not correspond to the ‘true’ intention of a given trial, followed by a correction of the response towards the correct target. By contrast, only 28.4% of CoM_{P+I} were associated with initial colour errors (see *Results* section page 28). This is in line with our interpretation of vertical movement corrections as ‘motor CoM’ that reflect corrections of initial response selection errors due to target uncertainty.

In conclusion, we agree with the reviewers that vertical changes can be interpreted as Changes of Mind. However, we believe that they largely reflect lower-level motor Changes of Mind, instead of changes about higher-order colour intentions given that they 1) were largely driven by late target onset, 2) were not correlated with participant’s strength of colour intentions, and 3) were instead associated with initial colour errors in our model. Nevertheless, we agree that we had not addressed these different possible interpretations of vertical changes in sufficient detail in the initial version of the manuscript.

We have now added a new section in our *Results* (section 2.2.4 ‘*Vertical movement corrections*’; pages 13–14) where we address possible interpretations of vertical changes in more detail and briefly summarise the arguments laid out above. Additionally, we have changed the title of our manuscript from “Two Ways to Change Your Mind” to “Alternative Ways to Change Your Mind” to reflect that our task may capture more than just 2 different types of CoM. Finally, we added a brief section in our discussion (pages 29–30) where we address the possibility of capturing purely endogenous CoM about voluntary intentions (in the absence of perceptual CoM):

“Note that our study focused on intention reversals that were triggered by a CoM about external, perceptual information, which subsequently caused a re-evaluation of the initial intention based on its trade-off with motor costs. Even though we observed some changes between colour targets in the absence of perceptual CoM (vertical movement corrections), these changes appeared to largely reflect lower-level motor CoM that were caused by initial colour errors due to target uncertainty, instead of reflecting genuine changes in higher-order colour intentions. An intriguing possibility for future studies may be to capture intentional CoM that are completely independent of any external changes, in order to further elucidate the processes through which endogenous decisions are updated continuously.”

2) *As we read it, dot motion sensory evidence in the model is represented as fixed inputs over time on a single-trial basis (as in the top left panel of Figure S4.1). However, motion evidence on a single-trial level in the random dot motion task is quite variable from timepoint-to-timepoint and trial-to-trial, especially at low dot motion coherences like in the current manuscript. As-is the model in the manuscript implies that perceptual changes are driven solely by internal neural noise, rather than a combination of neural noise and external changes in the stimulus. However, we know that moment-to-moment fluctuations in dot motion energy influence choices and changes of mind. Resulaj et al., 2009*

Figure 3a has a great depiction of average motion energy contributing to all trials vs. those when a Change of Mind occurred.

We highly recommend either addressing this simplification/limitation in the manuscript, or alternatively updating the model to account for noisy stimulus input (structured in a way consistent with noise in the dot motion stimulus). We know of two great resources for calculating motion energy given frames of dot motion stimuli: (1) https://github.com/KianiLab/Waskom_JVision_2018, and (2) <https://github.com/arielzylberberg/MotionEnergy>

We see in the Matlab and Python code provided, that the authors have normally-distributed noise for color and dot motion inputs commented out (e.g., Line 159 in ANM.py), and are interested as to why this is the case.

In contrast, we think that modelling color preference inputs as fixed over time is a reasonable assumption. Although related to point #1, this assumption would mean that any changes of color are errors, or correcting for initial errors. In our opinion, changes of dot-motion are especially interesting because, due to the noisy stimulus, changes can also go from initially-correct to also correct options (in the case of dot-motion in opposing directions at the beginning and end of a trial) - which speaks to our ability to adapt movements online to changes in the world.

Authors' reply: We thank the reviewers for this comment. We agree that assuming constant sensory inputs in our model is a simplification, given the moment-to-moment fluctuations in the random-dot motion stimulus that have been shown to drive Changes of Mind (Resulaj et al., 2009; Albantakis & Deco, 2011). Indeed, we had initially considered adding noise to the sensory inputs, either in the form of random Gaussian noise, or noise whose structure corresponds to the motion energy fluctuations of the random-dot stimuli on a within-trial basis. However, after further consideration, we decided to treat the stimulus as a constant input because our model was optimized to fit participants' *average* behaviour - both within and across participants. Thus, momentary fluctuations in sensory inputs on a single-trial level can be disregarded given that they will average out across trials and participants. Instead, we can approximate sensory inputs as a static value according to the constant motion coherence level in test trials. That is, in our model, we capture how decision-making is affected by sensory evidence as an average signal instead of considering dynamic intra-trial changes in the stimulus.

While it would in principle be possible to account for momentary fluctuations in evidence by fitting the model on a single-trial level, we did not attempt this given the overall low number of trials with CoM and the complexity of our model as it is (12 interconnected neural nodes with hierarchical organisation). Furthermore, using motion energy to simulate stimulus noise is problematic in our case since we used a reaction-time task in which the stimulus disappeared at response onset. Thus, the amount of available motion energy information from each presented stimulus is limited by participants' response times. A more appropriate design for an analysis like this would be to use a fixed stimulus duration.

As an alternative to using motion energy to model stimulus noise, we could add random Gaussian noise to the stimulus input. However, we believe that this would be redundant with the (Gaussian) neural noise in the network. In line with this, previous models have not typically included separate sources of external vs. internal noise (). Moreover, in contrast to what the reviewers propose, we believe that if we considered sensory noise, we would similarly have to model noise in the other input

sources. This is particularly true for cost inputs given that motor costs depend on (noisy) movement trajectories and (noisy) estimates of one's current distance to each alternative target. Similarly, it would be reasonable to assume noise in endogenous signals (such as memory signals; e.g., Shadlen & Shohamy, 2016), which provide moment-by-moment evidence about the colour intention. If we assumed different degrees of noise for each source of input (sensory, cost, intentional), we would need to include three additional parameters in the model. However, we believe that this would increase model complexity without increasing explanatory power given that the different sources of noise would simply increase overall variability in model (which is currently already accounted for by just a single neural noise parameter).

Based on the reasons mentioned above, we believe that assuming constant sensory input is a reasonable simplification of our model. We have now addressed this in our discussion section on pages 31–32:

“One limitation of the current model is that CoM is largely driven by neural noise within the network, while noise in the input sources (e.g., sensory evidence) is disregarded. Yet, previous studies have shown that, in addition to the (constant) coherence level of a trial, moment-by-moment fluctuations of sensory evidence affect decision making and can drive CoM (Resulaj et al., 2009; Albantakis & Deco, 2011). Our model does not account for such within-trial fluctuations in the sensory stimulus, given that it was optimized to fit participants' average behaviour across trials, and thus, momentary fluctuations in evidence will average out. Similarly, we assumed that inputs into intention nodes were constant. However, it is possible that intentional strength fluctuates both within and across trials. Finally, cost inputs may be noisy given that they rely on (noisy) estimates of one's current distance from each alternative target. Future studies might extend the current model by accounting for variability in each model input both within and across trials, and thus, consider alternative sources of noise in addition to neural noise.“

Minor Concerns:

1) *Figure 3C caption reads "Predicted effect [...]". Please clarify if this is real participant data from Experiment 2.*

Authors' reply: We thank the reviewers for pointing out the error in the figure caption. Figure 3C indeed shows real participant data from Experiment 2, instead of a predicted effect. We have now updated the figure caption accordingly.

2) *This manuscript had many abbreviations compared to other papers, which made it difficult to read. For example, the first paragraph of Page 19 referenced CoM, MT, SoA, CoMov, CoMov+Int, etc. We feel reasonably familiar with these topics and still found these passages hard to read. We are worried other readers outside of this specific area may have an even harder time deciphering these passages. Any effort to spell out some of these words, or substitute language, especially in these dense sections, will help improve the accessibility and impact of this work in our opinion.*

Authors' reply: We thank the reviewers for their helpful suggestion on how to make the manuscript more accessible. We have now spelled out these terms in some sections or rephrased our wording

where possible to reduce the number of abbreviations. We have further removed all instances of the abbreviations RDM (random-dot motion), ANM (attractor network model), and MELR (mixed-effects logistic regression) by spelling them out or changing our wording. We have kept the abbreviations for Sense of Agency (SoA) and Change of Mind (CoM). However, the terms CoMov and CoMov+Int have been replaced with CoM_P (perceptual CoM) or CoM_{P+I} (perceptual + intentional CoM). Finally, we have now added a glossary with important terms and acronyms, including brief descriptions of each of them, at the beginning of our manuscript.

3) *On Page 9, Line 209, the authors mention MELR, which is not defined until the Methods section.*

Authors' reply: Thank you for pointing this out. Please note that we have now spelled out the 4 instances of 'MELR' with 'mixed-effects logistic regression' in order to further reduce the number of abbreviations in the manuscript.

4) *Early vs. late target onset is first referenced in the results in an analysis. This factor is discussed in the full methods appearing at the end of the manuscript, but it would be helpful to mention this factor in the abbreviated methods within the main text.*

Authors' reply: We thank the reviewer for pointing this out. We have now added the following sentence in the abbreviated methods section in the main text: "Targets appeared either 700–1000 ms before the dot-motion onset (*early targets*; 50% of trials), or at the same time as the dot-motion stimulus (*late targets*; 50% of trials)."

5) *Page 4, Lines 77 onward, defined Changes of Movement as involving changes of "motor intentions". Then the authors immediately go on to talk about the independent concept "Changes of Intention". We understand that these are technical terms, but the intermixing of words made this argument difficult to follow. Perhaps changing the names of these terms might help? For example, Changes of Movement is confusing because participants need to change their movement to reflect either a new goal about dot motion, about color choice, or both. Perhaps terms like a perceptually-based change of mind and a internally-based change of mind.*

Authors' reply: We thank the reviewers for this suggestion. We have now replaced the terms 'Change of Movement' and 'Change of Movement+Intention' with 'perceptual CoM' (CoM_P) and 'perceptual + intentional CoM' (CoM_{P+I}), respectively.

6) *Labels for bar colors in figure 3C would be appreciated.*

Authors' reply: We apologize for the missing labels in the original version of the manuscript. We have now added the labels for the different types of CoM to Figure 3C.

7) *On the first paragraph of Page 17, We find the followup analysis of an interaction effect with a p-value of 0.097 unnecessary.*

Authors' reply: We thank the reviewers for pointing this out. We have now removed the follow-up analysis of the interaction effect.

8) *Selfishly, we recently published a very relevant review on this topic that might make a nice addition as a citation:*

Wispirski, N. J., Gallivan, J. P., & Chapman, C. S. (2018). Models, movements, and minds: bridging the gap between decision making and action. Annals of the New York Academy of Sciences.

...since this experiment does a great job of exactly filling the void we propose exists, it would be great to link these two papers!

Craig S. Chapman & Nathan Wispirski

Authors' reply: We thank the reviewers for this suggestion. We agree that this review paper is an excellent contribution to this area of research and is highly relevant for our current paper. Specifically, in their review, the authors argue that decision-making and movement execution should be viewed as a continuum of distributed processes that evolve dynamically over time. This is an essential feature of the computational model we propose in our current study. We have therefore added references to this review paper in the relevant parts of the manuscript, mainly in the section describing the theoretical background for our model. We have further edited the following section in the discussion:

“Additionally, in contrast to previous models of CoM, the current model explicitly allowed for an active role of action representations during the evolving decision-making process. That is, action nodes in the current model were not simply a mere ‘output’ system of higher-order decision-making areas, but instead, played an essential role in determining the final action outcome by integrating various sources of decision-relevant information to guide action selection in a gradual and continuous manner. Thus, instead of a serial, feedforward hierarchy from decision to action, our model proposes an interactive hierarchy, in which information related to action can itself feed back to modify the decision. This is in line with recent theories that view decisions and movements as highly integrated processes, which evolve continuously and gradually over time, instead of representing strictly serial and segregated processes (Cisek, 2007, 2012; Rushworth, Kolling, Sallet, & Mars, 2012; Yoo & Hayden, 2018; for a recent review, see Wispirski, Gallivan, & Chapman, 2020).”

Reviewer #3 (Remarks to the Author):

The authors aimed to examine how internal intentional and external environmental factors influence the change of mind and to provide an attractor network model as a tool to explain observed outcomes.

I found that the attempt to combine internal and external factors are interesting. However, I am not convinced that the outcomes they acquired can be generalized beyond this specific experimental paradigm. First of all, entire their arguments are depended on a couple of % differences in CoM frequency. Furthermore, their classification of CoM is much generous than previous studies and yet the overall occurrence itself is pretty low. It is unsurprising that internal conflict can be reflected in curved trajectories and combined with external factors (e.g. Dotan et al., 2019 TICs). Also it should reflect the strength of subject intention and importance. I think the authors have ignored accumulated over a decade of research using continuous trajectories and heavily relying on a particular study -- which happened to use the term CoM first, making it something new -- and build upon a complex argument. Therefore, I am not fully convinced by the novelty of claims and the generalizability/reproducibility beyond this specific task. It might have a better fit for much-specialized journal. I also found that the attractor network model is a reason to try. However, it is unclear why this model is much better than any other existing models.

Authors' reply: We thank the reviewer for providing feedback on our manuscript. Below, we address each concern raised by the reviewer in detail.

Please note that in response to comments raised by both reviewers 1 and 2, we have now changed the terms for different types of Changes of Mind (CoM) in our manuscript. We now refer to CoM that only involves a perceptual change (without change in colour intention) as 'perceptual CoM' (CoM_P) instead of 'Change of Movement' (CoMov) in our previous version. CoM that involve both a change with respect to the perceptual decision *and* colour intention are now referred to as 'perceptual + intentional CoM' (CoM_{P+I}) instead of 'Change of Movement + Intention' (CoMov+Int) in our previous version. We have replaced these terms throughout our manuscript and also use the new terms in our replies to the reviewers' comments below. We have furthermore added a glossary of these terms and their abbreviations, together with brief definitions, at the beginning of the manuscript in order to facilitate understanding. Finally, we have changed the title of our manuscript to "Alternative Ways to Change Your Mind: A Hierarchical Attractor Network Model of perceptual vs. intentional decision updates."

1. Concern regarding the lack of generalizability beyond our specific experimental paradigm

While our task involves a set of specific decisions about colour and random dot-motion stimuli, we believe that the mechanisms underlying these decisions are generalizable to a wide variety of other laboratory tasks, and indeed, to decisions people face in their daily lives. Specifically, our task was designed to capture scenarios in which people need to integrate multiple sources of evidence to make a choice: 1) An internally generated intention about a higher-order goal of an action (e.g., "I want to eat dinner at a Japanese restaurant"), 2) a perceptual decision that informed how to achieve that goal through a specific motor action ("e.g., finding the restaurant on a map in order to know which way to go") and 3) motor costs that informed whether the goal was worth pursuing or not (e.g., "is it worth

walking for 20 min in order to go to the Japanese restaurant, or is there an alternative restaurant nearby?"). Decisions that require integration of multiple sources of information are ubiquitous in real life. Yet, previous decision-making paradigms have typically studied decisions that are based on a single source of evidence in isolation (e.g., purely sensory evidence in the original random-dot motion task). We thus believe that our paradigm is in fact *more* representative of real-life decisions and *more* generalizable to other types of decisions than previous choice paradigms because it is not limited to one specific type of decision evidence.

By necessity, we had to operationalize each decision component (endogenous, perceptual, motor costs) by introducing specific task features/stimuli, which were optimized for testing our hypotheses. For example, we chose random-dot motion stimuli for perceptual decisions because these types of stimuli are highly suitable for studying sensory evidence integration. However, this does not mean that our findings are restricted to these stimuli. In fact, there is no reason to assume that our findings would be different for other types of perceptual choices (e.g., orientation or brightness judgments instead of motion direction judgments) or other types of action goals (e.g., rewards or auditory action outcomes instead of visual action outcomes).

Finally, we provided evidence that our computational model is generalizable across different variants of our paradigm. Specifically, although our model was optimized to fit behaviour of participants in Exp. 1, it accurately predicted behavioural outcomes for a different set of participants in Exp. 2 where target distance varied on a trial-by-trial basis. While the differences between Exp. 1 and 2 were relatively small, our model is in principle sufficiently abstract and flexible to be adapted for vastly different decision scenarios. For example, the quantity and quality of decision inputs is not constrained, and thus, our model can easily be extended to include other sources of decision evidence. Moreover, the hierarchical structure of our model reflects mechanisms that are highly relevant for many types of decision scenarios in the lab as well as in real life. For example, the majority of laboratory decision-making paradigms are characterized by an overarching task goal (e.g., be as fast and accurate as possible) that guides choice behaviour on a single-trial level. We propose a neurobiologically plausible mechanism through which higher-order goals may 'gate' lower-level sensorimotor information. Our model thus provides a unifying framework for how multiple, hierarchically-organised sources of decision evidence are integrated dynamically over time. We therefore believe that both our findings and computational model are generalizable beyond our specific paradigm, and provide important insights into the fundamental mechanisms through which agents decide between multiple available action alternatives.

2. Concern regarding the low frequency of CoM

We acknowledge that the frequency of CoM is relatively low, and we had indeed addressed this concern in our manuscript on pages 14–15: “Although the overall frequency of CoM was relatively low, the observed 7.6% CoM in test trials is clearly comparable with previous studies reporting 2–15% CoM in trials with similar motion coherences (Resulaj et al., 2009; Moher & Song, 2014; van den Berg et al., 2016). Further, several areas of cognitive theory, e.g., memory research, rely strongly on data from infrequent errors – no doubt because errors are highly informative about the processes generating performance (Loftus, 2005). Finally, the frequency of CoM varied systematically across trial conditions. Specifically, in line with previous studies on perceptual decision reversals (e.g., Resulaj et al., 2009), we found that CoM was more frequent when sensory noise was high and when the initial perceptual decision was erroneous.”

In this context, it is important to add that all differences in CoM reported in our study were statistically significant. Even though the absolute differences may have been small due to an overall low occurrence of CoM, the reported effects were consistent across participants and across Exp. 1 and 2. In all our analyses, we used statistically robust methods, such as mixed-effects models that have been recommended for analysing data sets with unbalanced and missing data points (Bagiella, Sloan, & Heitjan, 2000).

Finally, in contrast to the reviewers' claim, we used classification criteria for CoM that are highly comparable to previous studies. For example, in the original study by Resulaj et al. (2009), trials were classified as CoM when the initial hand movement exceeded 1 cm along the x-direction towards one target (~11% of x-distance from target) and covered an area of at least 0.1cm² between the midline and a straight line to the target location, but then ended in a different target. In our study, movement trajectories had to exceed 10% of the total travel distance on *both* the x- and y-axis, ensuring that the initial movement was indeed directed towards a specific target location and had travelled a substantial distance towards that target. Note that we used a relative criterion for CoM classification (instead of absolute travel distance/area) to ensure that CoM classification was not biased by differences in target configuration or distance (see *Methods* for details). Importantly, CoM typically occurred much later than the applied classification criterion. On average, CoM occurred roughly halfway through the movement, i.e., when 44.99% ($SD = 7.73\%$) of the total distance towards the initial target had been completed. We have now added this information in our manuscript (page 36) to further clarify that CoM trials in our task do not simply reflect trials with small movement deviations (e.g., due to movement noise), but instead, involved a clear initial movement towards one target followed by a change to a different target.

3. Concern regarding the novelty of our approach and findings

Our approach of measuring movement trajectories to identify CoM is not novel in and of itself and we have acknowledged previous work using the same methodology throughout our manuscript (e.g., Resulaj et al., 2009; Albantakis & Deco, 2011; Albantakis et al., 2012; Burk et al., 2014; Moher & Song, 2014; van den Berg et al., 2016; Atiya et al., 2019; Atiya et al., 2020). We focused on citing the work that specifically investigated decision reversals in perceptual decision-making. While there are other studies that have investigated the curvature of movement trajectories, these have largely looked at movement reprogramming in double-step paradigms (Goodale et al., 1986) or processes related to response conflict, such as in Flanker, Stroop, or dual tasks (Scherbaum, Gottschalk, Dshemuchadse, & Fischer, 2015; Erb, Moher, Sobel, & Song, 2016; Scherbaum & Dshemuchadse, 2019). This was not the primary aim of our study. Note that in our study, conflict trials only made up 10% of trials and merely served as a control condition to check whether participants generated colour intentions at trial start at all. Our main conclusions however did not rely on conflict between intentions with sensory evidence, but instead, required a *combination*, i.e., continuous integration, of one's own voluntary intentions with external sensory information and motor costs. Dynamic updates with respect to each of these decision variables resulted in different types of CoM (perceptual and/or intentional CoM), which were dissociated based on movement trajectories.

To our knowledge none of the previous studies have measured CoM in a task in which *multiple* sources of evidence (higher-order intention, perceptual decision, and motor costs) had to be integrated dynamically over time. Instead, previous tasks captured CoM in decisions that only involved a single source of evidence - typically sensory evidence from a random dot-motion stimulus. Although some

previous studies investigated the effect of motor costs on CoM (Burk et al., 2014; Moher & Song, 2014; Zgonnikov et al., 2019), in these studies, costs were treated as a static variable that simply caused a shift in the decision threshold. This neglects the fact that motor costs vary dynamically on a within-trial basis, for example, according to the distance to each target based on the current hand position. In our study, we investigated multiple decision components that each contributed continuously and dynamically to the evolving decision. In addition to sensory evidence and motor costs, our study also involved a higher-order, endogenous decision component (colour choice), which was independent of any external, sensory evidence. This allowed us to capture a new type of CoM, which had been neglected by previous studies: Changes of Intention in which participants deviated from their own initial action goal. Investigating decision reversals about such higher-order action intentions is highly relevant because they can have wide-ranging personal and social implications, such as when people change their minds about food choices or exercise routines (Goschke, 2014).

We have now edited our introduction to further emphasize the novel aspects of our study, and point out the gaps in the existing literature more clearly. We have further added a reference to the recent publication by Dotan et al. (Dotan, Pinheiro-Chagas, Al Roumi, & Dehaene, 2019) in our introduction and discussion.

4. Concern regarding the advantages of our attractor network model over other existing models

As mentioned above, previous studies of CoM have only considered a single source of evidence in isolation (e.g., sensory evidence from dot-motion stimulus; Resulaj et al., 2009; Albantakis & Deco, 2011; Albantakis et al., 2012; Burk et al., 2014; Moher & Song, 2014; van den Berg et al., 2016; Fleming, Putten, & Daw, 2018; Atiya et al., 2019; Atiya et al., 2020). As a consequence, existing computational models of CoM have neglected the processes through which multiple sources of information are integrated dynamically over time, leading to different types of CoM (e.g., CoM regarding perceptual decision vs. CoM regarding one's own intention). In particular, existing drift diffusion models have largely been limited to integration of evidence from a single modality. By contrast, network models readily allow for multiple sources of evidence to be integrated through interconnectivity between multiple neural nodes (Lo & Wang, 2006; Cisek, 2012; Christopoulos et al., 2015). Yet, previous attractor network models of CoM have only considered decision reversals driven by a single source of (sensory) evidence (Albantakis & Deco, 2011; Albantakis et al., 2012; Yan et al., 2016). Our current model provides an extension of previous work by introducing a unified framework for different types of Changes of Mind in voluntary actions, which are guided by several pieces of hierarchically-organised endogenous and exogenous information (voluntary intention, sensory evidence, and motor costs).

We discussed this limitation of previous models both in our introduction (page 5) and discussion (pages 30–31). We further addressed the following advantages of our current model in the discussion section on page 30:

“Past computational accounts of CoM largely neglected such integrative processes, and instead, focused on decisions that are purely driven by a single source of (perceptual) evidence (e.g., Resulaj et al., 2009; Albantakis & Deco, 2011). The current model provides a novel extension of this work by introducing a unified framework for different types of Changes of Mind in voluntary actions, which are guided by several pieces of not just weighted decision variables, but of hierarchically-organised endogenous and exogenous information. Additionally, in contrast to previous models of CoM, the current model explicitly allowed for an active role of action representations during the evolving

decision-making process. That is, action nodes in the current model were not simply a mere ‘output’ system of higher-order decision-making areas, but instead, played an essential role in determining the final action outcome by integrating various sources of decision-relevant information to guide action selection in a gradual and continuous manner. Thus, instead of a serial, feedforward hierarchy from decision to action, our model proposes an interactive hierarchy, in which information related to action can itself feed back to modify the decision. This is in line with recent theories that view decisions and movements as highly integrated processes, which evolve continuously and gradually over time, instead of representing strictly serial and segregated processes (Cisek, 2007, 2012; Rushworth, Kolling, Sallet, & Mars, 2012; Yoo & Hayden, 2018; for a recent review, see Wispinski, Gallivan, & Chapman, 2020). Moreover, in line with previous studies, our model assumes that multiple action representations evolve in parallel and that action selection is determined through a winner-take-all competition between these multiple co-existing affordances (Cisek & Kalaska, 2005; Cisek, 2007; Pastor-Bernier & Cisek, 2011; Thura & Cisek, 2014). Finally, motor outcomes of the model (i.e., simulated movement trajectories) affected subsequent decision updates by causing dynamic changes in motor costs (i.e., changes in distance to each target). Thus, our model provides a common framework for decision making and action selection, and accounts for their reciprocal relation, instead of assuming strictly separate and serial processing of decisions and actions. In this context, the model makes further predictions that can be directly tested in future studies. For example, it is plausible that firing rates of action nodes (i.e., the strength of decision evidence in favour of an action) are directly linked to more fine-grained, gradually-varying details of motor policies, such as movement speed or vigour, rather than mere categorical choices between action alternatives.

Finally, the current model implemented a hierarchical organisation. While previous work has largely focused on higher-order mechanisms related to meta-cognitive processes, e.g., uncertainty about sensory information (Atiya et al., 2019; Atiya et al., 2020), our model introduces top-down control through endogenous intentions that are independent from the sensory information itself. That is, we propose that abstract action goals that are generated internally can affect processing of external information on a lower, sensorimotor level. In line with this, previous studies have shown that within the frontal cortex, more anterior regions representing abstract information (e.g., goals) exert top-down control over more posterior regions involved in lower-level sensorimotor control (O’Reilly, 2010; Badre & Nee, 2018). Moreover, our current model proposes that noise control plays a crucial role in such top-down control. It has previously been suggested that higher-order areas exert top-down control by ‘gating’ inputs/outputs of lower-level areas (Badre & Nee, 2018). The noise reduction mechanisms implemented in the current model may be fundamental to such gating of information: That is, noise reduction can enhance action-relevant information, thus allowing voluntary actions to be shielded from noisy sensory distractions (Kilintari et al., 2018)”

References for review

- Albantakis, L., Branzi, F. M., Costa, A., & Deco, G. (2012). A multiple-choice task with changes of mind. *PLoS one*, 7(8), e43131. doi:10.1371/journal.pone.0043131
- Albantakis, L., & Deco, G. (2011). Changes of mind in an attractor network of decision-making. *PLoS computational biology*, 7(6), e1002086.
- Atiya, N. A. A., Rañó, I., Prasad, G., & Wong-Lin, K. F. (2019). A neural circuit model of decision uncertainty and change-of-mind. *Nature communications*, 10(1), 2287.
- Atiya, N. A. A., Zgonnikov, A., O'Hora, D., Schoemann, M., Scherbaum, S., & Wong-Lin, K. F. (2020). Changes-of-mind in the absence of new post-decision evidence. *PLoS computational biology*, 16(2), e1007149.
- Badre, D., & Nee, D. E. (2018). Frontal cortex and the hierarchical control of behavior. *Trends in cognitive sciences*, 22(2), 170-188.
- Bagiella, E., Sloan, R. P., & Heitjan, D. F. (2000). Mixed-effects models in psychophysiology. *Psychophysiology*, 37(1), 13-20. doi:10.1111/1469-8986.3710013
- Bogacz, R., Brown, E., Moehlis, J., Holmes, P., & Cohen, J. D. (2006). The physics of optimal decision making: a formal analysis of models of performance in two-alternative forced-choice tasks. *Psychological review*, 113(4), 700.
- Burk, D., Ingram, J. N., Franklin, D. W., Shadlen, M. N., & Wolpert, D. M. (2014). Motor effort alters changes of mind in sensorimotor decision making. *PLoS one*, 9(3), e92681. doi:10.1371/journal.pone.0092681
- Chambon, V., Sidarus, N., & Haggard, P. (2014). From action intentions to action effects: how does the sense of agency come about? *Frontiers in human neuroscience*, 8, 320.
- Christopoulos, V., Bonaiuto, J., & Andersen, R. A. (2015). A biologically plausible computational theory for value integration and action selection in decisions with competing alternatives. *PLoS computational biology*, 11(3), e1004104.
- Churchland, M. M., Byron, M. Y., Cunningham, J. P., Sugrue, L. P., Cohen, M. R., Corrado, G. S., . . . Scott, B. B. (2010). Stimulus onset quenches neural variability: a widespread cortical phenomenon. *nature neuroscience*, 13(3), 369-378.
- Cisek, P. (2007). Cortical mechanisms of action selection: the affordance competition hypothesis. *Philos Trans R Soc Lond B Biol Sci*, 362(1485), 1585-1599. doi:10.1098/rstb.2007.2054
- Cisek, P. (2012). Making decisions through a distributed consensus. *Current opinion in neurobiology*, 22(6), 927-936. doi:10.1016/j.conb.2012.05.007
- Cisek, P., & Kalaska, J. F. (2005). Neural correlates of reaching decisions in dorsal premotor cortex: specification of multiple direction choices and final selection of action. *Neuron*, 45(5), 801-814.
- Cos, I., Bélanger, N., & Cisek, P. (2011). The influence of predicted arm biomechanics on decision making. *Journal of neurophysiology*, 105(6), 3022-3033. doi:10.1152/jn.00975.2010
- de Winter, J. C. F., Gosling, S. D., & Potter, J. (2016). Comparing the Pearson and Spearman correlation coefficients across distributions and sample sizes: A tutorial using simulations and empirical data. *Psychological methods*, 21(3), 273.
- Deserno, L., Boehme, R., Mathys, C. D., Katthagen, T., Kaminski, J., Stephan, K. E., . . . Schlagenhauf, F. (2020). Volatility estimates increase choice switching and relate to prefrontal activity in schizophrenia. *Biological Psychiatry: Cognitive Neuroscience and Neuroimaging*, 5(2), 173-183.
- Dotan, D., Pinheiro-Chagas, P., Al Roumi, F., & Dehaene, S. (2019). Track it to crack it: Dissecting processing stages with finger tracking. *Trends in cognitive sciences*, 23(12), 1058-1070.
- Durstewitz, D., & Seamans, J. K. (2008). The dual-state theory of prefrontal cortex dopamine function with relevance to catechol-o-methyltransferase genotypes and schizophrenia. *Biological psychiatry*, 64(9), 739-749.
- Erb, C. D., Moher, J., Sobel, D. M., & Song, J. H. (2016). Reach tracking reveals dissociable processes underlying cognitive control. *Cognition*, 152, 114-126.
- Fleming, S. M., Putten, E. J., & Daw, N. D. (2018). Neural mediators of changes of mind about perceptual decisions. *nature neuroscience*, 1.

- Goodale, M. A., Pélisson, D., & Prablanc, C. (1986). Large adjustments in visually guided reaching do not depend on vision of the hand or perception of target displacement. *Nature*, 320(6064), 748-750. doi:10.1038/320748a0
- Goschke, T. (2014). Dysfunctions of decision-making and cognitive control as transdiagnostic mechanisms of mental disorders: advances, gaps, and needs in current research. *International journal of methods in psychiatric research*, 23(S1), 41-57. doi:10.1002/mpr.1410
- Hahne, J., Dahmen, D., Schuecker, J., Frommer, A., Bolten, M., Helias, M., & Diesmann, M. (2017). Integration of continuous-time dynamics in a spiking neural network simulator. *Frontiers in neuroinformatics*, 11, 34.
- Hsieh, T. Y., Liu, Y. T., & Newell, K. M. (2017). Submovement control processes in discrete aiming as a function of space-time constraints. *PloS one*, 12(12), e0189328.
- Kawabe, T. (2013). Inferring sense of agency from the quantitative aspect of action outcome. *Consciousness and cognition*, 22(2), 407-412.
- Khalighinejad, N., Schurger, A., Desantis, A., Zmigrod, L., & Haggard, P. (2018). Precursor processes of human self-initiated action. *Neuroimage*, 165, 35-47.
- Kilintari, M., Bufacchi, R. J., Novembre, G., Guo, Y., Haggard, P., & Iannetti, G. D. (2018). High-precision voluntary movements are largely independent of preceding vertex potentials elicited by sudden sensory events. *The Journal of physiology*, 596(16), 3655-3673.
- Lo, C. C., & Wang, X. J. (2006). Cortico-basal ganglia circuit mechanism for a decision threshold in reaction time tasks. *nature neuroscience*, 9(7), 956-963.
- Loftus, E. F. (2005). Planting misinformation in the human mind: A 30-year investigation of the malleability of memory. *Learning & Memory*, 12(4), 361-366.
- Mathys, C. D., Daunizeau, J., Friston, K. J., & Stephan, K. E. (2011). A Bayesian foundation for individual learning under uncertainty. *Frontiers in human neuroscience*, 5, 39.
- Mathys, C. D., Lomakina, E. I., Daunizeau, J., Iglesias, S., Brodersen, K. H., Friston, K. J., & Stephan, K. E. (2014). Uncertainty in perception and the Hierarchical Gaussian Filter. *Frontiers in human neuroscience*, 8, 825.
- Miller, P. (2016). Dynamical systems, attractors, and neural circuits. *F1000Research*, 5.
- Miyake, A., Friedman, N. P., Emerson, M. J., Witzki, A. H., Howerter, A., & Wager, T. D. (2000). The unity and diversity of executive functions and their contributions to complex “frontal lobe” tasks: A latent variable analysis. *Cognitive psychology*, 41(1), 49-100.
- Moher, J., & Song, J. H. (2014). Perceptual decision processes flexibly adapt to avoid change-of-mind motor costs. *Journal of vision*, 14(8), 1. doi:10.1167/14.8.1
- O'Reilly, R. C. (2010). The what and how of prefrontal cortical organization. *Trends in neurosciences*, 33(8), 355-361.
- Pastor-Bernier, A., & Cisek, P. (2011). Neural correlates of biased competition in premotor cortex. *Journal of Neuroscience*, 31(19), 7083-7088.
- Phillips, J. G., & Triggs, T. J. (2001). Characteristics of cursor trajectories controlled by the computer mouse. *Ergonomics*, 44(5), 527-536.
- Resulaj, A., Kiani, R., Wolpert, D. M., & Shadlen, M. N. (2009). Changing your mind: A computational mechanism of vacillation. *Nature*, 461(7261), 263-266. doi:10.1038/nature08275
- Rorie, A. E., Gao, J., McClelland, J. L., & Newsome, W. T. (2010). Integration of sensory and reward information during perceptual decision-making in lateral intraparietal cortex (LIP) of the macaque monkey. *PloS one*, 5(2), e9308.
- Rushworth, M. F. S., Kolling, N., Sallet, J., & Mars, R. B. (2012). Valuation and decision-making in frontal cortex: one or many serial or parallel systems? *Current opinion in neurobiology*, 22(6), 946-955. doi:10.1016/j.conb.2012.04.011
- Sato, A., & Yasuda, A. (2005). Illusion of sense of self-agency: discrepancy between the predicted and actual sensory consequences of actions modulates the sense of self-agency, but not the sense of self-ownership. *Cognition*, 94(3), 241-255.
- Scherbaum, S., & Dshemuchadse, M. (2019). Psychometrics of the continuous mind: Measuring cognitive sub-processes via mouse tracking. *Memory & Cognition*, 1-19.

- Scherbaum, S., Gottschalk, C., Dshemuchadse, M., & Fischer, R. (2015). Action dynamics in multitasking: The impact of additional task factors on the execution of the prioritized motor movement. *Frontiers in psychology*, 6, 934.
- Shadlen, M. N., & Shohamy, D. (2016). Decision making and sequential sampling from memory. *Neuron*, 90(5), 927-939.
- Thura, D., & Cisek, P. (2014). Deliberation and commitment in the premotor and primary motor cortex during dynamic decision making. *Neuron*, 81(6), 1401-1416.
- Todorov, E. (2004). Optimality principles in sensorimotor control. *nature neuroscience*, 7(9), 907-915.
- Todorov, E., & Jordan, M. I. (2002). Optimal feedback control as a theory of motor coordination. *nature neuroscience*, 5(11), 1226.
- van den Berg, R., Anandalingam, K., Zylberberg, A., Kiani, R., Shadlen, M. N., & Wolpert, D. M. (2016). A common mechanism underlies changes of mind about decisions and confidence. *Elife*, 5, e12192. doi:10.7554/eLife.12192.001
- Wispiński, N. J., Gallivan, J. P., & Chapman, C. S. (2020). Models, movements, and minds: bridging the gap between decision making and action. *Annals of the New York Academy of Sciences*, 1464(1), 30-51.
- Wolpert, D. M., & Ghahramani, Z. (2000). Computational principles of movement neuroscience. *nature neuroscience*, 3, 1212-1217. doi:10.1038/81497
- Wolpert, D. M., & Landy, M. S. (2012). Motor control is decision-making. *Current opinion in neurobiology*, 22(6), 996-1003. doi:10.1016/j.conb.2012.05.003
- Wong, K. F., & Wang, X. J. (2006). A recurrent network mechanism of time integration in perceptual decisions. *Journal of Neuroscience*, 26(4), 1314-1328.
- Yan, H., Zhang, K., & Wang, J. (2016). Physical mechanism of mind changes and tradeoffs among speed, accuracy, and energy cost in brain decision making: Landscape, flux, and path perspectives. *Chinese Physics B*, 25(7), 078702.
- Yeung, N., & Summerfield, C. (2012). Metacognition in human decision-making: confidence and error monitoring. *Phil. Trans. R. Soc. B*, 367(1594), 1310-1321.
- Yoo, S. B. M., & Hayden, B. Y. (2018). Economic Choice as an Untangling of Options into Actions. *Neuron*, 99(3), 434-447.
- Zgonnikov, A., Atiya, N. A. A., O'Hora, D., Rañò, I., & Wong-Lin, K. F. (2019). Beyond reach: Do symmetric changes in motor costs affect decision making?: A registered report. *Judgment and Decision Making*, 14(4), 455-469.

REVIEWERS' COMMENTS

Reviewer #1 (Remarks to the Author):

The authors' response and their revised manuscript have satisfactorily and thoroughly addressed my previous comments and concerns.

Reviewer #2 (Remarks to the Author):

We applaud Loffler and colleagues for their incredibly comprehensive response to our concerns as well as the concerns of the other reviewers and editor. We now find this manuscript suitable for publication in Nature Communications and commend the authors on an excellent paper.

The only minor quibble we would raise is with their continued insistence that a vertical CoM is more likely a "motor" rather than "intentional" change of mind. While we find their argument about the lack of a correlation to RT to be compelling, we still believe a significant percentage of these changes likely arise due to genuine changes in "preference" (itself a multifaceted concept). In our own lab, we see similar percentages of CoM's occurring in completely preference based reach-decisions. We would contend that even if the underlying preference is stable, other factors like boredom or novelty seeking might - on a given trial - create a different decision context. Since we don't know anything about how these difference biases develop over time, it is tough to make claims about the type of CoM based on the timing.

Regardless, this is more a point for further thought and we don't feel any changes need to be made to the paper.

Reviewer #3 (Remarks to the Author):

The authors have addressed all comments raised. The manuscript is read well and I do not have further comments.